# Axon tension regulates fasciculation/defasciculation through the control of axon shaft zippering

Daniel Šmít[1,2,3†], Coralie Fouquet[3†], Frédéric Pincet[4], Martin Zapotocky[1,2]*, Alain Trembleau[3]*

[1]Institute of Physiology, Czech Academy of Sciences, Prague, Czech Republic; [2]Institute of Biophysics and Informatics, First Faculty of Medicine, Charles University, Prague, Czech Republic; [3]Neuroscience Paris Seine – Institute of Biology Paris Seine, CNRS UMR8246, INSERM U1130, Sorbonne Universités, Paris, France; [4]Laboratoire de Physique Statistique, Ecole Normale Supérieure, PSL Research University, Université Paris Diderot Sorbonne Paris Cité - Sorbonne Universités, Paris, France

**Abstract** While axon fasciculation plays a key role in the development of neural networks, very little is known about its dynamics and the underlying biophysical mechanisms. In a model system composed of neurons grown ex vivo from explants of embryonic mouse olfactory epithelia, we observed that axons dynamically interact with each other through their shafts, leading to zippering and unzippering behavior that regulates their fasciculation. Taking advantage of this new preparation suitable for studying such interactions, we carried out a detailed biophysical analysis of zippering, occurring either spontaneously or induced by micromanipulations and pharmacological treatments. We show that zippering arises from the competition of axon-axon adhesion and mechanical tension in the axons, and provide the first quantification of the force of axon-axon adhesion. Furthermore, we introduce a biophysical model of the zippering dynamics, and we quantitatively relate the individual zipper properties to global characteristics of the developing axon network. Our study uncovers a new role of mechanical tension in neural development: the regulation of axon fasciculation.

*For correspondence:
zapotocky@biomed.cas.cz (MZ);
alain.trembleau@upmc.fr (AT)

[†]These authors contributed equally to this work

**Competing interests:** The authors declare that no competing interests exist.

## Introduction

In the developing nervous system, complex neural networks are built through the growth of axons from their neuronal cell body of origin toward their target(s), according to specific pathfinding patterns (*Chédotal and Richards, 2010*). These patterns are genetically controlled by molecular cues mediating interactions between axons and their environment, including other axons, cells and the extracellular matrix (*Kolodkin and Tessier-Lavigne, 2011*). The establishment of axonal projections from a given neural tissue to its final target is in many cases a multistep process, in which individual axons are sequentially guided from one area to another by a series of cues inducing specific decisions at the level of axonal growth cones (*Mann et al., 2004*). In many organisms and especially in vertebrates, given the generally high number of neurons generated in the various areas, and the need for their massive projections from one area to another, a fundamental principle governing axon pathfinding resides in the control of the fasciculation and defasciculation of their axons. This control is believed to be exerted at the level of axonal growth cones, which may choose to grow along other axons (fasciculation) or to detach from other axons (defasciculation) (reviewed in [*Honig et al., 1998*]). On the one hand, fasciculation ensures robust coordinated growth of a

**eLife digest** As an animal develops, neurons within the nervous system connect with one another to form complex networks. Each neuron has a long cable-like protrusion known as an axon that establishes connections with other neurons. The axon has a structure called the growth cone at its tip, which navigates toward its target in response to signals produced by the surrounding tissues.

Newly growing axons may bundle together or with previously grown axons, which helps them to move along a common path. Individual axons can later detach from the bundle to reach their specific target. It is generally thought that the growth cone controls axon bundling by latching on to the shaft of a neighboring axon and then moving along it. However, this viewpoint does not take into account possible dynamic adjustments in the adhesion of the shafts behind the growth cone.

Šmít, Fouquet et al. have now grown neural explants taken from the nasal tissue of mouse embryos in the laboratory and used video microscopy to record how the axons grew. The growing axons formed progressively larger bundles without direct involvement from the growth cones. Instead, the shafts of the axons stuck together in a way that resembles fastening a zipper.

Šmít, Fouquet et al. manipulated the 'axon zippers' and observed that zippering arises from a competition between two forces: the contact force that causes two axons to adhere to each other (which favors zippering) and the mechanical tension that arises from internal or external pulls on the axon (which favors unzippering). More research is now needed to directly observe zippering in developing animals in order to understand how it helps the nervous system to assemble.

number of axons along the paths initially established by pioneering axons (*Raper and Mason, 2010*). On the other hand, axon defasciculation, often associated to branching, is in many cases required for individual axons to reach with precision their specific target(s), which can be distributed in large areas (*Tang et al., 1994*; *Schneider and Granato, 2003*). For example, motor axons emerging from spinal somatic motoneurons fasciculate in tight bundles, they migrate in fascicles within spinal nerves, and they thereafter defasciculate to allow each individual axon subpopulation to innervate a specific muscle cell group (*Bonanomi and Pfaff, 2010*; *Huettl et al., 2012*).

While the interaction of growth cones with other axons has been the focus of numerous studies (*Honig et al., 1998*; *Tang et al., 1994*; *Lin and Forscher, 1993*; *Van Vactor, 1998*; *Kalil, 1996*), other aspects of the process of axon fasciculation have received much less attention. In particular, while it seems obvious that tight fasciculation of axons is aided by adhesion between their shafts, very little is known about the dynamics of shaft-shaft interactions, the underlying biophysical mechanisms, and their potential role in the regulation of axon fasciculation. The aim of the present paper was to address these issues, by analyzing axon-axon interactions and the resulting fasciculation/defasciculation processes in a convenient setting. We chose the mouse olfactory epithelium as a model system. This system has the advantage of comprising a single population of neurons, the olfactory sensory neurons (OSNs). During their normal development from the olfactory epithelium (OE) toward their target in the olfactory bulb (OB), OSN axons undergo a massive fasciculation step to form branches of the olfactory nerve, followed by their defasciculation and rearrangement in the OB to project to their specific target cells distributed throughout the OB glomerular layer (*Key et al., 2002*; *Nedelec et al., 2005*; *Strotmann and Breer, 2006*; *Mombaerts, 2006*; *Mori and Sakano, 2011*).

Since it is currently technically challenging to image mouse OSN axon fasciculation and defasciculation dynamics in vivo, and impossible to manipulate in vivo the individual axons in order to assess their biophysical properties, we chose to perform our study on embryonic OE cultured explants, grown on a permissive planar substrate. Using time lapse imaging, we recorded OSN axons as they grow from the explants, and characterized their dynamic interactions. Surprisingly, we observed that OSN axons interact extensively with each other through their axon shafts, leading to zippering and unzippering behaviors that trigger their fasciculation or defasciculation, respectively. In the present paper, we characterize the dynamics of these axon-axon shaft interactions, assess quantitatively the biophysical parameters of these processes, and develop a biophysical model of this dynamics. Micro-manipulations of individual zippers, as well as pharmacologically induced perturbations of the

fasciculated network, are used to demonstrate unzippering by forces of functionally relevant magnitude. Our analysis supports a framework in which axonal mechanical tension regulates fasciculation through the control of axon shaft zippering.

## Results

### Progressive fasciculation in cultures of olfactory epithelium explants is due to axon shaft zippering

OSN axons grow in cultures as unbranched axons. In our experimental conditions (see Materials and methods), the growth of these axons from OE explants was characterized by a sequence of three main stages: (1) advance of the growth cones and initiation of an axon network (first 24–48 hr), (2) maintenance of the growth cones at distance from the explants, but with little or no further growth (48–72 hr), and (3) retraction of the growth cones and collapse of the network (3–5 d). We analyzed in detail the intermediate stage, during which axon shafts interact and form bundles.

During this stage, the initial axon network, composed of individual axons or bundles of few axons, progressively evolves into a less dense network with thicker bundles, indicating that individual axons and/or small bundles fasciculate together to form larger bundles (*Figure 1* and *Video 1*). To characterize this process quantitatively, we selected a typical area of the network and manually segmented (see Materials and methods) the images into vertices (junction points and crossings of axon segments) and edges (lines connecting the vertices). *Figure 1D–G* shows the results of such a segmentation over a 178-min time interval, the red dashed lines representing the segmented edges, and star symbols representing the vertices of the network. Based on this image analysis, we determined the total length of the network and the total number of vertices (junction points), and found that both these quantities decreased approximately linearly with time in the interval examined (*Figure 1H*). This trend was observed in five out of six quantitatively analyzed experiments from different cultures, with an average reduction of $(20 \pm 16)\%$ in length of the network over the duration of the recordings (178 to 295 min). Topologically, such dynamics is reminiscent of the well-known coarsening of two-dimensional foams (*Glazier and Weaire, 1992*; *Weaire and Hutzler, 2001*); the underlying structures and forces, however, are substantially different (see Discussion).

To understand the processes that lead to this coarsening of the axon network, we examined its dynamics on finer time scales. On time intervals of the order of minutes, we observed elementary zippering processes, as shown in *Figure 2A,B*. In an advancing zippering process, two axons or axon fascicles progressively adhere to each other in a longer segment of contact and form a larger fascicle. Receding zippers leading to defasciculation of axons were also observed. The zipper vertex at the meeting point of the axons moves with a velocity of the order of $1 \frac{\mu m}{min}$ until it reaches a position of equilibrium (*Figure 2A*). In the example of *Figure 2B*, two adjacent advancing zippers lead to a clear decrease in the total length of the network.

Numerous and frequent zippers were observed throughout the network, as demonstrated in *Figure 3*, showing a selected time interval from the network dynamics of *Figure 1*. Blue arrows in *Figure 3A–C* point to vertices that will zipper in the following frames. Red dashed lines with arrows show the resulting zippered segment. The zippering processes in the upper left corner (the area marked by rectangle in *Figure 3E* and enlarged in *Figure 3G–J*) lead to a reduction in the number of vertices, from 3 to 1 (star symbols in *Figure 3G and J*).

### Axon shafts adhere to form two types of zippers: simple or entangled

To assess the axonal structure of the zippers, we performed scanning electron microscopic analyses of our cultures (*Figure 4A*). We observed at high magnification that numerous network vertices displayed structures as shown in *Figure 4B,C*, in which individual or small bundles of axons are adherent to each other along a defined segment, while remaining parallel to each other. A zipper with such structure is free to increase or decrease the length of the zippered segment, depending on the balance of the forces acting at the zipper vertex; we call such zippers 'simple zippers'.

More rarely, we observed zippers with entangled structure (*Figure 4E,F*). Such structure may prevent the zipper from unzippering past the entanglement point; further zippering, however, remains possible. In some instances, axons would cross on top of each other, without forming a zippered segment (*Figure 4D*). Such crossings (identified at the light microscopy level by a lack of visible

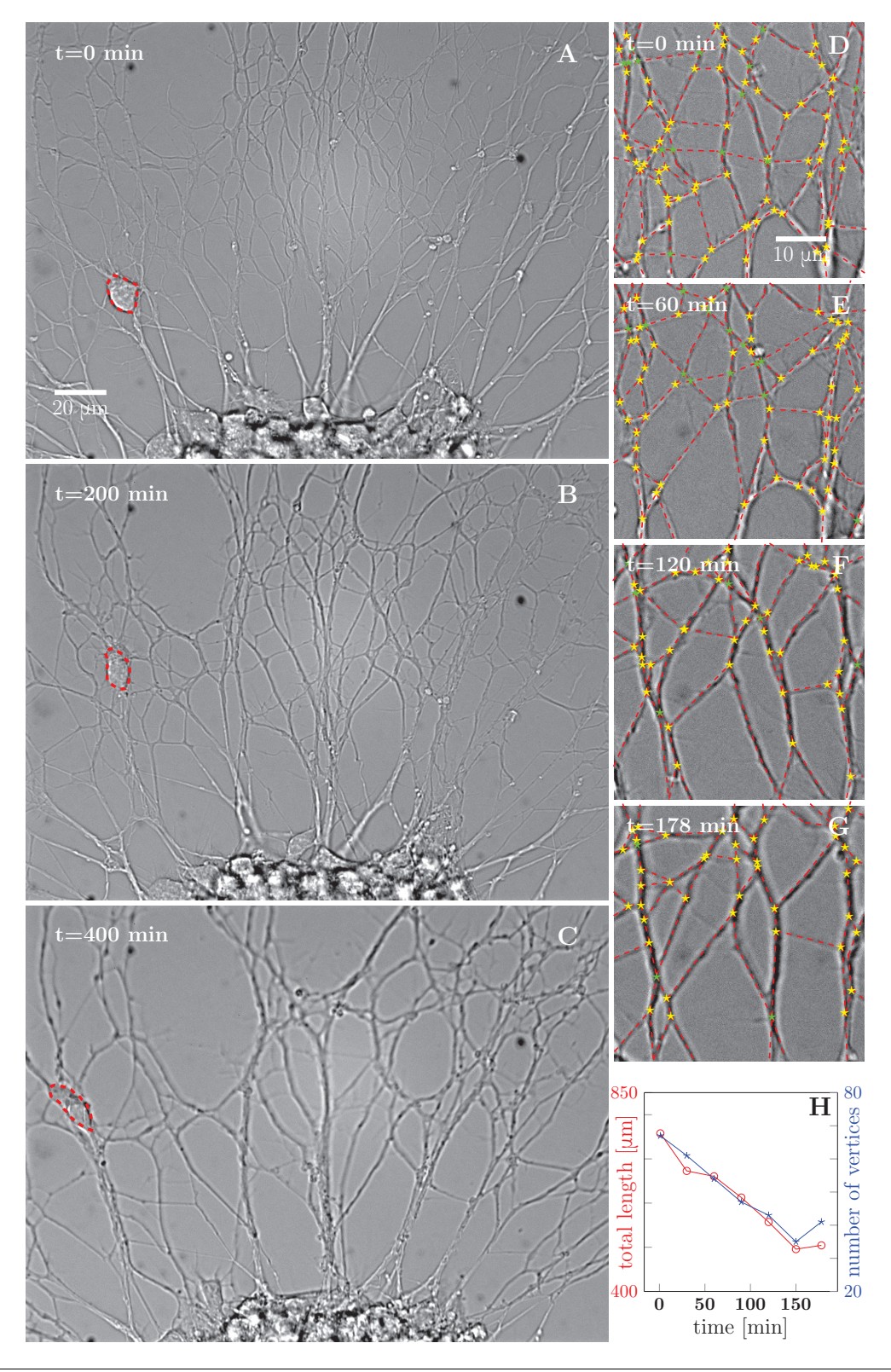

**Figure 1.** Progressive formation of fascicles in the evolving axon network. (A–C) Evolution of the axonal network growing from an explant during 400-min time lapse recording, after 2 days of incubation; the red dashed outline delineates a travelling ensheating cell. Progressive coarsening of the network and decrease of total length and density can be seen. (D–G) Red dashed lines outline the edges of the network, while yellow stars indicate junctions between axons or axon bundles, and green stars indicate crossings. (H) Quantification of total length and number of vertices of the network

*Figure 1 continued on next page*

*Figure 1 continued*

area depicted in panels (**D–G**), as a function of time (based on seven manually segmented video frames). Segmentation coordinates for panels D-G and data from panel H data are available in *Figure 1—source data 1*.

The following source data and figure supplements are available for figure 1:

**Source data 1.** Segmentation coordinates (**D–G**), plot data (**H**).

**Figure supplement 1.** An example of spontaneous defasciculation correlated with explant contraction.

**Figure supplement 1—source data 1.** Development of axon network over 135 min, with decoarsening visible in the lower right quadrant after $t$=65 min, video corresponding to *Figure 1—figure supplement 1*.

adherent segment and by no change in the axon direction) are marked by green symbols in *Figure 1D–G* and were not counted in the totals of *Figure 1H*. While the entangled zippers cannot be distinguished from the simple zippers with light microscopy, we examined high-magnification (1000×) SEM pictures (*Figure 4—figure supplement 1*) to find the following abundances of the three types of network vertices: 54% (134 out of 247 vertices) were simple zippers, 28% (69 out of 247) entangled zippers, and 18% (44 out of 247) crossings.

Besides axon shafts and their bundles, thin lateral protrusions emerging from the shafts are observed in the network (*Figure 4G*). These protrusions appear highly dynamic and occasionally mediate interactions at a distance when they extend and touch another axon shaft.

The observations reported above indicate that the progressive coarsening of the axon network results from zippering events driven by adhesion between the axon shafts.

## Manipulation of axon tension alters the relative abundance of zippering and unzippering

In recorded time lapse sequences of network evolution in 13 explants, we typically observed that the axon network coarsened in a manner similar to *Figure 1*, or in some cases appeared stable when the recording was performed over shorter time intervals. This indicates that, in general, zippering events dominated over unzippering events. In a few cases, however, a de-coarsening dynamics was seen in a limited area of the network. Upon examining these cases, we noticed that they were associated with an apparent contraction of the explant, thus generating a pulling force on the axonal network (*Figure 1—figure supplement 1*). Stimulated by this observation, we sought a pharmacological manipulation by which a similar effect could be induced in a controlled manner.

First, we envisaged treatments aiming at rapidly enhancing growth cone motility, in view of increasing the pulling force exerted by the GC on axon shafts. Since the molecular cues having such effects on OSN explant cultures remain unknown, we tested in a first approach Foetal Bovine Serum (FBS), assuming that some of growth factors it contains may stimulate axon outgrowth. Interestingly, while no obvious effect on the growth cones was observed upon FBS addition to the culture, we found that the application of 5% FBS reliably induced the explant pull. This is likely due to a cell-rounding effect of FBS on cultured neurons, previously demonstrated in *Jalink and Moolenaar (1992)*. In *Figure 5A–D*, an example is shown, with a resulting de-coarsening in the axon network. Often, however, the FBS-induced pull resulted instead in a rapid collapse of the entire axon network onto the explant, due presumably to a disturbance of the attachments of the axons to the substrate.

As an alternative means to influence axon tension, we tested blebbistatin, a well known inhibitor of neuronal Myosin II (NMII), previously

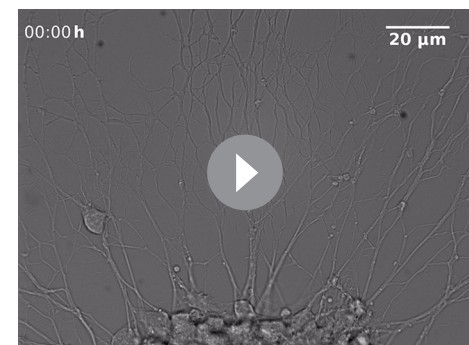

**Video 1.** Development and coarsening of axon network over 12 h, corresponding to *Figure 1*.

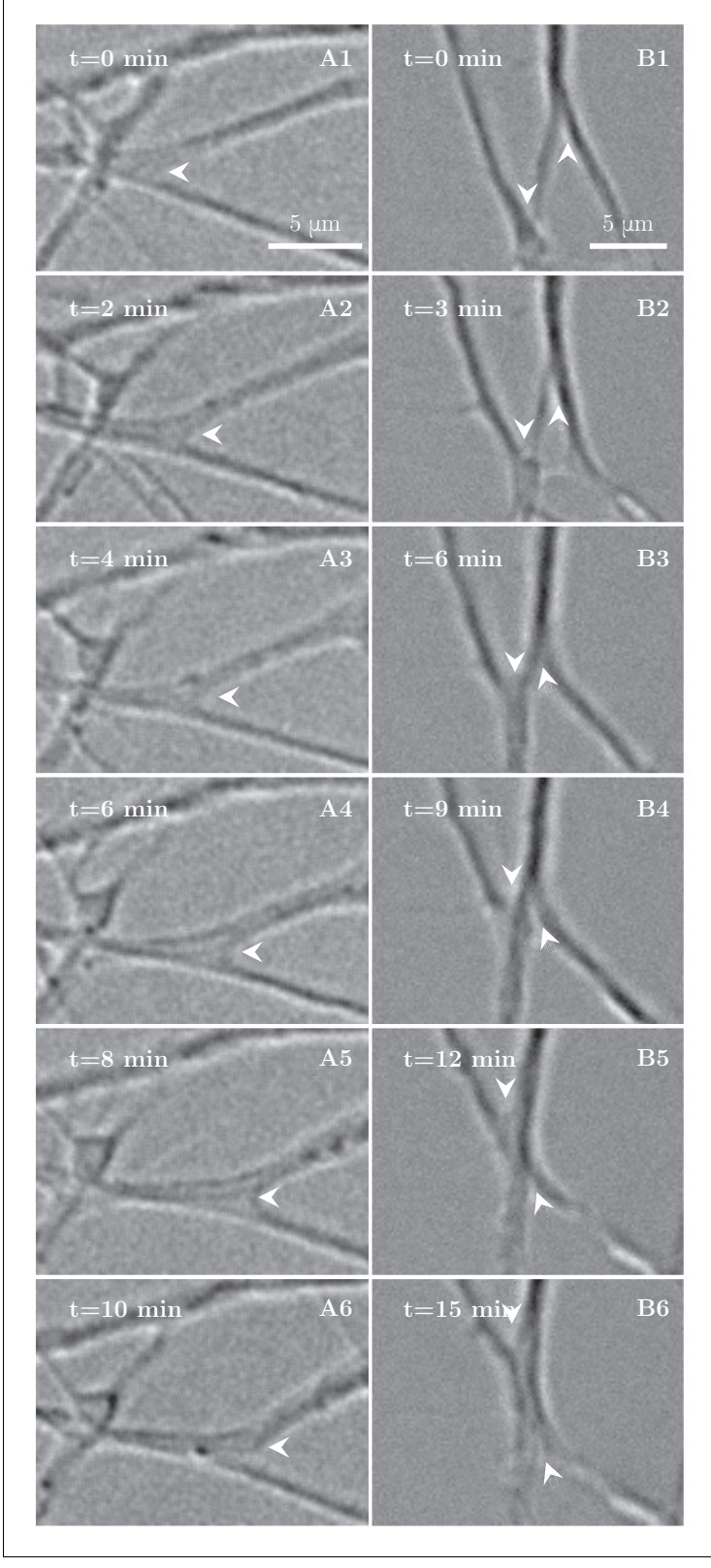

**Figure 2.** High-magnification images of individual axon zippers and their evolution in time. Zipper vertices are marked by arrowheads. (**A**) advancing zipper, (**B**) two associated advancing zippers. The total length of the network segments in B decreases during the zippering process.

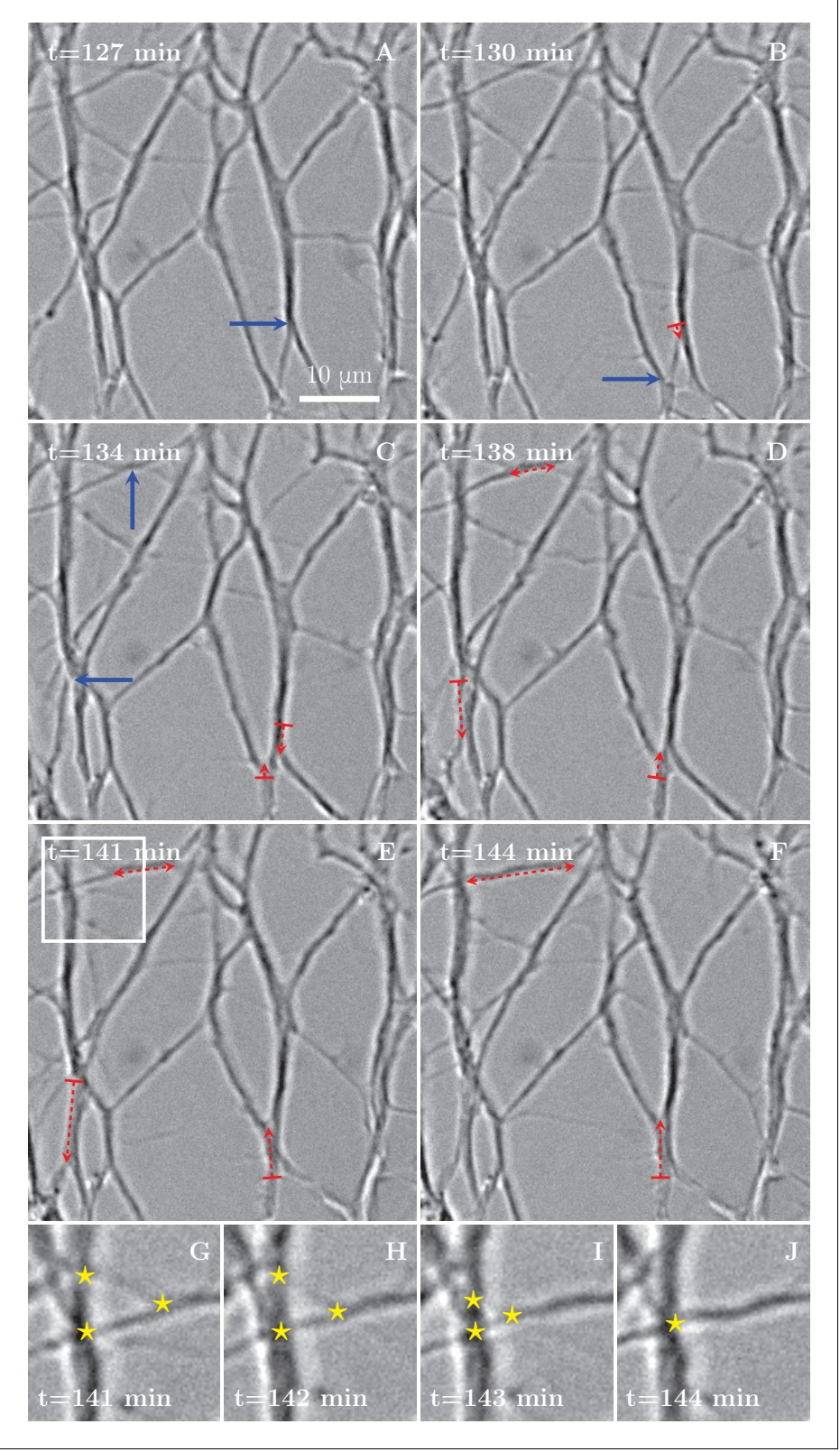

**Figure 3.** Zippering events drive the progressive formation of fascicles. (**A–F**) Six frames extracted from the time sequence shown in *Figure 1D–G*. The blue arrows indicate vertices that will start to zipper in the following frame, the red dashed arrows illustrate the direction and the increase in length of the advancing zipper. If two arrowheads are present, there are two vertices extending a single segment. Frames **G–J** are enlargements of the

*Figure 3 continued on next page*

*Figure 3 continued*

inset in panel E in the period between the frames E and F, illustrating three vertices (marked by stars in frames G to I) merging into a single vertex (in frame J).

shown to decrease cell cortex/membrane tension in a variety of non-neuronal cells (*Fischer-Friedrich et al., 2014*; *Ayala et al., 2017*). Somewhat surprisingly, in our culture system 10 µM blebbistatin (dissolved in dimethyl sulfoxide (DMSO)) did not show visible effect on axon tension, but rather had a stabilizing effect on the network: the spontaneous coarsening was inhibited while the individual zippers remained mobile. No such effect was observed in control experiments in which only DMSO was added.

We took advantage of the stabilizing effect of blebbistatin to facilitate the analysis of the coarsening or de-coarsening effects generated by subsequent treatments. In *Figure 5E–H* and *I–L*, two examples are shown in which FBS was applied to the pre-stabilized network and rapidly induced clear de-coarsening in parts of the network, with unzippering /defasciculation dominating over zippering events.

In the example of *Figure 5E–H*, the edge of the explant (visible near the left border of each frame) retreated by approximately 20 µm to the left, thus stretching the axon network in the horizontal direction. *Figure 5—figure supplement 1* evaluates how three candidate axon paths were deformed during the first 20 min after FBS was added (i.e. in between the frames F and G of *Figure 5*). The lengths of the paths increased by (8–23) µm, that is by 4–10% (panel E in *Figure 5—figure supplement 1*), while at the same time, the paths tended to become more straight (panel F). The axon segments thus became significantly stretched and also aligned in the direction of the pull, as expected for an object under increased tension. The stretching of the axons by ~15 µm is expected to result in a significant tension increase of over 1 nN (see Discussion). This tension increase is achieved within 20 min of FBS addition and precedes the changes in network configuration seen in *Figure 5G–H*.

As a complement to the observed de-coarsening induced by a pulling force, we sought to perturb the network dynamics by decreasing the tension in the axons. In previous literature, cytochalasin, an inhibitor of actin polymerization, was shown to significantly lower the tension of PC-12 neurites (*Dennerll et al., 1988*). Indeed we found that in our system, 2 µM cytochalasin B (dissolved in DMSO) induces a change in network dynamics consistent with a drop of average axon tension. In the example in *Figure 6A–C*, the application of cytochalasin B induced coarsening (panel C) in a network that was previously stable (panels A and B). As the networks generally have a tendency to coarsen, we sought to better isolate the effect of cytochalasin by applying it to networks that were pre-stabilized by blebbistatin. As shown in the example frames in *Figure 6D–F* and in the graphs in *Figure 6G–I*, cytochalasin B induces strong network coarsening within 30 min of application. No such effect was observed in control experiments in which only DMSO was applied.

## Force balance at the level of a zipper: competition of axon tension and axon-axon adhesion

To understand the conditions leading to zippering or unzippering, we analyze the force balance in a zipper of two axons (*Figure 7*). The contributing forces originate from the mechanical tension in each axon, the adhesion between each axon and the substrate, and the adhesion between axons in the zippered segment. In the following, we combine the mechanical tension $T_0$ (i.e. tensile energy per unit length of the axon) and the axon-substrate adhesion (i.e. energy of adhesion per unit length of the axon) into an effective tension parameter $T$. The zipper will be in static equilibrium when the effective tensile forces are in balance with the force of adhesion between the axons. Consider for simplicity a symmetric zipper, in which the tensions in both axons are equal to each other, $T = T_1 = T_2$. At the vertex of the zipper (*Figure 7A*), the force balance condition in the direction parallel to the zippered segment is given by

$$S = 2T\left(1 - \cos\frac{\beta}{2}\right) \tag{1}$$

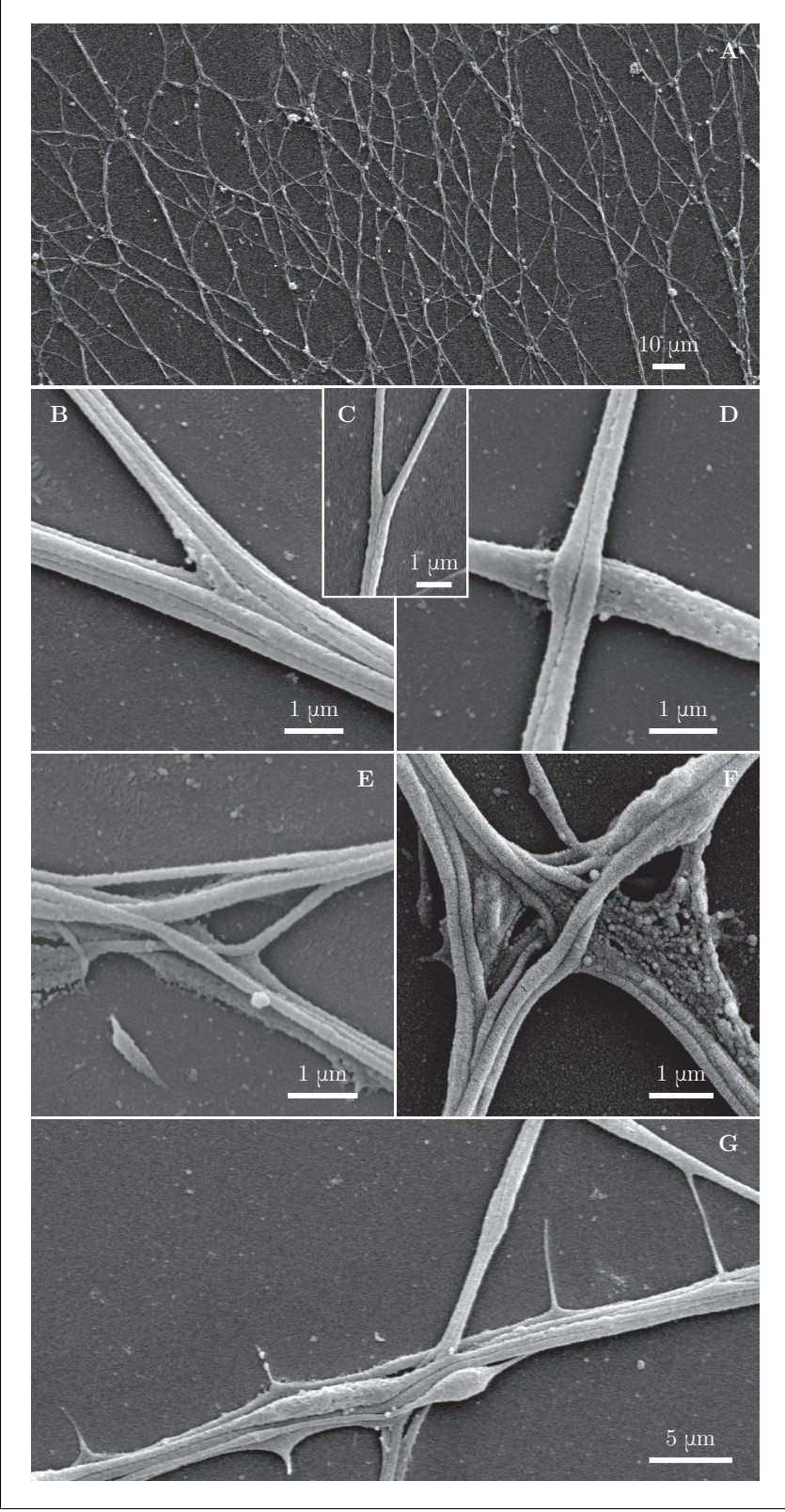

**Figure 4.** Fine morphological characterization of zippers with scanning electron microscopy. Panel (A) illustrates a large area of the culture observed at low magnification. (B–C) illustrate a laminar vertex structure formed between small axon bundles (B) or between individual axons (C). (D) illustrates crossing of axons. (E) and (F) illustrate more

*Figure 4 continued on next page*

*Figure 4 continued*

complex, entangled vertices. Such configurations are unlikely to easily unzipper. (G) shows thin lateral protrusions, often seen along axon shafts. These protrusions can attach to nearby axons and pull on them.

The following source data and figure supplements are available for figure 4:
**Figure supplement 1.** Quantification of abundance of axon crossings, simple zippers and entangled zippers.
**Figure supplement 1—source data 1.** Coordinates of marked points.

where $\beta$ is the zipper angle (*Figure 7A*) and $S$ is the force arising due to the adhesion between the axon shafts. The force $S$ may also be understood as the axon-axon adhesion energy per unit length of the zippered segment, and *Equation 1* derived from the minimization of total energy (see Materials and methods). If the tension $T$ changes so that the equilibrium condition (*Equation 1*) is no longer satisfied, zippering (in the case of tension decrease) or unzippering (in the case of tension increase) will result until a new equilibrium value of the zipper angle $\beta$ is reached. Such changes in axon tension may occur due to a rearrangement of the network configuration in the vicinity of the zipper (and hence a change in forces pulling on the segments of the zipper), or directly from changes in the basal tension generated by the pull of the growth cones and/or shaft cytoskeletal activity.

The balance of forces at a junction of three axon segments was previously considered in (*Bray, 1979*; *Condron and Zinn, 1997*; *Shefi et al., 2004*; *Anava et al., 2009*), and used to analyze the distribution of tensions in a branched axon network. Provided that the junction is not strongly attached to the substrate (*Shefi et al., 2004*), at a static branch point the tension force vectors in the three segments must add to zero (and there is no axon-axon adhesion force). Breaking this balance results in a fast shift in the position of the branch point and adjustment of the branch angles (*Condron and Zinn, 1997*); however, the material composition of the branches does not immediately change. In contrast, when the force balance of *Equation 1* is broken, new portions of the two unbranched zippering axons are brought into contact, increasing the length of the zippered segment at the expense of the unzippered segments.

## Measurement of axon tension allows to estimate the axon-axon adhesion energy

To support the explanatory framework presented above, we carried out micro-manipulation experiments designed to measure the magnitude of the inter-axon adhesion force $S$ and to investigate the dynamics of individual zippers. To determine $S$, we used observations of zippers in static equilibrium combined with measurements of the axon tension $T$. As seen from *Equation 1*, the knowledge of $T$ and of the zipper angle $\beta$ permits to calculate the magnitude of the adhesion force $S$.

It is known from previous literature that the typical value of mechanical tension in an isolated axon grown in culture is of the order of 1 nN ([*Dennerll et al., 1988*] reports a wide range of rest tension values, with the most common tension around 0.5 nN). Approaches using calibrated needles or Microelectromechanical systems (MEMS) had been successfully used to measure the tension of axons of dorsal root ganglia (DRG) neurons and PC-12 neurites (*Dennerll et al., 1988*, *1989*), as well as motor neuron axons in Drosophila embryo (*Rajagopalan et al., 2010*). In our case, the small diameter of OSN axons (about 200 nm) makes the use of such approaches difficult, because of the likely physical contact of the manipulator with the substrate resulting in an incorrect force reading, as well as in the detachment of the axon from the substrate. Optical tweezers technique would in principle allow the manipulation using microbeads attached to axons without touching the substrate, but does not permit to achieve manipulation forces comparable to 1 nN. Therefore, we decided to use the Biomembrane Force Probe (BFP) technique, in which a red blood cell is used as a force transducer (*Figure 8*). In this technique, streptavidin-coated glass microbeads (3 μm diameter) attached to biotinylated axons are manipulated by a biotinylated red blood cell aspirated in a micropipette (*Evans et al., 1995*; *Gourier et al., 2008*). By measuring the deformation of the red blood cell, one can calculate the force with which the bead is manipulated.

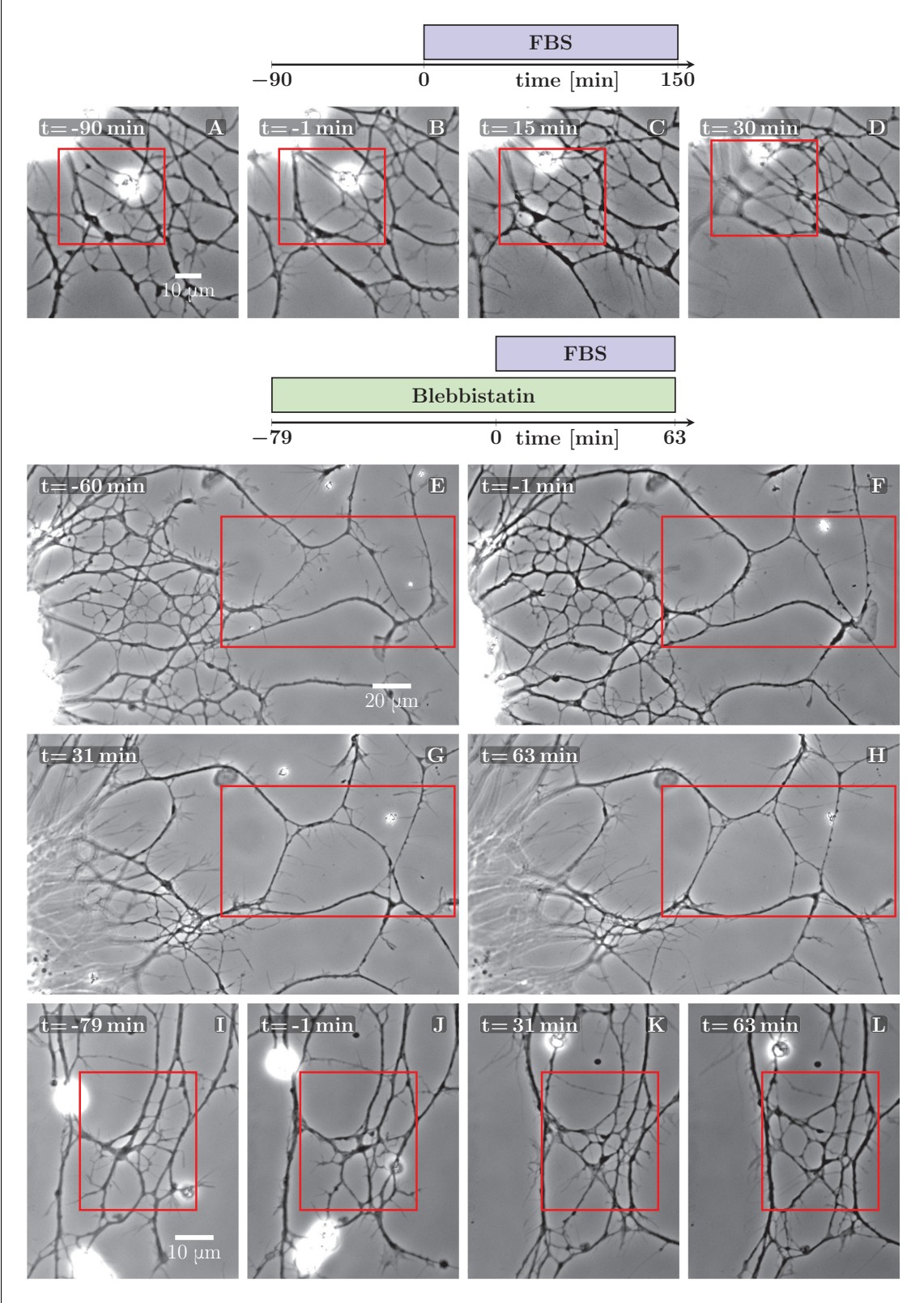

**Figure 5.** Defasciculation resulting from FBS-induced pull on the network. The schemes indicate the protocol of drug addition for the experiments that are shown on the frames below the schemes. (A–D) FBS was added to the culture at $t=0$ min. Decoarsening of part of the network (marked by the red rectangle) is visible. At a later stage, the network collapses. The full recording is provided as *Figure 5—source data 1*. (E–H) The culture was pretreated by blebbistatin added before $t=-60$ min. Little change is visible between $-60$ and $-1$ min. At $t=0$ min FBS is added, after which progressive

*Figure 5 continued on next page*

*Figure 5 continued*

movement of the explant border to the left can be observed, exerting a pulling force on the axons. As a result, unzippering occurs, the network defasciculates and several new loops appear in the area marked by the red rectangle. The full recording is provided as *Figure 5—source data 2*. (I–L) The culture was pretreated with blebbistatin (*t*=−79 min) and the network remained mostly unchanged until FBS was added (*t*=0 min). Defasciculation is visible in the frames K and L, where the area of interest is marked by the red rectangle. The full recording is provided as *Figure 5—source data 3*.

The following source data and figure supplements are available for figure 5:

**Source data 1.** Development of axon network over 240 min, treated with FBS at *t* =90 min, video corresponding to *Figure 5A–D*.

**Source data 2.** Development of axon network over 142 min, pretreated with blebbistatin, and treated with FBS at *t*=79 min, video corresponding to *Figure 5E-H*.

**Source data 3.** Development of axon network over 142 min, pretreated with blebbistatin, and treated with FBS at *t*=79 min, video corresponding to *Figure 5I-L*.

**Figure supplement 1.** Stretching of the axons due to FBS-induced pull on the network.

**Figure supplement 1—source data 1.** Coordinates of paths (A–D), plot data (E,F).

Using the BFP technique, we determined the tension in a collection of thin network segments, presumably individual axons, even though we cannot exclude that some of them might have been fascicles of several axons. The basis of the measurement is force equilibrium between the calibrated force of the probe acting on and deforming the axon, and a restoring force, which arises from the tension in the axon shaft. The measurement is described in *Figure 8*. The initially straight axon (*Figure 8A*) is deformed by displacing the micropipette and holding it in a fixed position (*Figure 8B*). The force equilibrium is reached: the pulling force $F_{\mathrm{BFP}}$ is balanced by the projection of the axon tension in the transverse direction $2T \sin \delta$, where $\delta$ is the angle of deflection of the axon (i.e. $180° - 2\delta$ is the angle at the apex of axon deformation). This operation is repeated for larger displacements (*Figure 8C*), until the red blood cell detaches from the bead, which generally occurs at a deformation angle of about $\delta \approx 5°$. *Figure 8D* shows the time course of the pulling force measured on the probe during this experiment, as well as the measured deformation angle. The force plateaux (labelled 1 to 5 in *Figure 8D* and marked by black boxes) correspond to the time intervals during which the micropipette position was held fixed. To extract the value of the tension in the axon, a linear fit is performed on the transverse projection of pulling force vs. $\sin \delta$ (*Figure 8E*). The slope of the fit line gives the tension $T = 906 \, \mathrm{pN}$ in the case of the experiment shown in *Figure 8*. The non zero intercept of the fit arises from calibration effects described in Materials and methods. Out of a several dozen measurements performed, we obtained a collection of eight measurements from seven axons that included at least three plateaux in each.

For one of the axons, two distinct values of tension were measured early and late in the experiment: (432 ± 157) pN and (1665 ± 219) pN. This increase in tension was likely caused by the strong stretching of the axon that occurred during this particular experiment—see *Figure 8—figure supplement 1*. Such stretching is unlikely to occur during spontaneous dynamics of the axon network (without added drugs), and we excluded the post-stretching data from the analysis. This experiment indicates, however, that the FBS-induced pulling (*Figure 5*) may have lead to very significant increases in axon tension.

Using the slope values and their errors calculated from the seven remaining linear fits, we estimated the distribution of the tensions in the axon population, shown in *Figure 8F*. The distribution is sharply peaked near 678 pN, with the mean value of 679 pN and the interquartile range (529–833) pN.

Technical limitations were encountered in these experiments, including the uncontrollable bead localization along the axons and with respect to zippers, as well as early detachment of beads from the red blood cell upon pulling. The seven measurements included in *Figure 8F* correspond to the most robust ones and were obtained with beads that were not necessarily in the vicinity of a zipper vertex. It was therefore not feasible to correlate the measured tension values with measured zipper

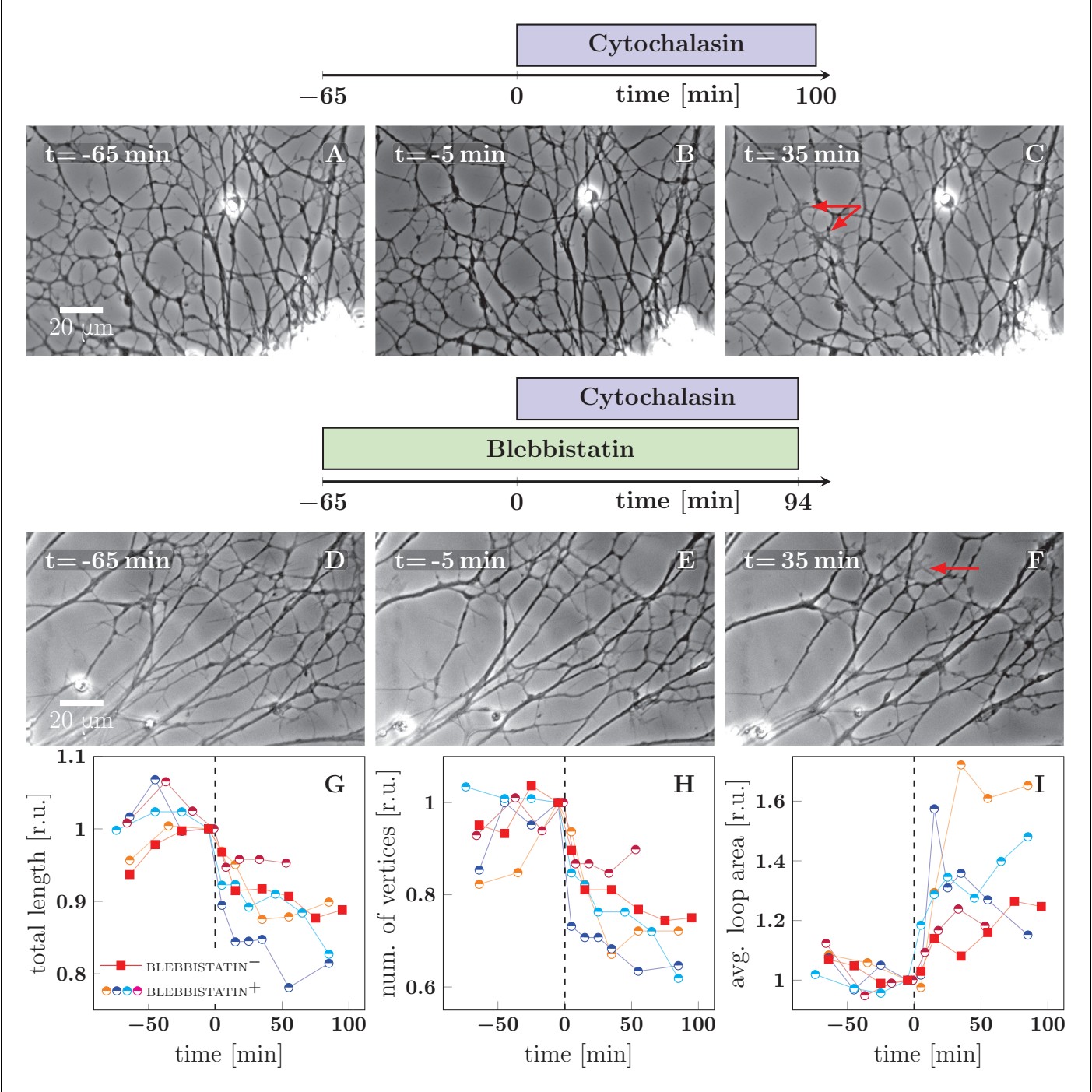

**Figure 6.** Cytochalasin-induced fasciculation of axon shafts. The schemes indicate the protocol of drug addition for the experiments that are shown on the frames below the schemes. (A–C) Cytochalasin was added to the culture at $t$=0 min. While there is little visible change between −65 and −1 min, the network exhibits coarsening between 0 and 35 min. The full recording is provided as *Figure 6—source data 2*. (D–F) The culture was pretreated with blebbistatin before $t$=−65 min. Little change is visible between −65 and −1 min. After cytochalasin addition at $t$=0 min, the culture exhibits coarsening. The full recording is provided as *Figure 6—source data 3*. The red arrows in frames C and F indicate prominent lamellipodia, which appear after the addition of cytochalasin. (G–I) The network statistics for the experiment of panel (A–C) (red squares), panel (D–F) (blue half-circles), and three other experiments with protocol equivalent to D–F, shown as *Figure 6—source data 4* (orange half-circles), *Figure 6—source data 5* (purple half-circles) and *Figure 6—source data 6* (cyan half-circles). (G) Total length of the axon network in the field. (H) Total number of vertices of the axon network in the field. (I) Average area of cordless closed loop in the axonal network. The networks were manually segmented and analyzed as indicated in Materials and methods. In (G–I), the data was aligned by the time of cytochalasin addition marked $t$=0 min and normalized by the value of the last measured timepoint before the drug was added. A sharp decrease of total length and of the number of vertices, as well as increase of average loop

*Figure 6 continued on next page*

*Figure 6 continued*

area, is seen within 30 min after *t*=0 min, indicating coarsening of the network triggered by cytochalasin addition. Segmentation data and frames are available in *Figure 6—source data 8* (please consult Materials and methods), source code used to generate the network statistics and input data is in *Figure 6—source data 1*, the data points plotted in panels G–I are in *Figure 6—source data 7*.

The following source data is available for figure 6:

**Source data 1.** ZIP archive; contains source code (Figure 6_source_code.m) and five ZIP archives with selection input data.
**Source data 2.** Development of axon network over 165 min, treated with cytochalasin at *t*=65 min, video corresponding to *Figure 6A–C* (red in graphs G–I).
**Source data 3.** Development of axon network over 159 min, pretreated with blebbistatin, and treated with cytochalasin at *t*=65 min, video corresponding to *Figure 6D–F* (blue in graphs G–I).
**Source data 4.** Video of development of axon network over 159 min, pretreated with blebbistatin, and treated with cytochalasin at *t*=65 min, orange data points in *Figure 6G–I*.
**Source data 5.** Video of development of axon network over 129 min, pretreated with blebbistatin, and treated with cytochalasin at *t*=67 min, purple data points in *Figure 6G–I*.
**Source data 6.** Video of development of axon network over 166 min, pretreated with blebbistatin, and treated with cytochalasin at *t*=75 min, cyan data points in *Figure 6G–I*.
**Source data 7.** Plot data (G, H and I).
**Source data 8.** ZIP archive; contains video frames and segmentation data underlying the analysis shown in the *Figure 6G–I*.

angles on the level of individual axons. Rather, we chose to obtain a separate set of measurements of equilibrium zipper angles.

As zippers that are entangled (as in *Figure 4E,F*) may remain static without satisfying the equilibrium *Equation 1*, we restricted our measurements to zippers that were observed to be mobile before reaching a static configuration. In the videorecordings of the developing network, we selected 17 such zippers that were approximately symmetric and appeared to consist of single axons (or possibly thin fascicles). We measured the zipper angles of the equilibrated configurations (requiring stability over at least 5 min), and based on these values estimated the distribution of equilibrium zipper angles in the zipper population (*Figure 7C*). The distribution is sharply peaked around 42°, with mean of 51.2° and interquartile range (34–60)°.

Based on the measured distributions of axon tensions and of equilibrium zipper angles, we then estimated the axon-axon adhesion force $S$. First, we assumed that the two distributions are related to each other through *Equation 1* and determined the value of $S$ resulting in their best mutual match (see Materials and methods), obtaining $S$=88 pN. In an alternative procedure, we estimated a joint distribution of the axon tensions and equilibrium zipper angles (treating the two variables as independent), and used *Equation 1* to compute the corresponding distribution of adhesion parameters $S$ (see Materials and methods). This procedure allows for the expected variability of the values of $S$ among zippers (e.g. due to different areas of contact), and gives a maximum interquartile range of $S$=(52–186) pN, with a median of 102 pN.

## Induced or spontaneous dynamics of individual zippers

To determine the axon adhesion force more directly, not relying on the measurement of axon tension, we attempted to unzipper selected zippers using a calibrated pulling force. These attempts were not successful, due to insufficient strength of the bond between the bead and the red blood cell. This resulted in the detachment of the red blood cell before any significant effect on the zipper. To overcome this limitation, we bypassed the red blood cell and bead and used the pipette to drag the axon directly. This allowed us to use forces sufficiently large to induce unzipping at the price of losing the knowledge of the force magnitude. *Figure 9* and the corresponding video *Figure 9—*

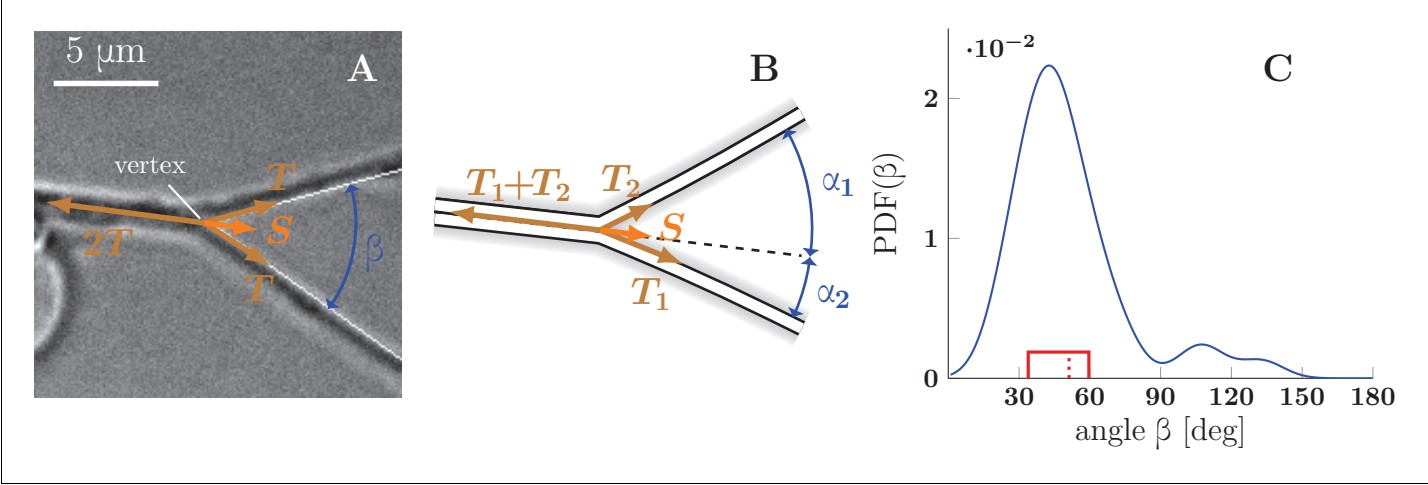

**Figure 7.** Force balance in static axon zippers. (A) Illustration of a symmetric zipper. The zipper angle $\beta$ is marked in blue. The arrows denote the vectors of tension $T$ and axon-axon adhesive force $S$. (B) illustration of an asymmetric zipper, the markings are the same as in A, but the tensions within the axons differ ($T_1 \neq T_2$). (C) distribution of initial and final equilibrium angles of measured zippers (17 zippers, 34 measurements) originating from four distinct cultures (each obtained from a different mother animal), transformed into a probability distribution using convolution with Normal kernel. The red dashed line marks the average angle value (51.2°) and the solid red box delimits the interquartile range (34°–60°). The values correspond to the full zipper angle $\beta$, which equals $\beta = \alpha_1 + \alpha_2$ in asymmetric case. Individual distributions of the angles $\alpha_1$, $\alpha_2$ were not recorded, because of prevailing symmetry of measured zippers. The distribution includes only those zippers, which were stable at least 5 min before and after the dynamics. The measured angles and the distribution of panel C are available in *Figure 7—source data 1*.

The following source data is available for figure 7:

**Source data 1.** Estimated angle distribution (C) and underlying experimental angle data.

*source data 1* show an example. By dragging one of the axons of a zipper, we increased the zipper angle beyond its equilibrium value, leading to unzippering accompanied by a decrease of angle (*Figure 9A–D*). Then, the axon was released by lifting the pipette. The axons snapped back to a smaller zipper angle which initiated a re-zippering process accompanied by an increase of the angle (*Figure 9E–F*), leading finally to the recovery of the initial configuration. Similar manipulations performed on other zippers either gave analogous results (*Video 2*), or in some instances, no unzippering (*Video 3*). However, this latter case is likely to be due to the structural organization of these particular zippers involving entangled axons (*Figure 4E,F*).

Similarly to these cases of induced unzippering/rezippering, we view the numerous individual zippering processes observed in the developing network (*Figures 2* and *3*) as arising from force perturbations that act on a zipper and move it to a new equilibrium configuration. These perturbations may consist in changes in the network geometry in the vicinity of the zipper, or in changes in mechanical tension within the axons that constitute the zipper. To characterize such spontaneous zippering dynamics, we tracked 17 individual zippering processes within the developing network and measured how the zipper configuration evolved. All 17 zippers selected for this analysis started from approximately stationary initial configurations, and reached a final configuration that remained stationary for at least five min. Selected typical examples are shown in *Figure 10*. The distance of the zipper vertex from the final equilibrium position is plotted as a function of time in *Figure 10A,B* (the time point when equilibrium is reached is defined as *t*=0). It can be seen that in both advancing (*Figure 10A*) and receding (*Figure 10B*) zippers, the zipper vertex moves with a velocity in the range $(0.3 - 2) \frac{\mu m}{min}$. *Figure 10C* shows that while some zippers (R3 and A6) converge with an approximately constant velocity, others (R4 and A5) have a weakly exponential velocity profile, with the velocity gradually decreasing as equilibrium is approached. The former case, in which the zipper stops rather abruptly near the equilibrium position, is observed in roughly $2/3$ of the evaluated examples. In *Figure 10D*, the smoothed zipper angle is plotted as a function of time for three advancing and two receding zippers. In these examples, the angle increases with time for advancing zippers (A1, A4,

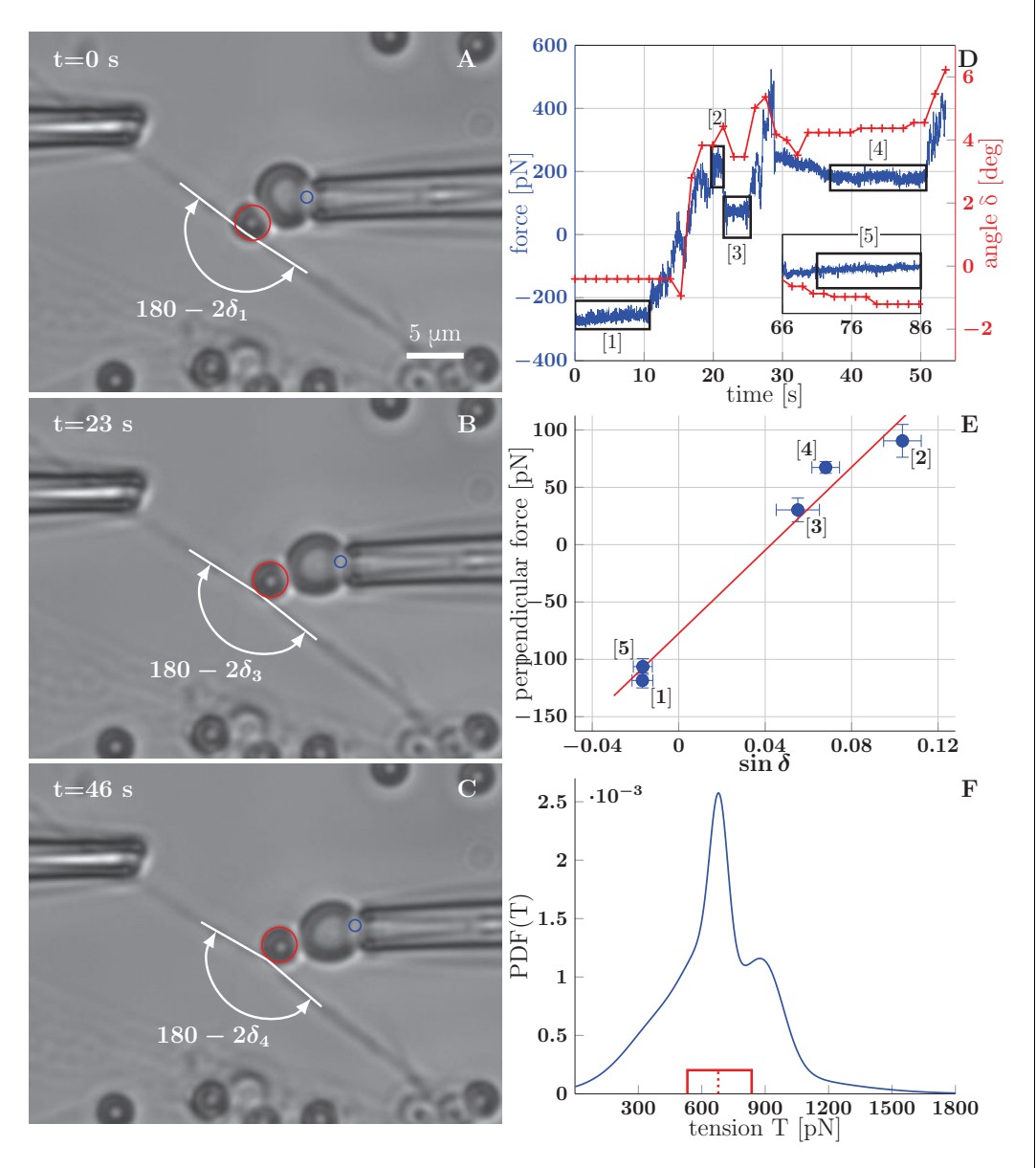

**Figure 8.** BFP measurements of axon tension. (A–C) illustrate a BFP experiment (the full recording is in the video *Figure 8—source data 1*) . (A) The bead is slightly pushed against the axon (with deflection angle $\delta<0$), negative deformation (compression) of the RBC is recorded. (B,C) Different stages of the probe exerting a pulling force on the axon; the RBC undergoes positive deformation (extension), the axon deflection angle $\delta>0$. Index $i$ of $\delta_i$ corresponds to the numbering of plateaux in panel D and data points in panel E. The tracked point on pipette and the tracked bead are marked by blue and red circles. (D) Time dependence of the force measured on the probe (evaluated for each frame at 65 fps), and the angle (evaluated each second). The deflection angle $\delta<0$ corresponds to deflection by pushing, $\delta>0$ means deflection by pulling. The deflection angle determines the lateral projection of axial tension acting at the apex, that is lateral tensile force $2T\sin\delta$. (E) Blue data points represent time-averaged qualities of individual plateaux (labelled by appropriate numbers), abscissa corresponds to average deformation $\sin\delta$ and ordinate to average perpendicular probe force $F_{\mathrm{BFP}}^{\perp}$. The error bars represent a standard deviation of the quantities during each plateau. The red line is a linear fit of BFP data points, i.e. $F_{\mathrm{BFP}}^{\perp}$ vs. $\sin\delta$, the slope corresponds to axon tension $2T$. Goodness of the fit is $R^2=0.97$. (F) Distribution of axon tensions, calculated as a normalized sum of linear fit results from all BFP experiments—each fit $j$ was represented by a Normal distribution, with mean at $\bar{T}_j$ given by the fit slope, and standard deviation $\sigma(T_j)$, given by the standard deviation of the fit. The tension mode value is 678 pN, mean 679 pN (designated by the dashed red line), interquartile range (529–833) pN (delimited by the red box). The distribution of tension in based on $N=7$ measurements, containing at least three force plateaux each, originating from four distinct cultures (each obtained from a different mother animal). The time course of force and angle (D), plateaux points and the fit (E), mean values of tension of all experiments and the distribution values (F) are available in the *Figure 8—source data 2*.

*Figure 8 continued on next page*

*Figure 8 continued*

The following source data and figure supplements are available for figure 8:

**Source data 1.** Illustration of a BFP experiment with overlays that mark the results of pipette and bead tracking (Video).
**Source data 2.** Time course of force and angle (D), plateaux averages and fit parameters (E), estimated tension distribution (F) and underlying experimental tension data.
**Figure supplement 1.** Tension increase during a BFP experiment with axon stretch.
**Figure supplement 2.** Estimation of the axon-axon adhesion parameter S.
**Figure supplement 2—source data 1.** Data plotted in *Figure 8—figure supplement 2A–C*.

**Figure supplement 2—source data 2.** Source code to process input data from *Figure 8—figure supplement 2—source data 1*.

A6) and decreases with time for receding zippers (R4, R5). In some other cases (typically those in which the zipper configuration was complex, e.g. influenced by side processes), the time dependence of the zipper angle was more irregular. The full dynamics of the zippers R4 and R5 is shown in the videos *Figure 10—source data 1* and *Figure 10—source data 2*.

## Dynamical biophysical model of zippering driven by imbalance of tension and adhesion forces

Our analysis of equilibrium zipper configurations (cf. *Equation 1*) was based on viewing the zippers as arising from the interplay of mechanical tension and inter-axon adhesion forces. To assess if the observed zipper dynamics is consistent with this framework, we developed a basic biophysical dynamical model, formulated as an effective equation of motion for the zipper vertex (see Materials and methods for the underlying assumptions and a full derivation). Consider the instantaneous configuration shown in *Figure 11A*; here, the axons are fixed at the points A,B,C (these may correspond to entangled connections with the rest of the network, to immobile adhesion points with the substrate, or to the soma or the growth cone), while the zipper vertex $V(x, y)$ is mobile. The condition for static equilibrium of the vertex (given by *Equation 1* in the case of a symmetric zipper and by *Equations 9,10* in the general asymmetric case) takes into account the mechanical tension in the axons and the force arising from axon-axon adhesion. When the vertex is moving, however, additional forces arise from energy dissipation. As shown below, including these frictional forces in the force balance condition permits to obtain an equation of motion, specifying the velocity of the vertex.

We first describe the frictional force arising from the stretching or shortening of axons (which necessarily occurs during zippering or unzippering). Within the linear viscoelasticity framework, the viscous stress in each axon is proportional to the local strain rate. Assuming a uniform elongation strain in between the axon fixed points, the strain rate is simply expressed as $\frac{\dot{L}}{L}$, where $L$ is the total length of the segments of the axon. During axon elongation or shortening, the total force acting in a cross-section of the axon is therefore

$$\tau = T + \eta^{\Updownarrow} \frac{\dot{L}}{L} \tag{2}$$

where $T$ denotes, as before, the axon tension, and $\eta^{\Updownarrow}$ is the elongation viscosity constant.

In addition to axon elongation/shortening, another possible source of energy dissipation consists in changes in the axon configuration in the immediate vicinity of the vertex. When the vertex advances during zippering, new regions of the axons undergo bending/unbending (internal structural changes), with corresponding viscoelastic losses. Possible non-equilibrium binding effects at the newly adhering membrane region may also result in dissipation. These energy losses are expected to result in a localized frictional force that acts at the vertex and is anti-parallel to the vertex velocity

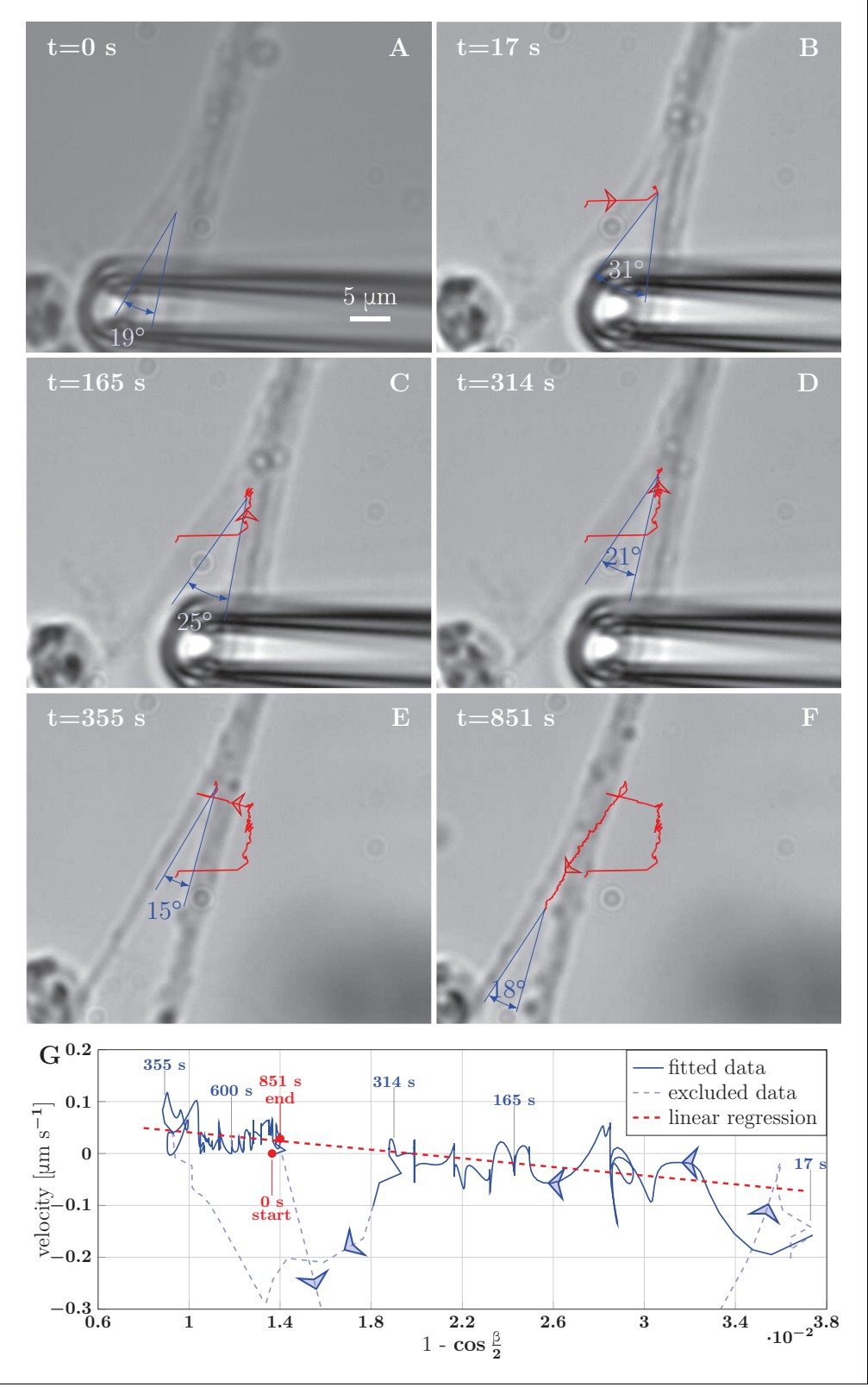

**Figure 9.** Example of induced unzippering and rezippering. (**A**) Initial state of the zipper before the manipulation. (**B**) Angle increases as vertex shifts to the side with the initial pipette displacement. (**C–D**) The axons unzipper, slowly. (**E**) Pipette is removed and axon released, the vertex shifts strongly to the left to equilibrate the lateral force imbalance. (**F**) Axons zipper back toward the initial configuration. (**G**) Blue line (with time stamps) shows the velocity and angle of the vertex during the manipulation. Data points belonging to the full line were fitted using linear regression (dashed red

*Figure 9 continued on next page*

*Figure 9 continued*

line). The goodness of the fit is $R^2$=0.48. The pale-blue dashed line corresponds to transients arising during manipulation (excluded from the regression). The values of angle were smoothed by a 20-frame Gaussian filter, and the velocity was calculated using convolution of positional data with derivative of the same Gaussian filter. The blue arrows show the direction of increasing time. Velocity, angle data and fit of panel G are available in *Figure 9—source data 2*.
The following source data is available for figure 9:

**Source data 1.** Induced unzipping experiment video corresponding to *Figure 9*.
**Source data 2.** Velocity and angle data, fit parameters (**G**).

component along the axis of the zipper; that is, this frictional force is collinear with the adhesion force $-S\,\widehat{VC}$. The magnitude of the combined zipper adhesion/friction force is

$$\chi = S - \eta^Z \vec{u} \cdot \left(-\widehat{VC}\right) = S - \eta^Z u^Z \tag{3}$$

where $\eta^Z$ is a friction constant and $u^Z$ is the 'zippering velocity', given by the projection of vertex velocity in the direction of advancing zipper (see *Figure 11A*). Thus, the friction force acts in the direction of the adhesion force during unzipping and in the opposite direction during zippering.

The balance of forces at a moving vertex may now be readily expressed. Consider for simplicity the case of a symmetric zipper (the asymmetric case is treated in Materials and methods and the Appendix). The dynamics preserves the symmetry, that is, an initially symmetric configuration ($T_1=T_2=T, \alpha_1=\alpha_2=\frac{\beta}{2}, L_1=L_2=L$) will remain symmetric during the course of zippering. Aligning the zippering direction (i.e. the direction of the zippered segment) with the $y$ axis, we have $u^Z = \dot{y}$ in *Equation 3* and $\dot{L} = (1 - \cos\frac{\beta}{2})\dot{y}$ in *Equation 2*. Replacing now, in the equilibrium equation *Equation 1*, $T$ by $\tau$ (*Equation 2*) and $S$ by $\chi$ (*Equation 3*), we obtain the condition expressing the total force balance in a moving vertex. Rearranging to express the zippering velocity $\dot{y}$, we get the equation of motion for a symmetric zipper

$$\dot{y} = \frac{S - 2T\left(1 - \cos\frac{\beta}{2}\right)}{2\eta^{\updownarrow}\frac{(\cos\frac{\beta}{2}-1)^2}{L} + \eta^Z}. \tag{4}$$

The terms $\cos\beta/2$ and $L$ on the right hand side are nonlinear functions of $y$ and are straightforwardly expressed in terms of the coordinates of the fixed points A, B, C. The resulting differential equation (*Equation 4*) cannot be solved in closed analytical form, but the predicted vertex trajectory $y(t)$ can be obtained by numerical integration.

We tested the equation of motion *Equation 4* by comparing it with the experimental recordings of induced zippering/unzippering dynamics in our system. We measured the zippering velocity $\dot{y}$ and the zipper angle $\beta$ during the experiment shown in *Figure 9A–F*; these quantities were evaluated at 1 s intervals and smoothed using a Gaussian kernel of half-width 10 s. *Figure 9G* demonstrates that the zipper velocity is linearly related to $1 - \cos\frac{\beta}{2}$. In this plot, the fast transients resulting from the axon manipulation are shown as pale blue dashed curves, while the zippering/ unzippering dynamics induced by the manipulation (once the axons relaxed into an approximately symmetric configuration) is shown as solid curves. The straight red line indicates the best linear fit (from which the fast transient manipulation segments were excluded).

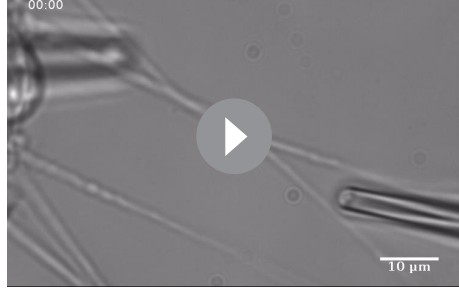

**Video 2.** Induced unzipping of a zipper segment delimited by two vertices on either side. In this case the unzipping becomes complete and the two constitutive bundles separate.

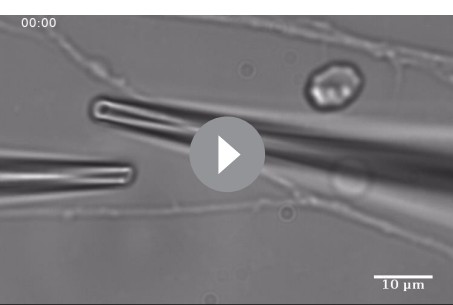

**Video 3.** Induced unzippering experiment. In this case the vertex does not recede, despite the large increase in zipper angle resulting from the manipulation by micropipette.

Comparing now to *Equation 4*, we see that such linear dependence is predicted when the friction in the vertex (i.e. term proportional to $\eta^Z$) dominates over the elongation friction ($\eta^Z u^Z \gg \eta^{\updownarrow} \frac{\dot{L}}{L}$). The slope of the linear fit is predicted to equal $-2T/\eta^Z$, while the predicted intercept is $S/\eta^Z$. From the ratio of the intercept ($0.0825 \frac{\mu m}{s}$) and the slope ($-4.1692 \frac{\mu m}{s}$) in *Figure 9G*, we therefore obtain an estimate of the ratio of the axon adhesion force $S$ to the axon tension $T$. This dynamical estimate gives $\frac{S}{T} = 0.04$, as compared with the typical value $\frac{S}{T} = \frac{162\text{pN}}{679\text{pN}} = 0.24$ that we obtained from the analysis of static configurations. The induced zippering experiment of *Figure 9* was performed with axon bundles located close to the explant boundary; these have larger total tension $T$ than the single axons forming the zippers used in our static analysis, while the adhesion parameter $S$ is expected to scale sub-linearly with the number of axons in the bundle; this may explain the lower dynamic $S/T$ ratio. Assuming a tension of order 2 nN, the slope of the fit indicates a value for the vertex friction constant of order $\eta^Z \sim 10^{-3} \frac{Ns}{m}$.

We now discuss the zippering dynamics in the case of an asymmetric zipper. The general equation of motion for the vertex is presented in the Appendix, and includes (in addition to the elongation and zippering friction introduced above) the friction of the axons with the substrate. *Figure 12* shows representative trajectories of the vertex obtained by numerical integration of the general equation of motion. The panel 12A displays a contour plot of the energy landscape $E(x, y)$, defined as the total tensile and adhesive energy of the zipper configuration with vertex located at $(x, y)$; this energy is given by *Equation 7* (in Materials and methods). The energy landscape plotted in *Figure 12A* corresponds to a zipper constituted by axons with tensions $T_1$=1 nN and $T_2$=1.5 nN and mutual adhesion strength $S$=0.2 nN. The marked 'final point' denotes the static equilibrium point of the landscape. The initial point of the trajectories in *Figure 12A* corresponds to the equilibrium zipper configuration for $T_1=T_2$=1 nN. Following a rapid increase (between time $t$=0 sand $t$=5 s) of the tension in the right axon by 0.5 nN, the zipper undergoes relaxation to the new equilibrium, driven by the force given by the gradient of the energy landscape displayed in *Figure 12A*. It is seen that different forms of dominating friction (black for viscous elongation, red for substrate friction, blue for vertex-localized friction) lead to distinct paths (*Figure 12A*) as well as time courses (*Figure 12B, D*) of the trajectory. For comparison, the red dashed curve in *Figure 12A* shows the gradient path, which would correspond to an isotropic and geometry-independent vertex friction tensor $\overset{\leftrightarrow}{H}$ (see Materials and methods).

Our experimental observations (as in *Figure 9*) show that a typical response of a zipper to a fast asymmetric perturbation consists of a fast lateral equilibration, followed by a slower dynamics during which the vertex moves parallely to the zippered segment. Such trajectory arises from our model in case of dominant zipper friction (blue line), while it cannot be achieved through the other friction mechanisms alone. We conclude that the velocity of zippering is primarily limited by the internal friction localized at the zipper vertex.

Our observations of spontaneous zippering processes in the developing network showed that the zippering velocity typically remained approximately constant (for $\sim 2/3$ of the events), with abrupt stop near the equilibrium point (see *Figure 10*). The velocity profiles obtained from the model in case of zippering resulting from abrupt perturbation, in contrast, are exponential or double-exponential (*Figure 12D*). An approximately constant velocity of zippering is obtained in the model, however, when the tension is assumed to increase gradually over an extended interval of minutes, rather then abruptly (*Figure 12E*). The corresponding trajectories are shown in *Figure 12C*. In this case, the paths obtained for different dominating forms of friction are similar to each other. This is a consequence of the gradual increase in tension: for all of the friction types considered, the relaxation dynamics is then sufficiently fast to allow the zipper vertex to closely track the equilibrium point of

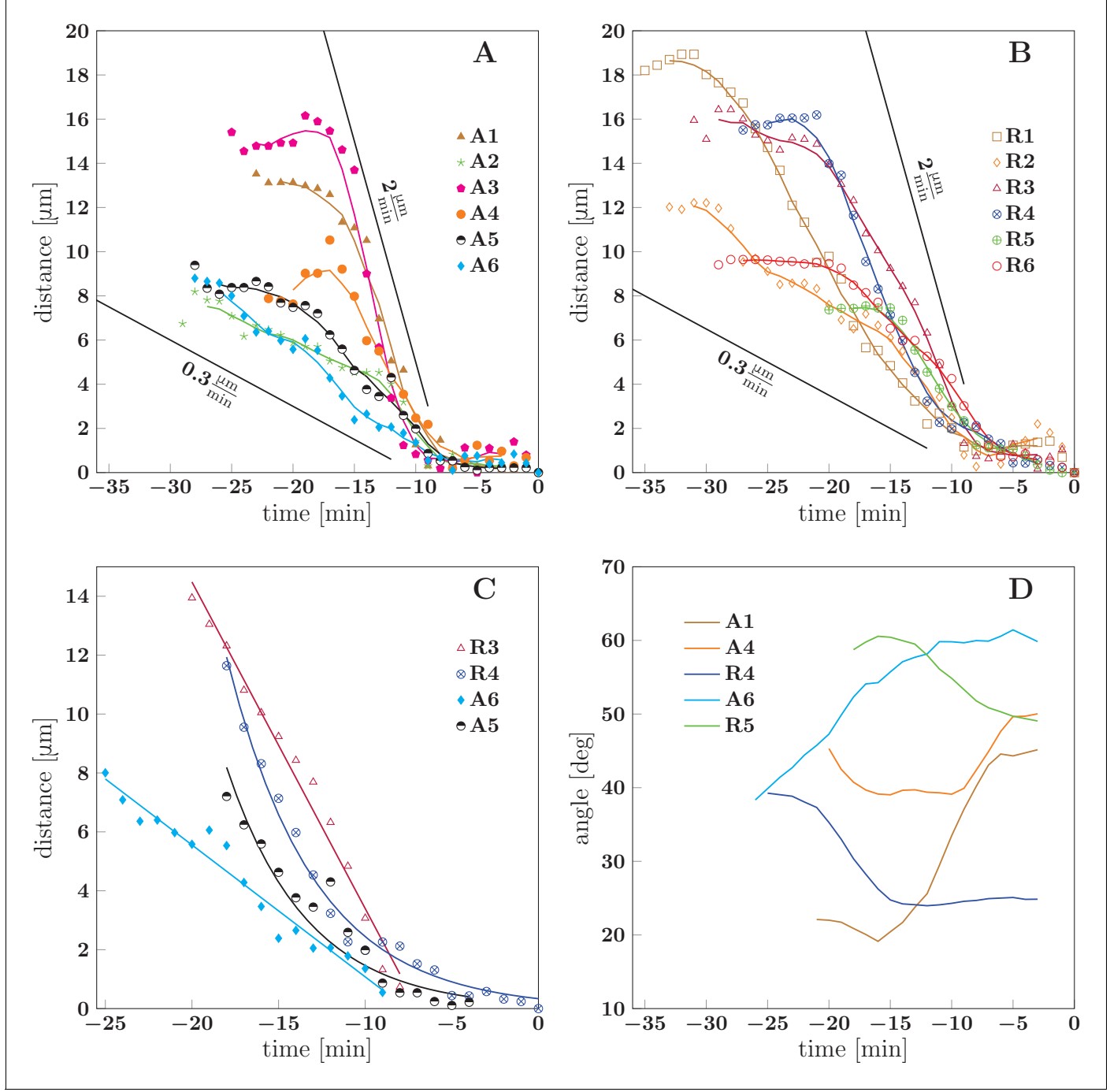

**Figure 10.** Dynamics of spontaneous zippering events in the evolving network. (A) and (B) show the convergence to equilibrium for selected advancing and receding zippers, respectively. The distance between the zipper vertex location at the given time and the final equilibrium position is given. The lines with slopes $(0.3 - 2.0)\frac{\mu m}{\min}$ delimit the typical zippering and unzippering velocities. (C) Fits illustrating approximately linear or exponential convergence in time. Linear fit equations: $d_{R3}(t) = -1.11(t + 6.94)$ and $d_{A6}(t) = -0.45(t + 7.59)$; exponential fit equations: $d_{R4}(t) = 14.95 \exp\{-0.20(t + 19.14)\}$ and $d_{A5}(t) = 10.42 \exp\{-0.22(t + 19.12)\}$. (D) Time course of zipper angles, smoothed by five-frame window. Note that the angle increases for advancing zippers and decreases for receding zippers. Data plotted in panels A, B and D are available in **Figure 10—source data 3**.

The following source data is available for figure 10:

**Source data 1.** Video of receding zipper R4 from **Figure 10**.

**Source data 2.** Video of receding zipper R5 from **Figure 10**.

*Figure 10 continued on next page*

*Figure 10 continued*

**Source data 3.** Plot data (**A**, **B** and **D**).

the energy landscape, which evolves on the time scale of minutes. These results suggest that in the developing network, the zippering is driven by gradual, rather than abrupt, changes in the forces that act at the zipper vertex. The resulting reconfiguration of the zipper may then act as a gradual perturbation acting on the zippers in the immediate vicinity.

To summarize, the comparison of predictions of the dynamical model with experimental observations supports a framework in which the zippering arises from an imbalance of tension and adhesion forces at the zipper vertex, and in which the zippering velocity is limited predominantly by friction arising from internal energy dissipation in the immediate vicinity of the moving vertex.

## Topological changes and loop stability in the evolving axon network

Following the analysis of the statics and dynamics of individual zippers, we establish a connection to the global dynamics of developing axon network.

The gradual decrease of the total network length with time (*Figure 1H*) indicates that in our experimental setting, zippering is overall more frequent than unzippering. The observed decrease of the total number of vertices (*Figure 1H*) is likewise a natural consequence of zippering. An advancing zipper vertex may eventually encounter another vertex and combine with it, resulting in a zipper consisting of thicker fascicles.

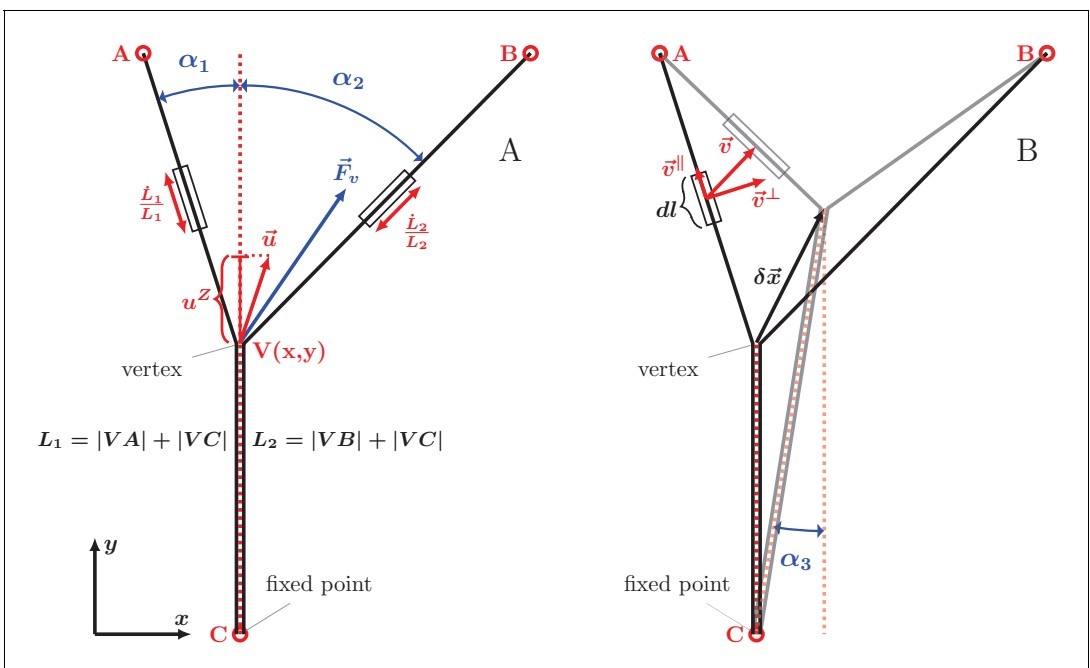

**Figure 11.** Geometry and notation for the dynamical model of axon zippering. (**A**) Illustration for the zippering dynamics model. $L_1$ and $L_2$ denote the lengths of the two axons. The red dotted line represents the adhered zipper segment and its extension beyond the vertex (i.e. zipper axis), aligned with the $y$-axis in the figure. The blue vector $\vec{F}_v$ represents the conservative forces (i.e. tension and adhesion) and the red vector $\vec{u}$ the resulting vertex velocity limited by friction. Projection of the velocity to the zipper axis, $u^Z$, determines the vertex-localized dissipative force, as $f^Z = -\eta^Z u^Z$. The strain rate, $\frac{\dot{L}}{L}$, determines the elongational viscous dissipative force, that is $f^{\updownarrow} = -\eta^{\updownarrow}\frac{\dot{L}}{L}$. (**B**) Illustration for the Appendix. $\delta\vec{x}$ represents a small displacement of the vertex. Vector $\vec{v}$ is the velocity of the element $dl$, in contrast to the velocity of the vertex, $\vec{u}$, in panel A. Axial and transverse substrate friction forces $\vec{f}^{\parallel}$ and $\vec{f}^{\perp}$ are proportional to the element velocity components $\vec{v}^{\parallel}$ and $\vec{v}^{\perp}$.

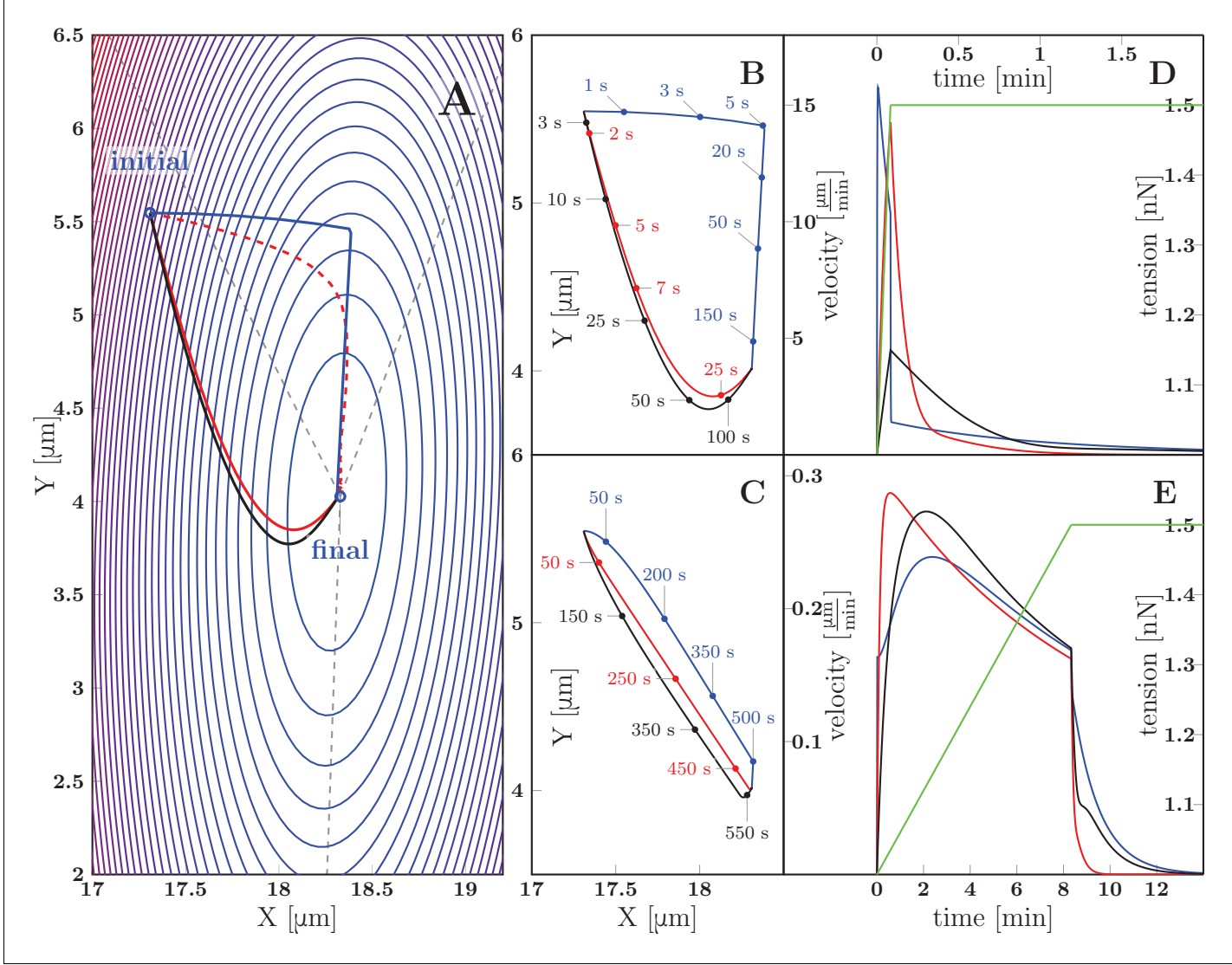

**Figure 12.** Predicted zippering dynamics resulting from applying a perturbation to a zipper initially in equilibrium, converging to a new equilibrium. (A) The landscape of tensile and adhesive energy (*Equation 7*) for the new equilibrium condition (specified by the parameter values: left tension $T_1=1\frac{nJ}{m}$, right tension $T_2=1.5\frac{nJ}{m}$ (up from $T_2=1\frac{nJ}{m}$ in the initial equilibrium), axon-axon adhesion $S=0.2\frac{nJ}{m}$). Blue contours indicate locations of equal energy. Gray dashed lines show axons in the final equilibrium, dashed red line is the gradient trajectory between equilibria,. The full lines indicate zipper vertex trajectories following a rapid increase of the right tension. Red: trajectory with dominant substrate friction ($f^{\parallel}+f^{\perp}$), blue: trajectory with dominant zippering friction ($f^Z$), black: trajectory with dominant elongation friction ($f^{\updownarrow}$). The trajectories with dominant friction types are represented by the same color code across all panels. (B) The same trajectories as in A, with time stamps. The tension in the right axon increased rapidly over 5 s and then was kept constant (see green line in panel D). (C) Trajectories during gradual perturbation, with dominant zippering friction (blue), substrate friction (red) and elongation friction (black) over 1000 s. The tension was gradually growing in the right axon over 500 s and then was kept constant (see green line in panel E). (D, E) velocity of vertex during transition, color code corresponds to panels B and C, green line represents the prescribed tensile force in the right axon during the transition. For each model run, one friction constant was set to a particular value to probe its effect on the trajectory, others were set to zero. The following values were used: axial substrate friction $\eta^{\parallel}=200$ Pa s, transverse substrate friction $\eta^{\perp}=200$ Pa s, elongation friction $\eta^{\updownarrow}=3000$ nN s, zippering friction $\eta^Z=\frac{nNs}{\mu m}$ (in this case, a small substrate friction value $\eta^{\parallel}=\eta^{\perp}=1$ Pa s was introduced to avoid a singularity when the motion direction was perpendicular to the zippering axis). Source code of the implemented zipper model used to generate the data is available in *Figure 12—source data 1*.

The following source data is available for figure 12:

**Source data 1.** Source code of zipper dynamical model to generate plot data (A–E).

A process of this type repeatedly observed in the developing network (*Figure 3G–J*) consisted of a gradual collapse of triangular loops, with the three vertices eventually converging into a single-vertex quasi-stable configuration. A possible underlying zipper structure is illustrated in *Figure 13A*. During this process, the loops typically retained their shape, that is the three zipper angles remained approximately constant during the collapse. Such dynamics is expected to result from a decrease in the tension of the axons that constitute the loop, such that the equilibrium zipper angle becomes larger than the current zipper angles. In such case, no stable redistribution of angles is possible and the vertices advance synchronously, keeping the loop shape invariant. This combined dynamics is therefore distinct from the elementary zippering process we considered in *Figure 11*, where it was assumed that the fixed points A, B, C were immobile, and consequently the zipper angle gradually increased as the zipper approached equilibrium. A strong support for this interpretation of the mechanism of loop collapse is provided by the experiments in which we used FBS to generate a pull on the network, hence increasing axon tension. As seen in *Figure 5F–H*, this manipulation leads to the rapid opening and expansion of triangular loops in the de-coarsening areas of the network.

Our induced zippering experiments and model analysis showed that the zippering transients resulting from sudden perturbations last for minutes, while the coarsening of the network develops over hours. Such separation of time scales indicates that the network is locally near the quasi-equilibrium state corresponding to the momentaneous values of the axon tensions. The network statistics reported in *Figure 1H* exhibit robustly monotonous time course and low volatility, which is consistent with this assumption and shows that large abrupt perturbations do not dominate the network dynamics. At a given time, the majority of vertices in the network are seen to be approximately static or fluctuating around an equilibrium position, while the proportion of steadily advancing or receding zippers is minor (see *Video 1*). In the following analysis, we will assume that the majority of the zippers have a zipper angle that is close to the equilibrium value given by *Equation 1* (see also Discussion).

## Progressive fasciculation is reflected in the network distribution of zipper angles

The observed decrease in total network length implies that larger fascicles are gradually formed. The limits of optical microscopy resolution did not allow us to reliably determine the size of fascicles forming individual zippers. However, the structure of the fascicles determines their tension and is therefore expected to be reflected in the equilibrium zipper angles (cf. *Equation 1*). To examine this relation, we extracted the distribution of the zipper angles in the network. At each analyzed time point, the network was manually segmented as in *Figure 1D–G* and the angles between the graph edges were measured. At each zipper vertex, the zippering angle was selected as the sharpest of the three angles between the edges, unless the observed configuration indicated otherwise. Crossings (marked by green stars in *Figure 1D–G*) were excluded from the statistics.

The analysis included a total of five experiments in which the network coarsened (each lasting for 178 to 295 min), with 7–10 time points per experiment at which the zipper angle distribution was extracted. The typical shape of the distribution is shown in *Figure 14A*. Note the marked underrepresentation of sharp zippering angles (below 20°). In the example of *Figure 14A* (which corresponds to *Figure 1D-H*), the median angle of the distribution shifted to lower values during the 3 hr interval (from 60° to 49°).

Evaluating the relation between the zipper angle distribution and the network coarsening, we found a consistent trend in the five analyzed experiments. The median zipper angle $\beta_M$ overall showed a positive correlation with the total network length $L$ (with the five correlation coefficients in the range (0.26–0.69)). Two examples are shown in the scatter plots of *Figure 14B*, where *experiment 1* corresponds to the time interval in *Figure 1H*.

To propose an explanation of this observation, we return to the distribution of single-axon tensions obtained using the BFP technique (*Figure 8F*). We assume that the distribution of tensions of zipper-forming axons (either single or fasciculated) matches the distribution from the BFP experiments. Treating the tensions of individual axons in a fascicle of size $n$ as independent random variables, it follows that the mean of the fascicle tension distribution scales as $\bar{T} \sim n$ and its standard deviation as $\sigma(T) \sim \sqrt{n}$.

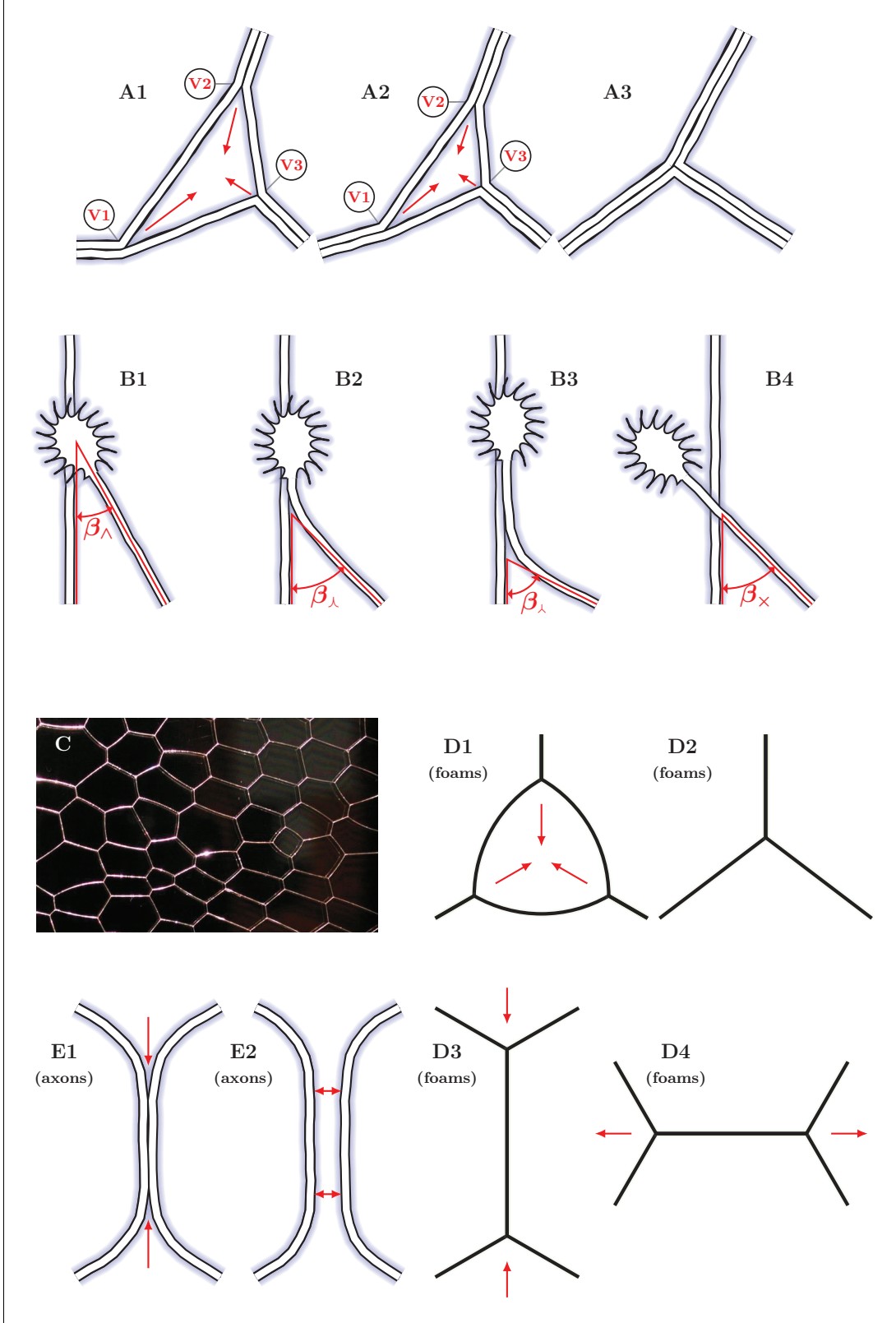

**Figure 13.** Elementary dynamical processes in fasciculating axon networks and in coarsening liquid foams (froths). (**A1–A3**) Two vertices are lost during the process of closing of a loop formed by three axon segments. The initial configuration starts to zipper at one or more vertices, gradually decreasing the total network length. A single junction formed by three pairs of fully zippered axons remains. (**B1–B4**) Possible outcomes of initial contact of two axons. (**B1**) Growth cone (GC) interacts with the shaft of another axon, (**B2**) small initial incidence angle allows incoming GC to adhere and follow the

*Figure 13 continued on next page*

*Figure 13 continued*

shaft, (**B3**) the two shafts zipper, increasing the contact angle, (**B4**) if the initial incidence angle exceeds the equilibrium zipper angle, no stable zippered segment can be formed and the growth cone crosses over. (**C**) Photograph of structure formed by a liquid foam restricted between two glass plates. The gas bubbles are separated by liquid walls that meet at triple junctions. (**D1–D4**) Schemes illustrating the elementary topological processes in liquid foams. (**D1**) A three-sided bubble with curved walls, containing gas under excess pressure. (**D1–D2**) The gas diffuses to neighboring cells and the three-sided bubble gradually collapses. This process is called *T2*. (**D3–D4**) In foams, the *T1* process leads to a reconnection of bubble walls, preserving the number of vertices of the network. (**E1–E2**) In the axon network, separation rather than reconnection results from unzippering. (**E1**) Two vertices delimiting a zippered segment start to recede, (**E2**) Once the adhered segment length decreases to zero, the two axons detach and separate (see experimental example in *Video 2*). Panel C is adapted from https://commons.wikimedia.org/wiki/Category:Foam\#/media/File:2-dimensional_foam.jpg: Foam\#/media/File:2-dimensional_foam.jpg by Klaus-Dieter Keller, released into the public domain by the author.

To evaluate how this is reflected in the distribution of zippering angles, we use *Equation 1* with an appropriately rescaled adhesion strength $S$. The adhesion force between two fascicles scales with their contact area and therefore with the fascicle surface. For a fascicle composed of $n$ axons, the surface is expected to scale as $\sim \sqrt{n}$ (assuming that the cross-section of the fascicle remains approximately circular, rather than flattened by strong adhesion to the substrate, which is supported by the SEM micrographs presented in *Figure 4*). Using these scaling rules and *Equation 6* (which follows from *Equation 1*, see Materials and methods), we can transform the distribution of tensions $p(T)$ into the distribution of zippering angles $q(\beta)$.

To qualitatively asses the changes of $q(\beta)$ with fascicle growth, we made two simplifications: (i) we replaced the experimental distribution of tensions with a lognormal distribution, $p(T) = \mathrm{PDF}_{\log}(\bar{T}, \sigma(T))$, of the same mean $\bar{T} = \bar{T}_{\mathrm{BFP}} = 0.68\mathrm{nN}$ and std $\sigma(T) = \sigma_{\mathrm{BFP}} = 0.25\mathrm{nN}$ and (ii) we used a single value of $S$ (appropriately scaled with $n$), ignoring its possible variance. We verified numerically that the lognormal approximation of tension distribution for fascicles of size $n \geq 2$ closely corresponds to the tension distribution obtained by $n$-fold convolution of the single-axon distribution. The distribution of zipper angles is then given by *Equation 6*, using the lognormal distribution of tensions with the two parameters related through scaling with $n$ as $\bar{T} = n\bar{T}_{\mathrm{BFP}}$, $\sigma(T) = \sqrt{n}\sigma_{\mathrm{BFP}}$, and using the scaled value of adhesive strength $S = \sqrt{n} \cdot 0.17\mathrm{nN}$. As shown in *Figure 14C*, this analysis predicts that a coarsening-induced increase in mean fascicle size $n$ leads to a lower median zipper angle, in agreement with the trend seen in the experimental data. The orange curve in *Figure 14C* is the angle distribution with parameters set to the BFP-derived values, while the green curve is the predicted distribution after rescaling of fascicle size $n$ by factor 1.50. This factor was obtained from the data in *Figure 1H* and from the expected scaling $n \sim 1/D$, where $D$ is the total network length per unit area.

## The structure of a sensory neurite plexus in *Xenopus* embryo is consistent with the dynamical zippering framework

Strong connection points can be established between our dynamical observations and the in vivo observations of Roberts and Taylor (*Roberts and Taylor, 1982*), who studied the formation of the sensory neurite plexus on the basal lamina of trunk skin in *Xenopus* embryos. In *Roberts and Taylor (1982)*, the neurite network on the trunk and the inside skin surface was examined using electron microscopy at magnification 1000–2500, and the angles between neurites that fasciculated or crossed ('incidence angles') were determined. As shown in the inset of *Figure 15*, the distribution of the fasciculation incidence angles in (*Roberts and Taylor, 1982*) is similar to the distribution of zipper angles measured in our system. Small angles (between 0° and 30°) are notably absent from the recorded angle distributions (see also *Figure 14A*), while these angles would be a priori expected to be equally represented in an isotropically growing network (and overrepresented in a network with a preferred direction of growth). Roberts and Taylor proposed that this was a result of zippering processes analogous to the ones that we directly observed in our study. Thus, if a growing axon encounters another axon at an initially small incidence angle and starts following it (*Figure 13B1,B2*), the segment behind the growth cone subsequently zippers and the incidence angle increases until the equilibrium zipper angle is reached (*Figure 13B3*). Our observations of zippering dynamics are consistent with this proposal. The under-representation of small angles in the zipper angle distributions

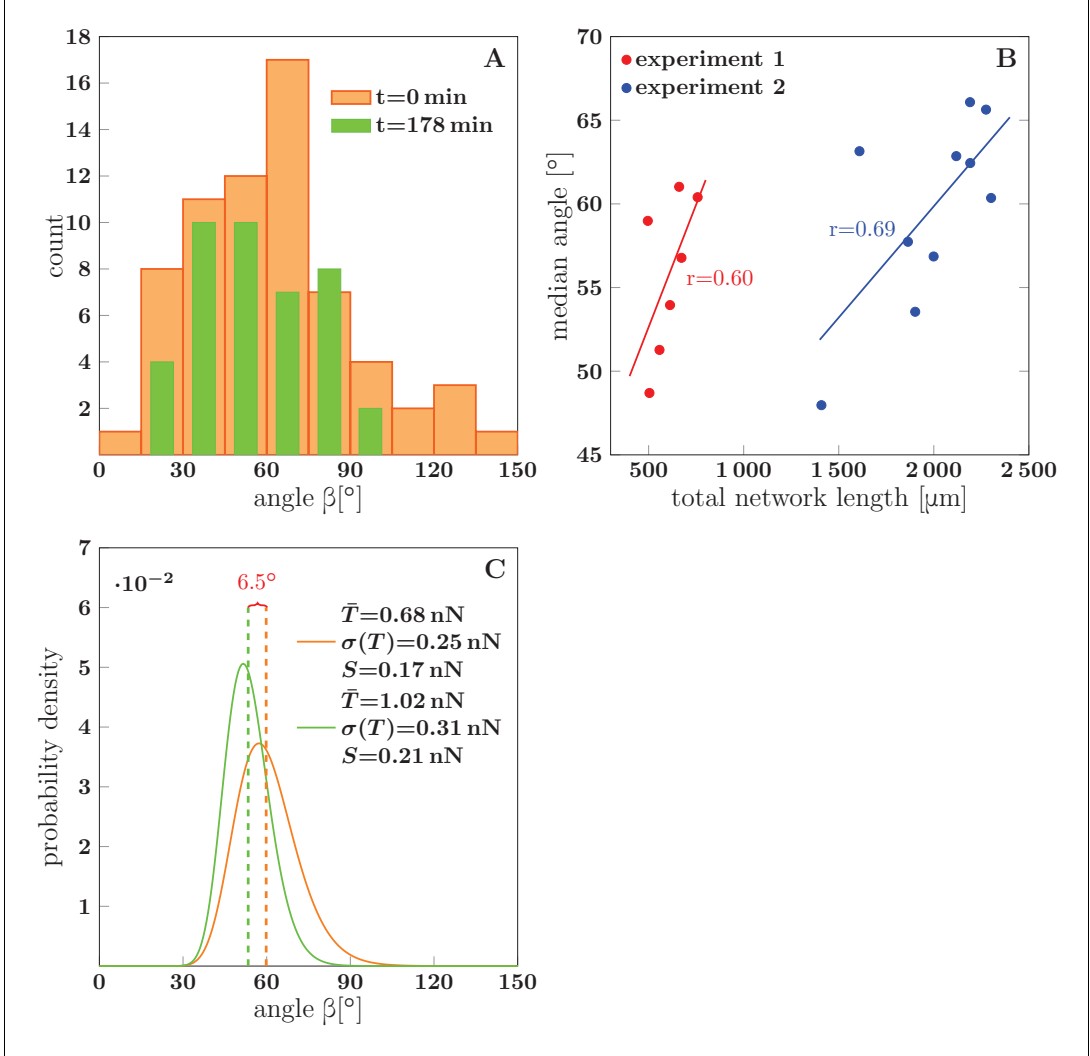

**Figure 14.** Evolution of the network distribution of zipper angles. (A) The distribution of zipper angles in the network configurations of *Figure 1D* (*t*=0 min, total 66 vertices) and *Figure 1G* (*t*=178 min, total 44 vertices). (B) Correlation between median angle $\beta_M$ and the total network length $L$ in two experiments; *r* denotes the Pearson correlation coefficient. (C) Predicted equilibrium zipper angle distribution PDF($\beta$) obtained as a transformation of distribution of fascicle tension PDF($T$) using *Equation 6* (see main text and Materials and methods). The distribution of tensions was approximated by a lognormal distribution $\text{PDF}_{\log}(\bar{T}, \sigma(T))$. The distribution plotted in orange corresponds to the values of tension from BFP experiments, $\bar{T}_{\text{BFP}}$=0.68 nN, $\sigma_{\text{BFP}}(T)$=0.25 nN, and adhesion parameter $S_0$=0.17 nN adjusted to match the initial median angle of *experiment 1*. The distribution plotted in green corresponds to parameters rescaled with mean fascicle size *n* as $\bar{T} \sim n$, $\sigma(T) \sim \sqrt{n}$, $S \sim \sqrt{n}$, with increase in mean fascicle size (1.50×) corresponding to *Figure 1D–G* (see main text). The change in median angle in panel C is 6.5°, as compared to 7.5° given by the trendline in panel B (*experiment 1*). The data of histograms (A), correlations (B) and distributions (C) are available in *Figure 14—source data 1*. The source code used to generate distributions in panel C is available in *Figure 14—source data 2*.

The following source data is available for figure 14:

**Source data 1.** Data of histograms (A), correlations (B) and distributions (C).
**Source data 2.** Source code to generate angle distributions (C) using *Equation 6*, see Materials and methods.

(*Figures 14A* and *15*) thus further supports our inference that most zippers are close to local equilibrium during the development of the network.

In addition to extracting the distribution of incidence angles for fasciculated neurites, Roberts et al. determined the probability for two neurites to cross (rather than fasciculate). This crossing probability $\Pi(\beta_{\text{inc}})$ was found to depend strongly on the incidence angle $\beta_{\text{inc}}$ (*Figure 15*). Using our

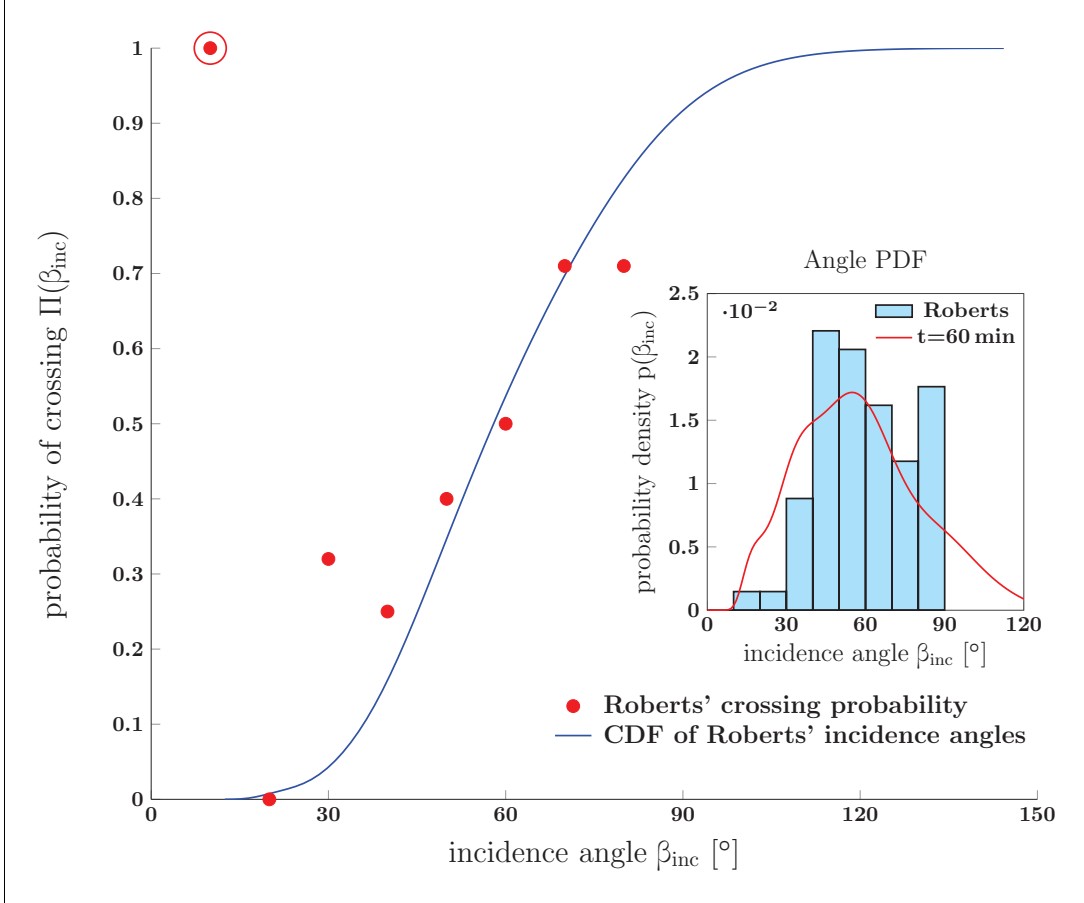

**Figure 15.** Prediction for the probability of crossing (rather than zippering) of two neurites, and its comparison to in vivo experimental data. The red dots show data from Fig. 14 of Ref. (**Roberts and Taylor, 1982**) (referred to as Roberts in the figure): the observed probability of crossing of two neurites as function of observed incidence angle $\beta_{inc}$. The blue line is the cumulative distribution function of angles of incidence obtained from Roberts' data. It was calculated as $\Pi(\beta_{inc}) = \int_0^{\beta_{inc}} p(\beta)d\beta$, where the PDF of angles of incidence, $p(\beta)$, was constructed by kernel-smoothing the Roberts' histogram. The crossing probability of interval $(0–10)°$ is an outlier (marked by the red ring) based on a single observed case. Inset: Histogram of angles of incidence as observed by Roberts (upper panel of Fig. 13 in **Roberts and Taylor (1982)**) . The red line is the PDF of vertex angles measured in our in vitro system (**Figure 1** at 60 min). The crossing probabilities and angles, experimental data and the distributions, are available in **Figure 15—source data 1**.

The following source data is available for figure 15:

**Source data 1.** Crossing probabilities and angles—data (**Roberts and Taylor, 1982**) and distributions estimates.

analysis framework, we can quantitatively explain this observed dependence. A given pair of axons will not fasciculate (zipper) if their equilibrium zippering angle $\beta_{eq}$ is smaller than their initial incidence angle $\beta_{inc}$. Any zippered segment formed in this situation would be unstable towards unzippering; the axons are therefore expected to cross while maintaining the initial incidence angle (**Figure 13B4**). Given that there is a distribution of equilibrium zippering angles in the network (see the previous section), the probability that two randomly chosen axons with initial incidence angle $\beta_{inc}$ will cross (rather than zipper) equals the probability of their equilibrium angle $\beta_{eq}$ being smaller than the incidence angle $\beta_{inc}$. This probability—the cumulative distribution function evaluated at $\beta_{inc}$—is computed in **Figure 15**, starting from the angle distribution taken from **Roberts and Taylor (1982)**. A good agreement with the crossing probabilities reported in **Roberts and Taylor (1982)** is seen. We thus successfully applied our framework to explain the network properties observed in the in vivo study of **Roberts and Taylor (1982)**, indicating that this framework is not limited to dynamics in culture.

## Discussion

Axon fasciculation is generally thought to be controlled during development at the level of growth cones, which may adhere to other axons in their environment (hence driving fasciculation), or may separate from other axons (hence driving defasciculation). Here, we provide strong evidence for an additional mechanism playing a critical role in regulating axon fasciculation, which does not involve growth cones, but takes place at the level of axon shafts, through zippering of individual axons or small bundles of axons. In our culture of embryonic olfactory epithelium explants, this mechanism resulted in a substantial reorganization of the structure of a grown axon network, on the time scale of 10 hr. The process of axon zippering has been rarely reported or discussed in previous literature. Axon zippering in vivo was inferred in *Roberts and Taylor (1982)* and was noticed for axons growing in culture in *Voyiadjis et al. (2011)* and *Barry et al. (2010)*. The process of zippering and the underlying biophysical mechanisms, however, were never studied. We thus decided to take advantage of our novel culture system, which presents the right balance of axon tension and axon-axon adhesion forces, to gain insight into this phenomenon. We undertook a detailed characterization of axon zippering and unzippering, in view of understanding its dynamics and possible biological significance.

### Axon zippering as the result of competition between mechanical tension and axon-axon adhesion

Using a combination of experimental observations and biophysical modeling, we showed that axon zippering arises from the competition of two principal forces: axon-axon adhesion and mechanical tension. The adhesion force favors an increase in the length of the zippered segment. The mechanical tension tends to minimize the total length of the axons, thus favoring unzippering. The relative strength of these two forces determines the vertex angle between axons in a zipper that reached static equilibrium. We used the BFP technique to measure the mechanical tensions of the OSN axons grown ex vivo, and obtained values (interquartile range (529–833) pN) comparable to tensions reported in the previous literature for PC-12 neurites grown in culture, on average around 650 pN (*Dennerll et al., 1988*). Combining this information with measurements of the geometry of zippers in static equilibrium, we extracted the magnitude of the axon-axon adhesion force, obtaining approximately $S \approx 100$ pN (with an upper bound on its spread, the interquartile range (52–186) pN). To our knowledge, this is the first experimental estimate of the force of adhesion between axon shafts, in any system. From the EM images in *Figure 4*, we estimate that the fraction of circumference participating in contact between two axon shafts is in the range (15–35)%. Assuming that 25% of the circumference adhered and converting the adhesion force $S$ to the adhesion energy per unit membrane area, one obtains $6 \times 10^{-16} \frac{\mathrm{J}}{\mu\mathrm{m}^2}$. This is comparable to the energy density for E-cadherin-mediated cell-cell adhesion, which we estimate from the separation force measurements of *Chu et al. (2004)* to be $(2 \times 10^{-16}$ to $4 \times 10^{-15}) \frac{\mathrm{J}}{\mu\mathrm{m}^2}$ (obtained as $\frac{F}{3\pi R}$, where $F$ is the separation force and $R$ is the cell radius).

We were able to unzip selected zippers by manipulating them with micropipettes, with consequent re-zippering after the manipulation was stopped. By fitting such induced zippering/unzippering dynamics to a basic biophysical model, we obtained an independent estimate of the axon-axon adhesion force, comparable with the estimate based on static observations. Comparing the shape of the observed zipper trajectories with the trajectories predicted by the dynamical model, we inferred that the zippering dynamics is limited by energy dissipation arising at the zipper vertex, and estimated the corresponding friction coefficient $\eta^Z$. Taking into account the typical zipper vertex velocity of $(0.3–2.0) \frac{\mu\mathrm{m}}{\mathrm{min}}$, the rate of energy dissipation during zippering or unzippering is of order $\sim 10^{-17} \frac{J}{\mathrm{min}}$. Our results provide a first systematic characterization of the statics and dynamics of individual axon zippers. The dynamical biophysical model that we developed (see Materials and methods) makes it possible to include axon zippering in mathematical models of axon guidance and bundling. Such studies previously focused on the dynamics of the growth cone, and modeled growth cone guidance by diffusible guidance cues (*Goodhill and Urbach, 1999*), the influence of tension forces and anchor points/focal adhesions on growth cone trajectory (*Li et al., 1995*; *Nguyen et al., 2016*), as well as contact interactions of growth cones with other axons (*Hentschel et al., 1999*;

*Chaudhuri et al., 2011*). The previous modeling studies did not, however, consider the dynamics of axon shafts.

In the previous literature, the tendency to fasciculate was often interpreted as arising from differential adhesion, that is, the growth cone having stronger adhesion to another axon than to the substrate (*Acheson et al., 1991*; *Roberts and Taylor, 1982*). The structure of the small fascicles observed in our system, with axons travelling on top of each other (*Figure 4*), does indicate that adhesion between axon shafts is stronger than axon-substrate adhesion. We note, however, that for the zippering of axon shafts to occur, such differential advantage is in principle not necessary. Two axon shafts (or axon fascicles) adhered to the substrate can gain mutual adhesion energy by initiating zippering, while remaining adhered to the substrate (as in the configurations shown in *Figure 4B–C*). In our biophysical analysis, we assumed that adhesion to the substrate was preserved during zippering, and we did not model the possible subsequent slower rearrangements (involving loss of substrate contact for some axons) in the internal structure of the zippered fascicle.

## Structure and dynamics of the axon network in light of the zippering framework

A striking observation, in our culture system, is that the axon network, initially established as a complex network of individual axons or small bundles of axons, progressively coarsens in time, leading to the formation of large fascicles of axons. The coarsening persists over time scales of about 10 hr, which is much longer than the typical time scale for an individual zippering process (10 min). It is therefore unlikely that the coarsening is a result of protracted equilibration of zipper configurations under stationary force conditions. A possible explanation of the slow coarsening dynamics may lie in a gradual decrease of the average axon tension as the culture matures, which would lead to increasing domination of axon-axon adhesion forces over tension, hence favoring zippering. This proposal is supported by the observed sequence of stages of the culture maturation: axon elongation in the early stage, insignificant elongation in the intermediate stage and axon retraction in the final stage. According to *Dennerll et al. (1989)*, axonal elongation is possible only when the axon tension exceeds a certain threshold (estimated as 1 nN for PC-12 axons); in our system, the arrest of growth in the intermediate stage may thus have resulted from a decrease of tension below such threshold. Further indications of decreased tension are observed near the end of the intermediate stage, when some growth cones visibly loose their grip to the substrate (see *Video 4*), and hence can no longer generate axon tension by pulling (*Lamoureux et al., 1989*).

To directly test if a decrease in average axon tension leads to coarsening, we used cytochalasin, a drug that was previously shown to significantly decrease the tension of PC-12 neurites (*Dennerll et al., 1988*). Indeed, we found that when the drug was applied to a slowly coarsening or stabilized network, a marked increase in coarsening rate resulted within 30 min of the application (*Figure 6*). Apart from the more pronounced coarsening, cytochalasin did not change the structure of the network, and the axons did not become visibly slack; this is consistent with the expected reduction of tension in all directions in the evolving network, with enough tension remaining to keep the axons taut. While we did not measure the axon tension in cytochalasin-treated cultures, we did observed morphological changes (the growth cone acquiring a stub-like shape with suppressed filopodia, and a reduction in number of axonal side processes) consistent with previous studies (*Dennerll et al., 1988*; *Letourneau et al., 1987*) in which cytochalasin-induced reduction of axon tension was assessed. Similarly to (*Letourneau et al., 1987*), we also observed that in cytochalasin-treated cultures, the axons took a longer time (16 min in two experiments, compared to 6 and 10 min for untreated network) to detach from the substrate and retract when exposed to trypsin, presumably because longer proteolysis of adhesion

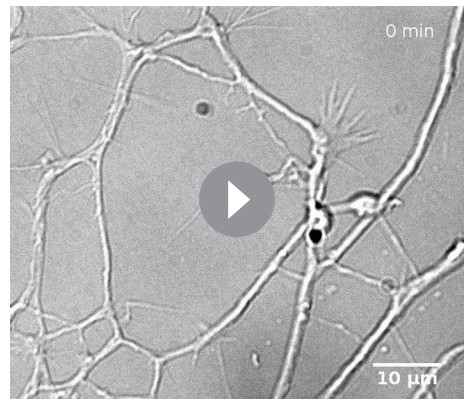

**Video 4.** Growth cone losing grip on the substrate.

molecules is needed before the reduced tension can detach the axons (*Letourneau et al., 1987*).

As a second strategy aiming to perturb axon tension, we tested blebbistatin, an inhibitor of NMII, previously shown to decrease cell cortex/membrane tension in non-neuronal cells (*Fischer-Friedrich et al., 2014*; *Ayala et al., 2017*) and to decrease tension generated in smooth muscle (*Ratz and Speich, 2010*). In growth cones of isolated DRG neurons, Sayyad et al. (*Sayyad et al., 2015*) showed that blebbistatin reduced the force exerted by lamellipodia, but surprisingly increased the force exerted by filopodia. Other studies found that the blebbistatin-dependent NMII inhibition may have opposite effects on axon extension, depending on the substrate. For example, in Ketschek et al. (*2007*), while inhibition of NMII promoted peripheral dorsal root ganglia (DRG) axon extension on polylysine, it decreased axon extension on laminin. Hur et al. (*Hur et al., 2011*), who observed a positive effect of blebbistatin on DRG axon extension on a laminin substrate (thus the opposite result), discussed how these puzzling differences may be due to differences in laminin concentrations, or to the different adhesive properties of polylysine (used in [*Hur et al., 2011*] and in our study) and polyornithine (used by [*Ketschek et al., 2007*]). These studies suggest that while blebbistatin decreases cell cortex contractility, its effect on axon shaft tension may depend on additional factors. In our model system, we did not observe an obvious effect of blebbistatin on axon tension or on axon extension. Further analyses would be required to explain why blebbistatin has a stabilizing effect on OSN axons grown in our experimental conditions.

Since the biological agents having the ability to specifically stimulate axon growth or motility in our cultured explants are currently unknown, we tested FBS for this purpose, because of its established content of a variety of bioactive molecules and growth factors. It turned out that FBS did not have any significant boosting effect on OSN growth cones but, very interestingly, it induced the apparent contraction of the whole explant itself, leading to the generation of pulling forces on axons from the explant core. This apparent contraction of the explant is likely to be the result of changes in the shape of the individual cells constituting the explant, if such changes lead to an overall rounding of previously flattened or elongated cells. In line with this hypothesis, in (*Jalink and Moolenaar, 1992*), the serum induced a rapid rounding of cultured differentiating neural cells, an effect which may likely be due to lysophosphaditic acid, which by itself induces both cell rounding and neurite retraction (*Jalink et al., 1993*).

A quantitative estimate of the increase in axon tension due to the FBS-induced pull may be obtained from the measured stretch of the axons and the expected axon elongation stiffness. The measurements of (*Dennerll et al., 1988*, *1989*) showed the spring constant of PC-12 and DRG neurites (which have baseline tension and length comparable to the axons in our system) to be of order 100 $\frac{\text{pN}}{\mu\text{m}}$. Assuming a similar axon stiffness for our system, the axon stretch of ∼15 μm in the FBS experiments (*Figure 5—figure supplement 1*) is expected to generate a tension increase of over 1 nN. This is further supported by our tension measurements in the experiment of *Figure 8—figure supplement 1*, where a pull of 6 μm was correlated with increase of tension by about 1.2 nN. The estimated FBS-induced tension increase of at least 1 nN (possibly several nN) is very significant compared to the typical axon tension (under 1 nN) that we recorded in the untreated networks. This tension increase, which was achieved within 20 min of the application of FBS, preceded a marked and rapid de-coarsening in parts of the network (*Figure 5F-H*).

We have thus shown on the network level that the extent of axon fasciculation can be regulated by changes in axon tension: an overall decrease in tension leads to zippering-driven coarsening, while an overall increase in tension leads to unzippering and de-coarsening. The estimated tension increase (of order 1 nN) generated endogenously by the FBS-induced explant pull is comparable in magnitude to active changes in axon tension demonstrated in previous literature, such as the tension recovery within 30 min after axon unloading in vitro (*Dennerll et al., 1989*) and in vivo (*Rajagopalan et al., 2010*). Our results therefore demonstrate the control of axon zippering by tension changes of functionally relevant magnitude.

As the axon network coarsens, the zippers become predominantly formed by axon fascicles, rather than by individual axons. We derived the expected distribution of tensions among the fascicles and combined it with the expected scaling of the fascicle-fascicle adhesion force (proportional to the surface area of the fascicle) to predict how the distribution of equilibrium zipper angles in the network depends on the mean number of axons per fascicle. This theoretical prediction was consistent with the observed relation between the median zipper angle and network coarsening, thus

supporting the framework in which zippering is controlled by the competition between tension and adhesion not only for individual axons, but also on the fascicle level.

A direct test of our theoretical models, be it on the single-zipper level or on the network level, would require a measurement of tension changes in the evolving network and following pharmacological manipulations. The tension generated by the pull of the growth cone may be efficiently assessed using traction force microscopy (*Style et al., 2014*). In this technique, the net traction force is determined from the deformation of a suitable hydrogel substrate with embedded tracer beads, and corresponds to the axon tension directly behind the growth cone (*Koch et al., 2012*). One may expect, however, that far behind the growth cone, where the zipper vertices are located, the axon tension can differ, due to force dissipation at substrate attachments along the axon and due to the tension generated directly within the axon shaft. A recently developed contact-less technique that may allow the monitoring of axon tension near the zipper vertex is thermal fluctuation spectroscopy (TFS), adapted to transverse fluctuations of long protrusions. In *Gárate et al. (2015)*, this technique was used to obtain time-resolved estimates of axial tension in PC-12 neurites. Each tension measurement by TFS requires, however, to expose the axon to hundreds of short laser pulses, and relies on a fitting of the measurements to a phenomenological biomechanical model of the axon shaft, in order to extract combinations of viscoelastic parameters. A TFS procedure validated in our system would potentially allow the monitoring of increases or decreases in axon tension, with sufficient temporal resolution (~10 s) to correlate these tension changes with the dynamics of individual zippering events in the developing network.

## Topological analogies between progressive axon fasciculation and the coarsening of liquid foams

Topologically, the structure of the axon network observed in our study is similar to the structure of foams (froths) with a low liquid fraction. A froth consists of gas bubbles that are separated by liquid film walls, with surface tension in the walls (*Weaire and Hutzler, 2001*). The typical structure of a 'two-dimensional froth' (obtained when the foam is restricted between two closely spaced glass plates) is shown in *Figure 13C*. Similarly to the axon network (*Figure 1*), the structure is defined by vertices (junctions) at which three segments under mechanical tension meet. In froths (*Weaire and Hutzler, 2001*; *Glazier and Weaire, 1992*), the tensions in the three walls are equal, resulting in approximately 120° angles at the junction. In the axon network, the triple junctions are formed predominantly by axon zippers. At each junction, the mechanical tension in the zippered segment is necessarily the largest (being the sum of tensions in the unzippered segments) but is effectively lowered by the axon-axon adhesion force (see *Equation 1*); the three angles between the segments are generally unequal. Closed loops that are formed by axon or fascicle segments are the topological equivalent of bubbles in the froth.

In froths, the walls of a bubble with fewer than six sides are on average curved, making the bubble slightly rounded rather than polygonal; this is associated with a surplus of air pressure (*Weaire and Hutzler, 2001*). Gas diffusion out (into the bubbles with lower pressure) leads to the shrinking and disappearance of the rounded bubbles. This *T2* process is illustrated for a three-sided bubble in *Figure 13D1, D2*. A second type of topological rearrangement (denoted *T1* in the literature) observed in coarsening froths is shown in *Figure 13D3, D4*. In this process, two bubbles (at top and bottom) come into contact, pushing out the bubbles at left and right; the bubble walls are reconnected. Through a combination of *T2* and *T1* rearrangements, the froth coarsens (*Weaire and Hutzler, 2001*; *Glazier and Weaire, 1992*); typically, the coarsening will proceed indefinitely (until the size of the container is reached). The froth coarsening is not driven by changes in wall tension.

No comparable coarsening mechanism, associated with pressure differences, exists in the axon network. However, we have repeatedly observed the shrinking and disappearance of loops formed by three axon/fascicle segments (see Results and *Figure 13A*); this is a topological analogue of the *T2* process known from foam dynamics. The reconnection of bubble walls (the *T1* process) has no topological analogue in the axon network. Rather, an attempt to implement such a process leads to a complete unzippering (*Figure 13E1,E2*), which is topologically equivalent to the third elementary process of foam dynamics—a wall rupture (*Glazier and Weaire, 1992*). Such a process was observed in the axon network only infrequently, consistent with the overall predominance of zippering over unzippering.

As explained in Results, the shrinking or expansion of a loop of axon segments can result from the zipper angles at the loop vertices being smaller than or larger than the equilibrium zipper angle, respectively. This is directly supported by our observations of expanding triangular loops in the experiments in which a pull was generated on the network (*Figure 5F-H*). Basic geometry implies that on average, the zipper angle is 60° at vertices that form a triangular loop (as in *Figure 13A*), and 90° at vertices that form an analogous rectangular loop. The equilibrium angles recorded from static zippers formed by single axons or small fascicles (*Figure 7C*) are predominantly below 90°, with the mean value of 51°. This suggests that at the early stages of network coarsening, rectangular loops (with sides formed by single axons) are unstable toward expansion, while triangular loops are closer to equilibrium and may either slowly retract or slowly expand. Such tendency may change in the later stages of network development, however, as axon fascicles can form loops with complex structure and modified equilibrium angles. A more detailed study would be required to characterize the stability of loops in the evolving axon network and its role in the coarsening process.

Analogies between froths and biological tissues consisting of closely packed cells have been investigated in previous literature (e.g. [*Kafer et al., 2007*; *Corson et al., 2009*]). Such analogies are complicated by two aspects: (i) the active cortical mechanics of the cells (which make the interfacial tension dependent on the cell configuration [*Kafer et al., 2007*; *Manning et al., 2010*]), (ii) and restrictions on the volume of animal cells (which prevent pronounced coarsening). However, in (*Corson et al., 2009*), a growing plant tissue (meristem of *Arabidopsis thaliana*) was converted into a 'living froth' when oryzalin was used to depolymerize microtubules attached to the cell walls. In the resulting tissue, the topology and geometry of the cell interfaces was consistent with a typical froth, and pronounced coarsening of the structure was observed during the plant growth. The system we investigated—the zippering axon network—presents a remarkable example of an ex vivo system exhibiting both topological and dynamical analogies to froths. In our system, the coarsening is not limited by any cell volume restrictions and can proceed rapidly, on the time scale of hours. However, structural features such as complex loop configurations and entangled zippers can limit the final extent of coarsening.

## Axon zippering in vivo: its regulation and functional significance

We report in the present paper that OSN axons grown on a planar glass substrate covered homogeneously with polylysine and laminin display extensive zippering behavior, raising the possibility that these axons, and more generally other types of axons with adhesive interactions, may zipper in vivo.

While a direct imaging of zippering dynamics in vivo is difficult to achieve in mice with current methods, strong indirect evidence for axon zippering has been obtained in some other model organisms. In *Xenopus* embryos, the geometry of the sensory neurite plexus on the basal lamina of trunk skins shows features that are likely to be the result of axon zippering, as proposed by Roberts and Taylor (*Roberts and Taylor, 1982*) and discussed in the last subsection of Results. This in vivo configuration has strong similarities to the ex vivo axonal network we studied—a result of the shared planar character and absence of obstacles to zippering in these two systems. In *C. elegans*, axon fascicles from the left and right ventral nerve cords fuse into a single fascicle if a specific medial interneuron is ablated at a late stage, when the axons have already reached their targets (*Aurelio et al., 2002*). This 'axon flip-over' phenotype appears very likely to be due to axon shaft zippering, as evidenced by the abnormal fasciculation profiles observed between shafts which never fasciculate in non-manipulated animals. It was further shown that this phenotype was absent in immobilized animals, indicating that axon zippering was facilitated by mechanical forces exerted during the wriggling locomotion of the worm (*Aurelio et al., 2002*). From a mechanistic point of view, the inhibition of zippering in wild-type animals is due to secretion by the medial interneuron of a 2-immunoglobulin-domain protein, which was proposed to bind and inhibit the activity of homophilic molecules expressed by the left/right contralaterally analogous axons (*Aurelio et al., 2002*). This study nicely illustrates the ability of axon shafts to zipper in vivo, in this case with detrimental developmental consequences. It further provides a framework in which inhibition of axon-axon adhesion negatively regulates zippering.

We next discuss the zippering potential of axons in the mammalian nervous system. In the developing neural systems, the high volume of extracellular space (*Lehmenkühler et al., 1993*) is highly compatible with zippering/unzippering of axons. Despite its reduction as the development

proceeds, this extracellular space (representing 15% of the volume of the adult cortical tissue [*Korogod et al., 2015*]) still provides a suitable environment for axon shaft dynamic interactions, as far as individual axons are not separated from each other by glial or parenchyme boundaries. Central and peripheral myelinated axons will obviously lose their zippering/unzippering abilities as soon as the myelination process begins. Similarly, unmyelinated axons within peripheral nerves, which become separated from each other by Schwann cell cytoplasmic processes as the nerves mature (*Elfvin, 1958*), will not be able to zipper thereafter. However, no such isolation by glial cells of unmyelinated axons occurs in the central nervous system, where shaft-shaft contacts persist in numerous areas.

Our work with OE explants was motivated by the projection pattern of the in vivo olfactory system, where the growth of OSN axons from the OE toward the OB leads to the formation of fascicles. Interestingly, the type of glial cell associated to these axon bundles, the olfactory ensheathing cells (OECs), never myelinate, cover nor even separate individual axons from each other: they instead wrap large bundles containing up to thousands of naked axons (*Li et al., 2005*). Within these bundles, olfactory sensory axons are free to interact directly with each other, from early development onwards, because of the continuous production of OSNs throughout life (*Farbman, 1994*). One can thus speculate that the zippering we observed in vitro may occur and serve some functions in vivo.

An appealing hypothesis is that the zippering of olfactory axons may participate in the sorting of olfactory axons. This sorting is indeed critical given the facts that each OSN expresses one odorant receptor (OR) gene, picked out of a large repertoire of roughly 1,000 OR genes, and that the axons of all OSNs expressing the same OR (these OSNs are distributed across a large zone of the OE) converge into a few glomeruli of the OB (reviewed in [*Mombaerts, 2006*; *Mori and Sakano, 2011*]). This projection pattern results from a multistep process involving the regulated expression of adhesion and guidance cues, some of which under the control of ORs (*Key et al., 2002*; *Nedelec et al., 2005*; *Strotmann and Breer, 2006*; *Mombaerts, 2006*; *Nishizumi and Sakano, 2015*; *Zapiec et al., 2016*; *Assens et al., 2016*). In addition, an OR-independent pre-sorting of OSN axons, leading to the segregation of Class I vs. Class II OSN axon types within the nerve branches, has been reported (*Bozza et al., 2009*). An OR-dependent sorting ultimately ensures that the axons of OSNs expressing the same OR are segregated from each other in purely innervated glomeruli. We did not carry out labeling to distinguish axon subtypes expressing for example specific adhesion molecules in our cultures, and therefore, we do not know if the dynamic axon-axon interactions we observed in vitro are related to any sorting process. Since it takes about 4 days for newborn OSNs to express their OR (*Rodriguez-Gil et al., 2015*), the OSN growing in our cultures probably do not express their OR at the time point of our analyses, precluding that any OR-dependent sorting would occur in these cultures. It remains to be investigated if zippering of olfactory axons, in conjunction with OR-specific adhesion between axon shafts or OR-specific tension differences, may play a role in the axon sorting process in vivo.

More generally, what are the expected functional consequences of regulated axon shaft zippering in vivo? First, early on as a growth cone navigates toward its target, the ability to zipper can regulate the probability with which it would cross a fascicle or not, as we illustrated in Results with the Roberts and Taylor data (*Roberts and Taylor, 1982*). Second, once growth cones are already at distance on the way toward their target area, the extent to which their shafts are fasciculated may be regulated through zippering or unzippering. During both the development and maturation of neural networks, ephaptic interactions between axons may be favored in tightly fasciculated segments, thus influencing the synchrony of transmitted action potentials, or generating ectopic spikes (*Bokil et al., 2001*). One could speculate that controlling the degree of fasciculation of axons through a regulation of zippering may be used to modulate such ephaptic interactions. Third, the resulting structure of the fascicles may have important consequences for subsequent steps of development and maturation of the networks. Indeed, while tightly fasciculated small bundles of pioneer axons constitute a robust path for follower axons, loosening the axons within fascicles might be beneficial for their myelination. Finally, in pathological contexts of axon regeneration following injury, or of axon demyelination, the unmyelinated axons or axon segments may zipper up in tracts. In tightly bundled tracts of partially demyelinated axons, ephaptic interactions are predicted to permit recovery of robust conduction (*Reutskiy et al., 2003*).

In light of our analysis, there are two principal ways by which zippering and hence the extent of fasciculation may be regulated in vivo. In a developing network, the growth cone activity or shaft

cytoskeleton activity (*O'Toole et al., 2015*) can change the axon tension (*Rajagopalan et al., 2010*) and hence influence fascicle structure on fast time scales (dozen minutes). On a slower time scale, fasciculation can be regulated by changes in CAM expression or their post-translational modifications. For example, it has long been established that axonal N-CAM is involved in axon-axon adhesion and regulated post-translationally by addition or removal of polysialic acid (PSA); high levels of PSA on N-CAM decrease cell-cell adhesion (*Hoffman and Edelman, 1983*; *Sadoul et al., 1983*; *Rutishauser et al., 1983*). Axon fasciculation is also regulated by external guidance cues, through a variety of signalling pathways. For example, matrix metalloproteases promote motor axon fasciculation in Drosophila (*Miller et al., 2008*), secreted Slit2 promotes motor axon fasciculation via an autocrine and/or juxtacrine mechanism in the mouse embryo (*Jaworski and Tessier-Lavigne, 2012*), EphA4 expressed by otic mesenchyme cells regulates in a non-cell autonomous manner the spiral ganglion axon fasciculation in the mouse auditory system (*Coate et al., 2012*), and Neuropilin1 mediates inter-axonal communications before and within the plexus region of the limb, thus regulating the fasciculation of sensory and motor projections (*Huettl et al., 2011*). It remains to be established if and how these signals affect axon-axon adhesion or possibly axon tension.

The tension of an axon shaft is influenced by the traction force generated by its growth cone, which in turn depends on the mechanical properties of its environment (reviewed in [*Athamneh and Suter, 2015*]), likely through micro-scale elastic deformation of adhesion complexes between the axon actin network and the substrate (*Athamneh et al., 2015*; *Mejean et al., 2013*). In (*Koch et al., 2012*), the growth cone traction force was found to increase with the substrate stiffness (except for very rigid substrates); a similar relation was found in non-neuronal cells (*Ghibaudo et al., 2008*; *Yip et al., 2013*) (reviewed in [*Kerstein et al., 2015*]). Spatial changes in substrate stiffness may therefore regulate the distal axon shaft tension and hence the extent of zippering, potentially triggering fasciculation/defasciculation of a population of axons during development, when their growth cones arrive to a specific target area. In (*Koch et al., 2012*), a ~1 nN gradual increase in growth cone net traction force was recorded for DRG neurons plated onto a stiff substrate. In our experiments with FBS-induced explant pull, we observed marked defasciculation following an estimated tension increase of comparable magnitude (see previous subsection). This suggests that significant changes in fasciculation may result from growth cone transitions between tissues with distinct elastic properties. Similarly, the general increase in stiffness of brain tissue during development (*Franze, 2013*) may gradually increase the GC traction force and as a consequence facilitate unzippering and defasciculation as the growing tracts differentiate. In comparison, the substrate stiffness in our ex vivo experiments is more homogeneous and static, simplifying the zippering-driven dynamics.

In conclusion, our work shows that adhesion-driven zippering of axon shafts can induce the formation of axon fascicles without a direct involvement of the growth cones. However, active changes in the pulling force at the growth cone, and hence in axon tension, may be used as a mechanism to control the extent of zippering and to regulate fasciculation/defasciculation. Mechanical tension has been shown to play important roles in neural development (reviewed in [*Franze, 2013*; *Franze et al., 2013*]), and recent studies have demonstrated that changes in axon tension can affect the formation of neural circuits by regulating neurite differentiation (*Lamoureux et al., 2002*), axon branch survival (*Anava et al., 2009*), as well as synaptic structure (*Siechen et al., 2009*). Our work introduces a novel role of axon tension in neural circuit assembly: the regulation of fasciculation/defasciculation through the control of axon shaft zippering.

## Materials and methods

### Olfactory epithelium explant cultures

All animal procedures were approved by the Île de France Ethics Committee. Pregnant female Swiss mice were sacrificed by cervical elongation at embryonic day 13.5 (E13.5), embryo were extracted from the uterus, and olfactory epithelium explants were prepared from the posterio-dorsal quarter of the septum and turbinates as follows. First, these posterior and dorsal parts of septa and turbinates were cut into pieces in L15 medium (Gibco 21083, Gibco Thermo Fisher Scientific, Waltham, Massachusets) maintained on ice at 4°C, before being subsequently incubated for 30 min at 25°C in a solution of 1:1 of Trypsin 0.25% (Gibco 25050) and Pancreatin 4X USP (Gibco 02-0036DG) to allow

the OE to separate from the lamina propria. Enzymatic reactions were stopped by adding 10% Fetal Bovine Serum (FBS, Gibco 10270), and the biological material was rinsed in ice-cold L15 containing 5% FBS. Pieces of tissue were transferred into a glass Petri dish in which the OE sheets were cut, using a micro-scalpel, into small pieces of about (100–200) μm diameter each. Explants were then carefully transferred into 50 mm diameter IBIDI video dishes (Biovalley, Illkirch, France) that included a 35 mm glass coverslip (for BFP experiments), or into IBIDI μ-slide eight well #1.5 polymer coverslip (Biovalley 80826) (for time lapse acquisition), previously coated with poly-L-lysine (0.2 $\frac{mg}{ml}$, Sigma P1524, Sigma, St-Louis, Missouri) and Laminin (0.02 $\frac{mg}{ml}$, Sigma L2020), and maintained in culture (37°C, 5% $CO_2$) until the day of experiment in a culture medium of DMEM/F12 (Gibco 31331) containing 1% N2 (Gibco 17502), 0.1 $\frac{mg}{ml}$ Gentamycin (Sigma G1272), 1.5% D-Glucose (Sigma G8769), 1% BSA (Sigma A4161) and 7 $\frac{\mu g}{ml}$ Ascorbic acid (Sigma A4403). We typically prepared and put in culture 40 to 60 explants per set of experiments coming from 10 to 12 embryos.

## Videomicroscopy

10 mM Hepes was added to the explant cultures 1 hr before starting time lapse acquisitions. In some experiments, the cultures were treated with FBS (Gibco, 5% final concentration), blebbistatin (Sigma B0560, 10 μM in culture medium containing a final concentration of 0.1% DMSO), cytochalasin B (Sigma C6762, 2 μM in culture medium containing a final concentration of 0.1% DMSO), or trypsin (Gibco 25050, 0.25% in culture medium). Videomicroscopy was performed on a Leica DMI 6000B (Leica, Wetzlar, Germany) inverted microscope in a thermostated chamber (37°C, 7% $CO_2$ at the rate 10 l/h, (87–95) % relative humidity) using a DIC 63× NA 1.40 IMM, or a dry phase contrast objective 40× NA 0.75 Leica HCX PL APO, and a CCDcoolSNAP HQ2 camera (Photometrics, Tucson, Arizona) driven by Metamorph 7.1, in a multiple acquisition mode. Typically, 9 Z steps with an interval of 1 μm were acquired each minute for each of the 8 to 10 positions chosen around explants. Recording of each experiment lasted 2 to 19 hr.

## Analysis of the videomicroscopy recordings

The pool of recordings of network evolution contained 13 explants where no drug was added, 15 explants where cytochalasin was added (pre-treated with blebbistatin in 11 cases), and 10 explant where FBS was added (pre-treated with blebbistatin in five cases). Major criteria for selection for quantitative analysis were good contrast, culture survival, and sufficient area and density of the network. The reported data are based on analyses performed for (i) $N = 6$ recorded experiments with no added drug (originating from three individual animals), of which network coarsening did not occur in one experiment, which was therefore excluded from statistical analyses, and (ii) $N = 12$ recorded experiments in which cytochalasin was applied (originating from six individual animals), eight of which were pretreated with blebbistatin. The quantitative analysis was performed on video recordings in which the axonal network showed clear evolution lasting over 1 hr. Initial preprocessing and manual segmentation were performed using the distribution Fiji (*Schindelin et al., 2012*) of the project ImageJ (*Schneider et al., 2012*). The field was cropped to restrict the region of interest and 6 to 10 images (frames) uniformly spaced in time were chosen from the course of the recording. The network of axons was then manually segmented by drawing individual selection lines over the image. In some cases, successive frames were consulted to decide whether a line is an axon to include or a transient side-process. The list of segmentation selections was exported to Matlab (*The Mathworks, Inc, 2015*), where a set of custom-made functions was used to: (i) convert the list of selection lines into a graph data structure, (ii) detect cordless loops in the graph, (iii) semi-automatically measure zipper angles or determine crossing points, (iv) calculate the network statistics (notably the zipper angle distribution, total network length, number of vertices, average area of cordless loops), and (v) determine correlations between these statistics.

The segmentation selections underlying the analysis shown in *Figure 6G-E* can be found in the *Figure 6—source data 8* source file. They can be displayed in ImageJ by opening the TIFF file and the corresponding ZIP file with the segmentation; the segmentation can be laid over the image by checking the box 'Show All' on the 'ROI Manager' window. The Matlab script and input data which can be used to perform the fully automatic post-segmentation steps are provided in *Figure 6— source data 1*. Segmentation coordinates for *Figure 1* are provided in *Figure 1—source data 1*. Regarding the data of dynamics of individual zippers, 17 events were measured by manually tracking

the coordinates of the zipper vertex in consecutive frames. The measurement was accepted only if the vertex remained in static equilibrium 5 min before and after the transition. Zipper measurements were obtained from 14 explants originating from four mother animals. They were chosen from the networks of low density to minimize disturbance from the areas adjacent to the zipper.

## Scanning electron microscopy

Scanning electron microscopy explants were cultured on a 14 mm diameter coverslip (as described above), fixed for 1h at 4°C in 2% glutaraldehyde prepared in 0.1 M sodium cacodylate buffer, rinsed in cacodylate buffer, dehydrated in a series of graded ethanol baths, and dried using a critical point dryer (Quorum Technologies CPD7501, Laughton, UK). They were finally mounted on a carbon stub and sputter-coated. Observations were made using a Cambridge Instruments Stereoscan 260 scanning electron microscope equipped with a digital camera.

## Force measurements with BFP

The implementation of the BFP method (*Evans et al., 1995*) was adapted from (*Gourier et al., 2008*). Basically, this method uses a force transducer composed of a biotinylated red blood cell (RBC) held by a glass micropipette (treated with BSA), and a streptavidin-coated glass microbead (3 μm diameter), linked to the RBC by a streptavidin-biotin bond. In our experimental design, the bead was attached to axons of the culture previously treated for surface biotinylation. Within the range of forces measured in our experiments, the RBC force-deformation relation is linear, and the RBC behaves as a spring of stiffness $k$ determined by the geometry of the probe and by the aspiration pressure $\Delta P$ within the pipette:

$$k = R_p \Delta P \frac{\pi}{\left(1 - \hat{R}_p\right)} \frac{1}{\log\left(\frac{4}{\hat{R}_c \hat{R}_p}\right) - \left(1 - \frac{1}{4}\hat{R}_p - \frac{3}{8}\hat{R}_p^2 + \hat{R}_c^2\right)} \tag{5}$$

where $R_p$ (0.6–1.0 μm) and $R_C$ (0.75–1.2 μm) are the internal pipette radius and the radius of contact between the RBC and the bead respectively. The hat designates the corresponding radius divided by the radius of the aspirated unstrained RBC (2–3 μm). An adjustment of the pressure allows to set up the desired stiffness, $k = 100 - 400 \frac{\text{pN}}{\mu\text{m}}$. By measuring the extension of the RBC, we could calculate the force exerted by the probe on an attached axon.

For these experiments, 2 days in vitro (DIV) OE explants, cultured in 50 mm IBIDI dishes, were biotinylated using EZ-Link Sulfo-NHS-SS-Biotin (Thermo Fisher Scientific 21328, Waltham, Massachusets) according to the manufacturer instructions. Dishes were then transferred into the thermostated chamber (37°C) of the Leica DMIRB inverted microscope equipped with micropipette manipulators and a CCD digital camera (purchased form JAI, Yokohama, Japan). Streptavidin beads were added to the culture, a micropipette (1.5–2 μm inner diameter) was filled with the culture medium and fixed onto the mechanical micropipette manipulator (*Gourier et al., 2008*). The diameter of the pipette was measured using the 40× objective and the CCD camera. Biotinylated RBCs were added to the culture medium (*Gourier et al., 2008*).

Then, a RBC was aspirated into the micropipette ($\Delta P$=200–250 Pa) and put in contact for at least two 2 min with a bead attached to an axon or a small axon bundle. After an adhesive contact had been formed between the bead and the RBC, the pipette was slowly moved (see *Figure 8—source data 1*) in order to pull or to push the axon(s), being recorded by the CCD camera. In favorable cases, when the bead-axon contact adhesion was sufficiently strong, the pulling or pushing of the pipette lead to a deformation (elongation or compression respectively) of the RBC, often resulting in lateral deflection of the axon (see *Figure 8—source data 1*). After the pipette movement, we paused to let system relax; an equilibrium would be reached between the force induced by the probe and the transverse projection of the reaction force of the axon axial tension. Several steps of pulling or pushing were performed for each bead, gradually increasing applied force and axon deflection, until the bead detached from the RBC. The whole process was recorded on the CCD camera and the recording analyzed.

## Analysis of BFP data

The analysis was based on the captured recordings, recorded at rate of 65 fps. The BFPTool software package (*Šmít et al., 2017*) was used to subdivide each recorded video into intervals suitable for automated analysis, and then to track the pipette and bead position with sub-pixel precision. For the algorithms used, please refer to (*Šmít et al., 2017*). The distance of the centre of the bead and a fixed point on the pipette tip were used to represent the length of RBC. The length of unloaded RBC was determined from a frame where the bead first touches the approaching RBC. Having established the RBC stiffness (from geometry and pressure, using *Equation 5*), the applied force was calculated for every frame. The force was corrected by a projection to the direction normal to manipulated axon.

Then, the stable plateaux would be identified in the force time course, and for each, the average force $F_i$ applied by the probe (over the duration of the plateau $i$) and the angle of axon deflection $\delta_i$ would be determined. Such pairs of values, $(2 \sin \delta_i, F_i \sin \phi_i)$, constituted our data points for each experiment ($\phi_i$ represents the angle between pipette axis and axon axis). Finally, linear interpolation of the acquired data points was performed to obtain the tensile force within the axon as the slope of the interpolating line. Non-zero intercept of the interpolation line was often present; this happens when the selected reference distance does not truly correspond to the unstrained size of the RBC. With the slope determined from several plateaux (we chose experiments having at least three), this offset does not influence the resulting calculated axonal tension.

The uncertainty of the tension measurement has three sources. The uncertainty of stiffness of the BFP, $\delta(k) \approx 14\%$, given by the limited precision of measurement of probe radii and aspiration pressure (see *Equation 5*). The uncertainty of RBC deformation measurement; while pipette pattern matching is generally very robust and precise, the tracking of the bead centre is more sensitive to perturbations and can introduce an error of (10–50)nm—see (*Šmít et al., 2017*). Lastly, the most important source of measurement uncertainty is the deflection angle $\delta_i$; the value of the deflection angle is small (<5°) while the precision of measurement is limited by diffuse edges of axons at $\Delta \delta_i \approx 0.5°$. The change of angle is small between consecutive frames (at 65 fps), so the precision can be improved by averaging several measurements, giving an upper limit on the relative error, $\delta(\delta_i) \leq 25\%$ for the smallest angles measured.

The axon tension is obtained by a linear regression of time-averaged quantities; the variability of applied force and deformation angle over the duration of each plateau is shown by error bars in *Figure 8E*. The final reported error of axon tension measurement is the standard deviation of the slope of regression.

## Calculation of distributions of biophysical parameters in the axonal population

Each BFP experiment resulted in a value of tension and its experimental uncertainty for the given axon. This pair of parameters was used to construct the corresponding normal distribution, representing the tension of each axon. These Gaussian distributions were added and their sum normalized, to approximate the distribution of tensions within the whole axonal population.

Similarly, the set of the measured equilibrium zipper angles (described in Results) was transformed into a distribution, by convolving the dataset with a Gaussian kernel (using Matlab's kernel distribution functions).

Two complementary approaches were used to estimate the value of axon-axon adhesion strength $S$: in the first, the measured distribution of tensions and the distribution of zipper angles are fully determined by each other, while in the second approach, the tensions and angles are treated as statistically independent variables. In the first approach, a fixed value of $S$ was assumed, and *Equation 1* was used to transform the measured distribution of tensions $p(T)$ into a distribution of angles $q(\beta)$:

$$q(\beta) = \frac{p(T(\beta))}{\left|\frac{d\beta}{dT}\right|} = p\left(\frac{S}{2\Phi}\right) \frac{S}{4} \frac{\sqrt{2 - \Phi}}{\Phi^{\frac{3}{2}}}, \text{where } \Phi = 1 - \cos\frac{\beta}{2} \tag{6}$$

where the relation between $T$ and $\beta$ is specified by *Equation 1*. The correspondence (evaluated as correlation) between the distribution $q(\beta)$ and the experimentally obtained angle distribution was maximal for $S$=88 pN (correlation coefficient 0.813). As shown in *Figure 8—figure supplement 2A*, the experimentally determined distribution was wider than the transformed distribution, suggesting

that in reality the zippers do not all have the same value of $S$. In the second approach, we estimated an upper bound on the spread of $S$ values. We constructed the joint distribution, *Figure 8—figure supplement 2B*, of tensions and angles as the product of the measured tension and angle distributions (thus treating the tension and the angle as mutually independent), and computed the distribution of adhesion strengths $S$ defined by this joint distribution and *Equation 1*. To do so, the values of $S$ were discretized in 1 pN bins and the probabilities of (tension, angle) pairs that gave $S$ in the given bin were integrated. The resulting distribution of $S$ is shown in *Figure 8—figure supplement 2C*. As the tensions and the angles are in reality expected to be partially dependent, the obtained interquartile range $S=(52–186)$ pN should be viewed as the upper bound on the spread of $S$ values. The obtained median (102 pN) is consistent with the value of $S$ obtained from the first approach. The Matlab scripts for performing the calculations described in this subsection are provided as source files associated with *Figure 8—figure supplement 2*.

## Dynamical model of axon zippering

For the general asymmetric axon zipper (as in *Figure 11A*, with mobile vertex $V$ and fixed points A, B, C), the static equilibrium condition and the equation of motion were derived as follows.

We assumed that the vertex motion is sufficiently slow to allow the tension forces to keep the axon segments straight during zippering or unzippering; this is consistent with the experimental observations (*Figure 3*). Mechanical stresses were assumed to be uniform along each axon. We neglected elastic forces arising from axon bending in the immediate vicinity of the zipper; in zippers formed by single axons or small fascicles, the axons form a sharp bend (*Figure 3*), indicating that the bending rigidity is low.

This assumption is further supported by the following quantitative arguments. Considering the flexural rigidity of a microtubule $(EI)_{MT} \lesssim 1 \times 10^{-1} \mathrm{nN}\,\mu\mathrm{m}^2$ (*Pampaloni et al., 2006*), at most 10 microtubules in each axon (*Fadić et al., 1985*), and a radius of curvature of the axons $R \approx 1\,\mu\mathrm{m}$ at the vertex, the density of energy of flexure can be estimated as $\frac{10 \cdot (EI)_{MT}}{2R^2} \lesssim 1 \times 10^{-1} \mathrm{nN}$ ([*Roark et al., 2002*, p. 127), that is an order of magnitude lower than the axial tensile energy density (the axon tension, see Results). For a bundle of axons, the bending energy is expected to scale quadratically with the number of axons, while the tension scales linearly. The two energy densities are therefore expected to become comparable in the vicinity of the zipper vertex only for bundles of $\geq 10$ axons. We note that while the energy stored in the elastic flexure is neglected in our model, the energy dissipation resulting from the disruption of microtubule-associated cross-linking proteins and other bending-related structural changes is included in the empirical vertex-localized friction force introduced in Results.

When formulating the dynamical model of zippering, we further assume that the tension in the constituent axons is constant in time. As the time scale for a simple zippering or unzippering process is of order 10–20 min (see Results), one cannot in principle exclude active adjustments of axon tension accompanying the zippering or unzippering, or a coupling to active intracellular transport processes. In previous literature, a recovery of tension within 15–60 min was shown for axons that were made slack following a large rapid distension (*Rajagopalan et al., 2010*). Compared to such distension experiments, however, the unzippering dynamics is gradual, and we assume no active tension regulation.

Under these assumptions, the instantaneous zipper configuration is fully specified by the Cartesian coordinates $(x, y)$ of the vertex. The total tension/adhesion energy of the configuration is given by

$$E(x,y) = T_1(|VA| + |VC|) + T_2(|VB| + |VC|) - S|VC| \tag{7}$$

where $|VX|$ denotes the length of the given axon segment, $T_1$ and $T_2$ the values of (effective) tension in the two axons (equivalent to the tensile energy per unit length), and $S$ the energy of inter-axon adhesion per unit length of the adhered segment $VC$. (Treating the tensions $T_1$ and $T_2$ as constants independent of the axon length, we neglect possible Hookean elasticity contributions.) The spatial gradient of the potential energy $E$ defines the mechanical conservative force $\vec{F}_v$ that effectively acts at the vertex and drives the dynamics. The vector $\vec{F}_v$ thus points in the direction along which the energy decreases fastest upon a displacement of the zipper vertex. One straightforwardly obtains

$$\vec{F}_v = -\nabla E(x,y) = T_1 \widehat{VA} + T_2 \widehat{VB} + (T_1 + T_2 - S) \widehat{VC} \tag{8}$$

where $\widehat{VA}$ indicates the unit vector in the $VA$ direction (and similarly for $VB$, $VC$). The right-hand side of **Equation 8** can be interpreted as the vector sum of the forces with which the axon segments $VA$, $VB$ and $VC$ pull at the vertex. The last term, $-S\,\widehat{VC}$, in **Equation 8** is the force of inter-axon adhesion, which has magnitude $S$ and is always oriented anti-parallely to the zippered axon segment $VC$.

A zipper is in a static equilibrium when $\vec{F}_v = \vec{0}$. Spatial components of the force $\vec{F}_v$ can be conveniently expressed in terms of the zipper angles $\alpha_1$ and $\alpha_2$ (see **Figure 11A**). In the direction along the zippered segment, the force equilibrium condition then becomes

$$-(T_1 + T_2 - S) + T_1 \cos\alpha_1 + T_2 \cos\alpha_2 = 0 \tag{9}$$

while in the perpendicular direction

$$T_1 \sin\alpha_1 - T_2 \sin\alpha_2 = 0. \tag{10}$$

Given the parameters $T_1$, $T_2$ and $S$, the **Equations 9,10** specify the angles $\alpha_1$ and $\alpha_2$, and hence the equilibrium vertex position $(x,y)$. It is readily shown that the equilibrium defined by **Equations 9,10** is stable (i.e. $E(x,y)$ has a local minimum at the equilibrium point). In the special case of a symmetric zipper (i.e. $T_1 = T_2$), **Equation 10** implies $\alpha_1 = \alpha_2$ and **Equation 10** becomes equivalent to **Equation 1** used in our static data analysis.

A nonzero driving force $\vec{F}_v$ will result in motion of the vertex, with a velocity $\vec{u} = (\dot{x}, \dot{y})$ such that $\vec{F}_v$ is balanced by an effective frictional force acting at the zipper vertex. (Expressed in terms of energy, the rate of change of $E(x,y)$ when the vertex moves must equal the rate of energy dissipation in the entire zipper configuration—see Appendix). Assuming a frictional force proportional to the vertex velocity, the resulting equation of motion is

$$\vec{F}_v = \overset{\leftrightarrow}{H}(x,y)\vec{u} \tag{11}$$

where the friction tensor $\overset{\leftrightarrow}{H}$ is independent of $\vec{u}$ but may in general depend on the zipper configuration geometry, specified by the vertex position $(x,y)$. In the simplest case of isotropic and geometry-independent friction, $\overset{\leftrightarrow}{H} = c \cdot 1$ is a constant multiple of unit tensor and the integration of **Equation 11** results in a trajectory that follows the gradient of $E(x,y)$. In case of anisotropic and/or geometry-dependent friction, however, the vertex trajectory deviates from this path.

The form of the vertex friction tensor depends on the dominant mechanism of energy dissipation. In the main text, we introduced two forms of internal energy dissipation in the axons—the viscosity of elongation/shortening, and the vertex-localized dissipation. As the corresponding frictional forces are collinear with the axon tension and with the axon-axon adhesion force, respectively, one may substitute the 'dynamically corrected' tension and adhesion magnitudes (**Equations 2,3**) into the static equilibrium condition (**Equations 9, 10**), and obtain two coupled equations for the zipper velocity components $\dot{x}$, $\dot{y}$. It is straightforward to see that in general, the vertex friction tensor resulting from either of these frictional forces is anisotropic. $\overset{\leftrightarrow}{H}$ depends on the geometry (i.e. the vertex position relative to the fixed points A, B, C) in the case of elongation viscosity but is geometry-independent in the case of vertex-localized friction.

A third form of energy dissipation—friction between the axons and the substrate—is evaluated in the Appendix. In this case, no simple prescription for generalizing the static equilibrium equation at the zipper vertex is available. The corresponding equation of motion is derived by integrating the energy dissipated along the axons, and equating the total rate of dissipative energy loss with the rate of gain of tension/adhesion energy. The Rayleigh dissipation function formalism is used for the unified treatment of all three forms of friction we consider.

## Acknowledgements

We thank Christine Gourier and Pierre Soule for helpful discussion regarding BFP experiments, Susanne Bolte, Jean-François Gilles and the imaging platform of IBPS, Michael Trichet and Virginie

Garnier and the electron microscopy platform of IBPS for SEM imaging, as well as Mohamed Dou-lazmi and Sinan Haliyo for helpful technical discussions. We are grateful to Tomáš Vomastek, Coralie Fassier, Diana Zala and Zsolt Lenkei for useful discussions on pharmacological approaches, and to Isabelle Dusart, Charles Greer and Boris Zalc for their feedback on the manuscript or input regarding the functional significance of our work. We also wish to thank the three anonymous reviewers for their highly constructive comments.

# Additional information

## Funding

| Funder | Grant reference number | Author |
| --- | --- | --- |
| Université Pierre et Marie Curie | | Frédéric Pincet<br>Alain Trembleau |
| Centre National de la Recherche Scientifique | | Frédéric Pincet<br>Alain Trembleau |
| Akademie Věd České Republiky | RVO#67985823 | Martin Zapotocky |
| Czech Science Foundation | 14-16755S | Martin Zapotocky |
| First Faculty of Medicine at Charles University | GAUK 396213 | Daniel Šmít<br>Martin Zapotocky |
| Barrande Czech-French Cooperation program | 7AMB12FR002 | Martin Zapotocky<br>Alain Trembleau |
| Institut National de la Santé et de la Recherche Médicale | | Alain Trembleau |
| Agence Nationale de la Recherche | ANR-2010-BLAN-1401-01 | Alain Trembleau |
| National Institutes of Health | 5R01DC012441 | Alain Trembleau |
| Agence Nationale de la Recherche | ANR-11-IDEX-0004-02 | Alain Trembleau |

The funders had no role in study design, data collection and interpretation, or the decision to submit the work for publication.

## Author contributions

DŠ, Conceptualization, Resources, Software, Formal analysis, Methodology, Writing—original draft, Writing—review and editing; CF, Validation, Investigation, Visualization, Methodology, Writing—review and editing; FP, Formal analysis, Supervision, Validation, Methodology, Writing—review and editing; MZ, Conceptualization, Formal analysis, Supervision, Funding acquisition, Validation, Investigation, Methodology, Writing—original draft, Project administration, Writing—review and editing; AT, Conceptualization, Resources, Supervision, Funding acquisition, Validation, Investigation, Methodology, Writing—original draft, Project administration, Writing—review and editing

## Author ORCIDs

Martin Zapotocky, http://orcid.org/0000-0002-0437-2528
Alain Trembleau, http://orcid.org/0000-0002-8290-0795

## Ethics

Animal experimentation: Procedures involving animals and their care were conducted according to European Parliament Directive 2010/63/EU and the 22 September 2010 Council on the protection of animals.

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

## Appendix 1

This appendix formulates the effective equation of motion for the zipper vertex, taking into account three forms of energy dissipation (i.e., three distinct frictional forces). It complements the section Dynamical model of axon zippering in Materials and methods, where only two forms of friction were considered and a symmetric zipper was assumed. We use the Euler-Lagrange formalism with Rayleigh function, calculated as the total energy dissipation rate in the whole zipper configuration.

## Assumptions

The following assumptions are used: (1) The axons are modelled as one-dimensional viscoelastic filaments with negligible bending energy. (2) The axon segments remain straight between the vertex and the fixation point (the straightening dynamics is assumed to be faster than zippering dynamics). (3) Longitudinal strain is assumed to be uniform along each axon (the strain redistribution along axons is assumed to be faster than the zippering dynamics). (4) The dissipative forces are linear functions of local velocities and are mutually independent.

## Euler-lagrange formalism with Rayleigh dissipation function

Please refer to **Figure 11A** for the geometry of the zipper configuration and for basic notation. The potential energy $U$ of the configuration is given by the sum of tensile and adhesive energies $E(x, y)$, as expressed in **Equation 7** of main text. The kinetic energy $E_K$ is negligible. The Lagrange function $L = E_K - U$ therefore becomes

$$L(x,y) = -(T_1 + T_2 - S)\underbrace{\sqrt{(y-y_C)^2 + (x-x_C)^2}}_{=L_C} - T_1\underbrace{\sqrt{(y-y_A)^2 + (x-x_A)^2}}_{=L_A} - T_2\underbrace{\sqrt{(y-y_B)^2 + (x_B-x)^2}}_{=L_B}.$$

where $L_A = |VA|$, $L_B = |VB|$, $L_C = |VC|$ (see **Figure 11A**). The equation of motion for the zipper vertex is given by the Euler-Lagrange equations

$$\frac{\partial L}{\partial q_i} - \frac{d}{dt}\frac{\partial L}{\partial \dot{q}_i} = Q_i \tag{A1}$$

where $q_i$ are the spatial coordinates of the vertex, $\dot{q}_i$ are the corresponding velocities, and $Q_i$ are frictional forces to be derived from the Rayleigh dissipation function. On the left-hand side, the second term vanishes for the velocity-independent Lagrangian, while the calculation of the first term is straightforward.

Assumption 4 allows us to use the formalism of Rayleigh dissipation function $D$ to express the non-conservative forces in **Equation A1**. This function is defined as $D = \sum_{i,j} \frac{1}{2} K_{ij} \dot{q}_i \dot{q}_j$ and the generalized forces are given as $Q_i = -\frac{\partial D}{\partial \dot{q}_i}$. Here $K_{ij}$ is a symmetric, positive definite matrix of generalized coefficients of friction. The coefficients are independent on velocity but may depend on the coordinates $q_i$. Evaluating the left- and right-hand side of **Equation A1**, the equation of motion becomes, in Cartesian coordinates,

$$-(T_1+T_2-S)\frac{x-x_C}{L_C(x,y)} - T_1\frac{x-x_A}{L_A(x,y)} - T_2\frac{x-x_B}{L_B(x,y)} = -K_{12}(x,y)\dot{y} - K_{11}(x,y)\dot{x} \tag{A2}$$

$$-(T_1+T_2-S)\frac{y-y_C}{L_C(x,y)} - T_1\frac{y-y_A}{L_A(x,y)} - T_2\frac{y-y_B}{L_B(x,y)} = -K_{21}(x,y)\dot{x} - K_{22}(x,y)\dot{y} \tag{A3}$$

To complete these equations, the coefficients $K_{ij}(q_i)$ must be specified. In the next section, three distinct forms of friction are introduced. The general form of the corresponding matrix $K_{ij}$ is given in section Rayleigh function.

## Energy dissipation rates

In our model, we consider friction forces of three distinct origins. The elongational viscosity $\eta^{\Updownarrow}$ and the vertex-localized zippering friction $\eta^Z$ were already discussed in the main text. In addition, here we introduce a frictional interaction with the substrate, while allowing for anisotropy with respect to the axon shaft orientation.

## Substrate friction

This type of friction arises due to the motion of the axons with respect to the substrate, and depends on the entire geometry of the zipper configuration. Consider an axon segment $j$ connecting the vertex with one of the fixed points A,B,C (see *Figure 11A*). The frictional force acting on an element $dl$ of this segment is assumed to be a linear function of the element velocity $\vec{v}(l)$, where $l$ denotes the distance from the fixed point (see *Figure 11B*). We allow for anisotropic friction, with the friction coefficient eigenvalue $\eta^{\parallel}$ for motion parallel to the axon segment (axial friction) and $\eta^{\perp}$ for motion normal to the axon segment (transverse friction). The rate of energy dissipation in the axon element $dl$ is then given by $dR(l) = dR^{\parallel}(l) + dR^{\perp}(l) = \frac{1}{2}\eta^{\parallel}(\vec{v}(l)\cdot\vec{t}_j)^2 dl + \frac{1}{2}\eta^{\perp}(\vec{v}(l)\cdot\vec{n}_j)^2 dl$, where $\vec{t}_j$ and $\vec{n}_j$ denote the unit vectors tangent and normal to the axon segment $j$.

The integration of the dissipation rate in the whole axon segment is simple in the case of transverse friction. Here $\vec{v}(l)\cdot\vec{n}_j = \frac{l}{L_j}\vec{u}\cdot\vec{n}_j$, where $L_j$ is the length of the axon segment and $\vec{u}$ is the vertex velocity. The dissipation rate in the whole segment $j$ is therefore

$$R_j^{\perp} = \frac{1}{2}\eta^{\perp}(\vec{u}\cdot\vec{n}_j)^2 \int_0^{L_j}\left(\frac{l}{L_j}\right)^2 dl = \frac{1}{6}\eta^{\perp}L_j(\vec{u}\cdot\vec{n}_j)^2.$$

The total transverse dissipation rate in each axon is given by the sum of these contributions from the two segments that constitute the axon.

The axial friction case is slightly more complicated, as the elongations within the two axon segments cannot be treated independently. However, as we assume uniform strain all along the axon, the rate of change in length of a given segment may be obtained as the rate of change in length of the whole axon, multiplied by the proportion of this segment in the total length of the axon. I.e., for the two segments constituting the left axon in *Figure 11A*, we have $\dot{L}_A = \frac{L_A}{L}\dot{L}$ and $\dot{L}_C = \frac{L_C}{L}\dot{L}$, where $L = L_A + L_C$. The rate of change in the total length of the axon is simply expressed in terms of the vertex velocity as $\dot{L} = -\vec{u}\cdot(\vec{t}_A + \vec{t}_C)$. (Note that the segment elongation rates cannot be obtained directly by a projection of the vertex velocity. Consider the counter-example in which the two axon segments have identical direction (the axon is straight), and the zipper vertex moves along this direction. Then the total length of

the axon is unchanged and there is no elongation within either segment, while $\vec{u} \cdot \vec{t}_A$ and $\vec{u} \cdot \vec{t}_C$ are nonzero.)

The local elongation velocity within segment $j$ can now be obtained as $\frac{l}{L_j}\dot{L}_j$, where $l$ is the distance from the fixed point. By integration similar to the case of transverse friction, we obtain the total energy dissipation rate due to axial friction in the segment $VA$,

$$R_A^{\parallel} = \frac{1}{2}\eta^{\parallel}\left(\frac{L_A}{L}\dot{L}\right)^2 \int_0^{L_A}\left(\frac{l}{L_A}\right)^2 dl = \frac{1}{6}\eta^{\parallel}\frac{L_A^3}{(L_A + L_C)^2}(\vec{u} \cdot (\vec{t}_A + \vec{t}_C))^2$$

and similarly for the segment $VC$.

## Elongational viscosity

As discussed in the main text section Results, this friction is due to the viscosity of axon elongation. Consider again the left axon in **Figure 11A**, composed of the segments $VA$ and $VC$. The strain rate $\dot{\epsilon} = \frac{\dot{L}}{L}$ can be expressed as $\frac{\vec{u} \cdot (\vec{t}_A + \vec{t}_C)}{L_A + L_C}$ and, according to Assumption 3, is uniform in the whole axon. The rate of energy dissipation in an element $dl$ of the axon is $dR^{\updownarrow} = \frac{1}{2}\eta^{\updownarrow}\dot{\epsilon}^2 dl$, which trivially integrates to the total dissipation rate in the whole axon, $R^{\updownarrow} = \frac{1}{2}\eta^{\updownarrow}\frac{1}{L_A + L_C}(\vec{u} \cdot (\vec{t}_A + \vec{t}_C))^2$.

## Zippering friction

As discussed in the main text section Results, this friction is a phenomenological description of the dissipation processes occurring in the immediate vicinity of the zipper vertex. We assume that the corresponding dissipation rate depends only on the velocity of zippering $u^Z$, given by the projection of the vertex velocity $\vec{u}$ on the zipper axis $\vec{t}_C$ (see **Figure 11A**). Therefore, the dissipation rate is $R^Z = \frac{1}{2}\eta^Z(u^Z)^2 = \frac{1}{2}\eta^Z(\vec{u} \cdot \vec{t}_C)^2$.

## Rayleigh function

Combining the dissipation mechanisms introduced in the previous sections, the total dissipation rate for the left axon (consisting of segments $VA$ and $VC$) is

$$D = \frac{1}{2}\eta^{\parallel}\left[\frac{1}{3}\frac{L_A^3 + L_C^3}{(L_A + L_C)^2}\right](\vec{u} \cdot (\vec{t}_A + \vec{t}_C))^2 + \frac{1}{2}\eta^{\updownarrow}\frac{1}{L_A + L_C}(\vec{u} \cdot (\vec{t}_A + \vec{t}_C))^2 +$$
$$\frac{1}{2}\eta^{\perp}(\vec{u} \cdot \vec{n}_A)^2\frac{1}{3}L_A + \frac{1}{2}\eta^{\perp}(\vec{u} \cdot \vec{n}_C)^2\frac{1}{3}L_C + \frac{1}{4}\eta^Z(\vec{u} \cdot \vec{t}_C)^2 \tag{A4}$$

(where for the convenience of notation, we assigned half of the vertex-localized zippering friction to the left axon and half to the right axon).

Note that each term in **Equation A4** consists of three distinct parts, e.g.

$$R_j^{\parallel} = \frac{1}{2}\underbrace{\eta^{\parallel}}_{\text{friction constant}}\underbrace{\frac{1}{3}L_j\left(\frac{L_j}{L_A + L_C}\right)^2}_{\text{geometric factor}}\underbrace{(\vec{u} \cdot (\vec{t}_A + \vec{t}_C))^2}_{\text{velocity projection}}.$$

The geometric factor depends on the dimensions of the zipper configuration, while the velocity projection selects the component of vertex velocity $\vec{u}$ in the appropriate direction.

The whole Rayleigh function can be conveniently written in matrix notation, where the friction constants and the geometric factors are combined into a diagonal matrix $\overleftrightarrow{A}$ and the velocity projection is achieved using another matrix $\overleftrightarrow{P}$ that acts on the vertex velocity $\vec{u}$:

$$\overleftrightarrow{A} = \frac{1}{2} \begin{bmatrix} \eta^{\parallel}\left[\frac{1}{3}\frac{L_A^3+L_C^3}{(L_A+L_C)^2}\right] + \eta^{\Updownarrow}\frac{1}{L_A+L_C} & 0 & 0 & 0 \\ 0 & \eta^{\perp}\frac{1}{3}L_A & 0 & 0 \\ 0 & 0 & \eta^{\perp}\frac{1}{3}L_C & 0 \\ 0 & 0 & 0 & \frac{1}{2}\eta^Z \end{bmatrix}$$

and

$$P = \begin{bmatrix} \mathbf{t}_A + \mathbf{t}_C \\ \mathbf{n}_A \\ \mathbf{n}_C \\ \mathbf{t}_C \end{bmatrix} = \begin{bmatrix} t_{A,x} + t_{C,x} & t_{A,y} + t_{C,y} \\ n_{A,x} & n_{A,y} \\ n_{C,x} & n_{C,y} \\ t_{C,x} & t_{C,y} \end{bmatrix}$$

In this notation, the matrix of Rayleigh coefficients $K_{ij}$ for the left axon can be obtained as $\overleftrightarrow{K} = \overleftrightarrow{P}^T \overleftrightarrow{A} \overleftrightarrow{P}$. (The Rayleigh function in **Equation A4** is then reproduced as $D = \frac{1}{2}K_{ij}\dot{q}_i\dot{q}_j = (\overleftrightarrow{P}\vec{u})^T\overleftrightarrow{A}\overleftrightarrow{P}\vec{u}$.) Note that as the elements of the matrices $\overleftrightarrow{A}$ and $\overleftrightarrow{P}$ depend on the coordinates $(x,y)$ of the vertex, the Rayleigh coefficients are, in general, functions of $x$ and $y$. The matrix of Rayleigh coefficients for the whole configuration is given as the sum of the matrices for the left and right axon.

With the matrix of Rayleigh coefficients determined, the equation of motion for the zipper vertex (**Equations A2,A3**) becomes

$$\overleftrightarrow{K}^{-1}(x,y)\begin{pmatrix} x\left(\frac{T_1}{L_A(x,y)}+\frac{T_2}{L_B(x,y)}+\frac{T_1+T_2-S}{L_C(x,y)}\right) - \frac{T_1 x_A}{L_A(x,y)} - \frac{T_2 x_B}{L_B(x,y)} - \frac{(T_1+T_2-S)x_C}{L_C(x,y)} \\ y\left(\frac{T_1}{L_A(x,y)}+\frac{T_2}{L_B(x,y)}+\frac{T_1+T_2-S}{L_C(x,y)}\right) - \frac{T_1 y_A}{L_A(x,y)} - \frac{T_2 y_B}{L_B(x,y)} - \frac{(T_1+T_2-S)y_C}{L_C(x,y)} \end{pmatrix} = \begin{pmatrix} \dot{x} \\ \dot{y} \end{pmatrix}.$$

This system of two coupled nonlinear differential equations is readily solved numerically.

