## [Decision Letter]

Thank you for submitting your article "Axon tension regulates fasciculation/defasciculation through the control of axon shaft zippering" for consideration by *eLife*. Your article has been favorably evaluated by David Van Essen (Senior Editor) and three reviewers, one of whom is a member of our Board of Reviewing Editors. The reviewers have opted to remain anonymous.

The reviewers have discussed the reviews with one another and the Reviewing Editor has drafted this decision to help you prepare a revised submission.

Your paper has been well received, and all three reviewers thought it was of interest to the field, extending across subfields and disciplines. The theoretical and biophysical aspects of axon fasciculation have rarely been considered and so this study would be of great value. Please see the attached reviews.

We will consider a revised manuscript that addresses the concerns of the reviewers. Aside from the requests for textual edits and clarification of presentation for those not in theoretical biology and biophysics, the major revisions called for the inclusion of analyses that measure tension at different time points in culture, and/or interfere with tension, to demonstrate a correlation between tension and the ratio of zippering/unzippering events. We believe that these experiments would substantially strengthen the paper.

Please let us know as soon as you are able whether these requested experiments look to be feasible and if so, whether you anticipate that they can be completed in less than about two months' effort. Please be advised that your revised manuscript will be re-reviewed and acceptance is not guaranteed.

Reviewer # 1

This manuscript uses embryonic mouse olfactory epithelium explants to study axon fasciculation, namely to address the biophysics of axon shaft-shaft interactions in fasciculation/defasciculation dynamics. The work is well executed and well presented, and represents a paradigm shift in how the field thinks about and understands fasciculation, a phenomenon classically considered to occur at the level of growth cones rather than along the axon shaft.

One of the strengths of the manuscript – its relevance to researchers across disciplines – is also its main challenge. The subject matter is relevant to developmental neurobiologists and those with a biophysical background. Some sections may be either unclear in their content or significance to those in the former audience pool. For instance, most developmental neuroscientists are likely to be unfamiliar with the details and significance of foam dynamics. This part of the Discussion as well as some of the nuances of the modeling could be clarified and better summarized in order to reach a wider audience and successfully communicate the full depth of the work. That being said, the authors have largely succeeded in presenting their findings in a way that is of interest and relevance to a wide and somewhat diverse audience.

1) The estimates of force while expressed in Equation 1, should ideally be tested experimentally; this would take extra experiments that may take quite some time to execute, but would be important to perform.

2) The authors touch on the possibility of the increase in zippering resulting from progressive decrease in axon-substrate adhesion, but this idea could be fleshed out a bit more. Have substrates other than polylysine/laminin been tested? Or has the concentration of laminin been varied?

3) In the fifth paragraph of the Results, the authors indicate that entangled axon structures appear more rarely. Is it possible to roughly quantify the relative ratios of parallel, crossing, and entangled axon-axon interactions? It would be informative to provide a distribution of types of axon shaft interactions.

4) The conclusion presented in the third paragraph of the Discussion is reasonable and accurate, i.e. that axon fasciculation observed in the experiments presented are likely OR-independent. However, the experiments and results presented here do not rule out the possibility that OR expression (or other molecules in this or other systems) could change the axon shaft-shaft dynamics if observed at a different time point. Obviously directly studying the effects of different molecular expression on axon zippering behavior is beyond the scope of this study, but it would be interesting to provide a further discussion of how molecular expression might affect zippering. This is addressed somewhat in the twelfth paragraph of the Discussion, but perhaps it is worth either combining these two parts of the Discussion or fleshing out the first section somewhat.

5) Despite the thorough biophysical, computational and microscopical analyses, the question remains as to the functional significance of axon zippering along shafts observed in this culture setting.

A) Fasciculation is often studied during early development, mainly in context of a pioneer-follower axon paradigm. Are there conceivable functions relevant to circuit development that do not involve follower-axons? The authors should speculate on how dynamic axon fascicles are throughout later development and during the lifespan of a circuit. A study on *C. elegans* (Aurelio et al., 2002, Science 295: 686) describes an Ig-domain molecule important for maintenance of axon tracts. One could imagine that in some diseases or in the normal lifespan that if some axons degenerated, there might be zippering (of unmyelinated axons) in tracts to preserve integrity. In addition, after injury (transection) of axons, any regenerating axons, or the remaining axons, would "zipper up" in the fashion described. The authors are urged to read up on axon regeneration, especially the recent work of Zhigang He (Bei et al., 2016, Cell 164:219) after optic nerve crush or transection of retinal ganglion cell axons more distally, that could suggest some points of discussion in this regard.

B) The Discussion would benefit from addressing some other differences between the in vitro and in vivo systems, for instance, space constraints of axons growing along developing tracts, which are often within glial or parenchymal borders.

C) Likewise, it would be important to compare what is known about stiffness /fluidity of brain tissue (both immature, where there is much extracellular space) and mature tissue where there is little, as well as glial processes, etc. compared to a tissue culture substratum/plastic or glass.

*Reviewer #2:*

This paper presents a very elegant experimental and theoretical analysis of the biophysics of axon zippering. The data is of high quality, and the modelling is well done. The model is a nice balance of biophysical detail and simplicity. A highlight is being able to compare the high-quality images with diagrams of the model (e.g. Figures 4 and 5) which is a great aid to intuition. The annotations on the images are also very helpful in communicating the methods efficiently.

I list below a number of issues which could potentially be addressed to improve the manuscript.

1) The current manuscript does not appear to offer any new insight into the molecular mechanisms underlying fasciculation/zippering (this is not a criticism, just an observation).

2) It is perhaps not clear enough what new insight has been gained from this work about how neural wiring develops in vivo. At the very least I suggest expanding on the three points in the eleventh paragraph of the Discussion.

3) There is a lot of analysis of the topological properties of the axon network (e.g. number of vertices). This may be of mathematical interest, but the biological relevance is currently not clear. Does this network behaviour confirm or rule out particular microscopic mechanisms, or is some importance of the structure itself being claimed?

4) There are a few places where results seem to be based on very low n values. For instance, in Figure 1H the results seem to be just from one small window from one explant. It would be reassuring to know that these results generalise. More generally, the number of explants contributing to each result should always be stated.

5) Readability could be improved. Although very clearly written, two problems are as follows. (i) The manuscript is currently extremely long – about 15,000 words by my estimate, making it a bit of a marathon for the reader. (ii) Neither the Results or Discussion sections are currently split up into any clearly delineated subsections.

6) New results are introduced in the Discussion section (Figure 14): I feel these should be in the Results.

7) The reader is assumed to be familiar with the biophysics of foams. Some introductory background information in this regard would be helpful, and more discussion of what the reader should take away from this comparison.

*Reviewer #3:*

In this manuscript, Smit et al. present measurements of axon tension in a 2D culture system, observe axonal network morphologies, zippering and unzippering events of axons, and develop a mathematical model to describe the process. The biophysics of axon fasciculation has so far rarely been considered, and a thorough understanding of the process would be important to better understand the development of the nervous system.

While the presented idea is very nice, and the model looks good to me, the experiments (on which the model is based) seem to be rather preliminary and not conclusive. I have the following specific concerns with the manuscript in its current form:

1) The relation of branching angles (which are at least mathematically very similar to un/zippering angles) and axon tension has been investigated in the past. Particularly the 1979 paper from Dennis Bray and Shefi at al., Neurocomputing, 2004, which already present mathematical models to connect tension and axon morphology / branching angles, need to be cited.

2) The measurement of axon tension (only in static neurons) using the BFP technique is not ideal. Biotinylation of axons might already change their mechanical properties, calibration of the force probe (an RBC) seems to be a bit crude and not very accurate (how large is the estimated error of the measurements?), numbers of experiments are very small, and a dynamic system could not be investigated with it. To validate the model, and to directly test the relation between tension and zippering angle, one would have to measure the development of tension during zippering and unzippering events, which really should be done. This would also show if un/zippering of axons in the network is primarily because of a change in tension or adhesion. Other methods (such as magnetic tweezers or the analysis of membrane fluctuations, see Garate et al., Phys Biol, 2015) might be better suited for such measurements.

3) The force balance in Equation 1 does not seem to take the force required to deform / bend the axons into account. I'm not sure if this is an oversimplification.

4) An important experiment, which is currently missing but which would be required to test Equation 1, is the direct measurement of the adhesion force. This should be done. Again, potentially suitable techniques include magnetic tweezers, calibrated microneedles, or AFM.

5) Ideally, the system should also be experimentally perturbed to test the model. For example, tension could be manipulated by interfering with contractility (e.g., using blebbistatin), and adhesion using knockdown of cell adhesion molecules.

6) It appears that the zippering process is considered a purely passive event. Active cellular processes, such as transport etc., will, however, also be involved. This should at least be discussed.

7) The interpretation of the development of zippering angles shown in Figure 12C is not supported by the data shown. For example, the statement "the mean angle decreased slightly from 62° to 56°" suggests a steady decrease over time, which, however, is not the case. The distributions at 60 minutes and 180 minutes look very similar to me. The statement "The predicted decrease in mean zipper angle is thus 7.5° (marked in Figure 13), in good agreement with the experimentally observed change of 6° (Figure 12)" doesn't work for that reason – it is only 1° difference after 2 hours (if comparing 60 mins and 180 mins), which might very well be noise. I doubt that a statistical analysis (which should be done) would reveal a low P value. Also, the boxplots actually suggest that the distribution is more asymmetric at 180 mins than at 60 mins – so I'm not convinced by the usefulness of the skewness parameter s. N is very low here as well for a proper statistical analysis. Together, I don't see the trend the authors mention, and I strongly recommend them to reconsider the interpretation of these experiments.

[Editors' note: further revisions were requested prior to acceptance, as described below.]

Thank you for resubmitting your work entitled "Axon tension regulates fasciculation/defasciculation through the control of axon shaft zippering" for further consideration at *eLife*. Your revised article has been favorably evaluated by David Van Essen (Senior Editor), a Reviewing Editor and two reviewers.

The two reviewers found that you have addressed the bulk of their comments and that your revised manuscript is improved, but there are some remaining issues that need to be addressed before acceptance, as outlined below.

One of the reviewers had a remaining concern – to test your model experimentally and directly measure using the BFP technique the change in tension following the application of blebbistatin. The hope was that you could have related the changes you observed in network morphology to changes in tension.

When the Senior Editor asked on February 25 whether this was possible, you wrote in a letter on March 3 that several technical and logistical factors made it extremely difficult to make the requested tension measurements. You explained why the experiment would not be possible at present, and provided significant new information that further strengthened your case.

In our discussion online, reviewer 3 further commented that in published experiments, blebbistatin decreases tension and agrees that there is currently no study showing that this is also true for axons. He pointed out that if axonal tension was decreased after treatment, then the network would not behave as the model predicts, and therefore, this reviewer continued to feel that it would be important to show how blebbistatin impacts axonal tension. However, he now acknowledges, based on your response letter, that this experiment is not feasible in your lab.

We are therefore pleased to inform you that we will proceed with your manuscript without including the force measurements.

We request revisions that would incorporate the data and arguments in your rebuttal letter:

1) Comment on the blebbistatin experiments, as above.

2) Discuss the possibility (and caveats) of using thermal fluctuation spectroscopy).

3) Highlight your (novel) use of embryonic olfactory epithelium explants, highlighting that this preparation presents "the right balance of axon tension and axon-axon adhesion" that provides insight into axon zippering and progressive fasciculation.

4) Add from your response letter the figure on FBS-induced increase of axon tension, as Figure 5—figure supplement 1.

5) Add the data from the figure on cytochalasin-induced reduction of axon tension, pointing out, as per your prediction, that the cytochalasin-induced tension reduction was expected to be comparable for all directions within the network, in contrast to the FBS-induced pull in one dominant direction.

6) Strengthen the "take-home" message that your results demonstrate that the extent of axon zippering can be controlled by axon tension forces, and that future work could aim for force measurements to directly verify the increase in tension in the FBS experiments and the decrease in tension in the cytochalasin experiments.

7) Throughout the manuscript, stress that axon zippering is controlled by tension changes of functionally relevant magnitude.

We all agree that the paper shows a fundamentally new mechanism of axon fasciculation via axon zippering, and that it will be an important contribution to the field.

*Reviewer #2:*

I am happy with how the authors have addressed my comments, and have nothing further to add.

*Reviewer #3:*

The authors have addressed most points and have overall improved the manuscript.

However, they fail to provide experimental evidence confirming their model. Particularly the experiments aiming to perturb network tension are inconclusive: As the authors state in their rebuttal, the question here was to find a means to experimentally perturb axon tension. However, I do not see any measurement of axon tension in the perturbed systems (which could be done on single axons, bypassing the technical limitations of BFP), so how do we know if and how the treatments changed axon tension? What if, for example, axon tension does drop after blebbistatin treatment and yet the network zippering does not change as predicted? This would contradict the model. Thus, these experiments would only be meaningful is axon tension was measured.

I am also a bit puzzled by the BSA experiments. It is not clear to me what BSA is doing, or how a cell (body) rounding should lead to the contraction of the whole network? How do the authors know that FBS induces strong pulling of axons? Tension was not measured. Instead, FBS could, for example, simply interfere with adhesion, which could easily cause the observed retraction and / or de-coarsening without changing tension at all. I don't think that this approach is very helpful. Combining it with blebbistatin treatment, whose effect on the network is also not understood, does not help.

As a minor point, I do not understand the motivation of the authors to use drugs known to interfere with growth cone motility. Is growth cone motility coupled to axon tension? In none of the figures presented growth cones are shown, so they are probably not relevant for the zippering / unzippering of the established network?

What is shown in the new Figure 5 are a few images of parts of a network. Without proper quantification, it is very difficult to say what the effect of the treatments is. For example, the authors state that blebbistatin has a stabilizing effect on the network; however, Figure 6G-I indicate that the network is behaving the same way in treated and untreated cultures before cytochalasin treatment?

Finally, I still think that it would be important to directly measure the adhesion force to test Equation 1 (which was also stated by reviewer 1).

---

## [Author Response]

*[…] Reviewer #1:*

This manuscript uses embryonic mouse olfactory epithelium explants to study axon fasciculation, namely to address the biophysics of axon shaft-shaft interactions in fasciculation/defasciculation dynamics. The work is well executed and well presented, and represents a paradigm shift in how the field thinks about and understands fasciculation, a phenomenon classically considered to occur at the level of growth cones rather than along the axon shaft.

*One of the strengths of the manuscript – its relevance to researchers across disciplines – is also its main challenge. The subject matter is relevant to developmental neurobiologists and those with a biophysical background. Some sections may be either unclear in their content or significance to those in the former audience pool. For instance, most developmental neuroscientists are likely to be unfamiliar with the details and significance of foam dynamics. This part of the Discussion as well as some of the nuances of the modeling could be clarified and better summarized in order to reach a wider audience and successfully communicate the full depth of the work. That being said, the authors have largely succeeded in presenting their findings in a way that is of interest and relevance to a wide and somewhat diverse audience.*

We thank reviewer 1 for this very positive evaluation of our paper.

We recognize that, in the first version of the manuscript, some of the biophysical sections were not easy to fully understand for most developmental neurobiologists. In revising the text of the manuscript, we paid close attention to this point.

We consider that our new presentation of the relation to foam dynamics (subsection “Topological analogies between progressive axon fasciculation and the coarsening of liquid foams”) is now fully accessible to a wide audience including neuroscientists, thanks to a didactic text and clear figure illustrations (Figure 13). Likewise, we clarified some important aspects of the modeling in order to allow non-specialists to more readily understand our approach.

*1) The estimates of force while expressed in Equation 1, should ideally be tested experimentally; this would take extra experiments that may take quite some time to execute, but would be important to perform.*

Indeed, while starting work on this project, we had planned to directly measure the adhesion force (denoted S in Equation 1) by unzippering axon shafts using a calibrated force. In conjunction with the determined tension T (using BFP or another method), this would have permitted the direct test of Equation 1 on the level of an individual zipper. Unfortunately, this procedure was not possible to carry out using the BFP technique at our disposal, as described in the manuscript. Instead, we were able to unzipper axon shafts using an uncalibrated force manipulation (as described previously in the originally submitted manuscript); the results of these induced unzippering experiments are consistent with Equation 1, but do not provide a direct quantitative test of it.

Given the limited time we had to carry out the revisions, and the expected longer time frame that would be needed to implement an alternative calibrated manipulation setup for our system, we gave priority to the pharmacological manipulations of axon tension.

*2) The authors touch on the possibility of the increase in zippering resulting from progressive decrease in axon-substrate adhesion, but this idea could be fleshed out a bit more. Have substrates other than polylysine/laminin been tested? Or has the concentration of laminin been varied?*

We indeed envisaged modifying axon-substrate adhesion because it would have allowed us to test directly the consequences on zippering. However, given the focus of our studies – namely the coarsening of a previously established network, we were concerned that changing the substrate of our culture would be likely to lead to changes in the initial development of the axon network. Hence, it would have been very difficult to establish a causal link between a new substrate with different adhesive properties, and possible changes in the dynamics of coarsening through zippering. Moreover, very little is known about the molecular mechanisms involved in OSN axon/substrate adhesion in culture. Thus, the extent to which such substrate change (or changes in laminin concentration) would change the axon/substrate adhesion would have been difficult to assess. It would require a large body of experiments that included many different substrate conditions to find conditions in which adhesion would have seem to be changed while the initial establishment of the network would have been unchanged.

To bypass these difficulties, we used trypsin to induce a modification of adhesion once the network was developed on our polylysin/laminin substrate. This treatment rapidly degrades extracellular proteins, among them cell-cell and cell-substrate adhesion proteins. Unfortunately, the trypsin treatments induced a rapid collapse of the network (retraction onto the explant due to detachment of axons from the substrate), before any visible change in axon-axon adhesion. This experiment thus nicely confirmed that the entire network was under tension, but did not allow us to study changes in zippering.

For all these reasons, as we undertook new experiments in the perspective of revising this paper, we decided to focus our efforts into experiments aiming at modifying axon tension.

*3) In the fifth paragraph of the Results, the authors indicate that entangled axon structures appear more rarely. Is it possible to roughly quantify the relative ratios of parallel, crossing, and entangled axon-axon interactions? It would be informative to provide a distribution of types of axon shaft interactions.*

We thank reviewer 1 for this suggestion. We have now quantified the relative abundance of simple and entangled zippers, as well as axon crossings, based on electron micrographs of the network configuration. We now provide these data in the manuscript, and show the underlying analysis in a supplementary figure (Figure 4—figure supplement 1).

*4) The conclusion presented in the third paragraph of the Discussion is reasonable and accurate, i.e. that axon fasciculation observed in the experiments presented are likely OR-independent. However, the experiments and results presented here do not rule out the possibility that OR expression (or other molecules in this or other systems) could change the axon shaft-shaft dynamics if observed at a different time point. Obviously directly studying the effects of different molecular expression on axon zippering behavior is beyond the scope of this study, but it would be interesting to provide a further discussion of how molecular expression might affect zippering. This is addressed somewhat in the twelfth paragraph of the Discussion, but perhaps it is worth either combining these two parts of the Discussion or fleshing out the first section somewhat.*

The Discussion has now been extensively reorganized, and it includes a full subsection dedicated to the regulation and functional significance of axon zippering in vivo, in which this issue is addressed.

*5) Despite the thorough biophysical, computational and microscopical analyses, the question remains as to the functional significance of axon zippering along shafts observed in this culture setting.*

We recognize that these aspects were not discussed enough in the first version of the paper. As reviewer 1 will see, this very important point is now extensively discussed in the “Axon zippering in vivo: its regulation and functional significance” subsection of the Discussion.

*A) Fasciculation is often studied during early development, mainly in context of a pioneer-follower axon paradigm. Are there conceivable functions relevant to circuit development that do not involve follower-axons? The authors should speculate on how dynamic axon fascicles are throughout later development and during the lifespan of a circuit. A study on C. elegans (Aurelio et al., 2002, Science 295: 686) describes an Ig-domain molecule important for maintenance of axon tracts. One could imagine that in some diseases or in the normal lifespan that if some axons degenerated, there might be zippering (of unmyelinated axons) in tracts to preserve integrity. In addition, after injury (transection) of axons, any regenerating axons, or the remaining axons, would "zipper up" in the fashion described. The authors are urged to read up on axon regeneration, especially the recent work of Zhigang He (Bei et al., 2016, Cell 164:219) after optic nerve crush or transection of retinal ganglion cell axons more distally, that could suggest some points of discussion in this regard.*

We are very grateful to reviewer 1 for suggesting these specific points and papers. The work from Aurelio indeed provides a wonderful example illustrating how axon shafts may zipper in vivo, in this case with detrimental consequences, and further shows how inhibition of axon/axon adhesion negatively regulates zippering. This paper is now discussed in this section. We also briefly speculate on the possible consequences of zippering in pathological conditions, and upon regeneration. However, it was unclear to us how to directly link our framework to the paper by Bei et al.

*B) The Discussion would benefit from addressing some other differences between the* in vitro *and* in vivo *systems, for instance, space constraints of axons growing along developing tracts, which are often within glial or parenchymal borders.*

This was indeed a great suggestion, and we have now addressed these differences between the in vitro and in vivo system, including space constraints, and with a particular attention on the glia/axon interactions in peripheral and central systems (subsection “Axon zippering in vivo: its regulation and functional significance”, third paragraph).

*C) Likewise, it would be important to compare what is known about stiffness /fluidity of brain tissue (both immature, where there is much extracellular space) and mature tissue where there is little, as well as glial processes, etc. compared to a tissue culture substratum/plastic or glass.*

Indeed, the changes in tissue stiffness may have important consequences for axon zippering. We now discuss this aspect in a newly added paragraph in the eighth paragraph of the subsection “Axon zippering in vivo: its regulation and functional significance”.

*Reviewer #2:*

*This paper presents a very elegant experimental and theoretical analysis of the biophysics of axon zippering. The data is of high quality, and the modelling is well done. The model is a nice balance of biophysical detail and simplicity. A highlight is being able to compare the high-quality images with diagrams of the model (e.g. Figures 4 and 5) which is a great aid to intuition. The annotations on the images are also very helpful in communicating the methods efficiently.*

*I list below a number of issues which could potentially be addressed to improve the manuscript.*

*1) The current manuscript does not appear to offer any new insight into the molecular mechanisms underlying fasciculation/zippering (this is not a criticism, just an observation).*

We agree that we did not provide any new insights into the molecular mechanisms underlying fasciculation/zippering. This was out of the scope of this paper which rather aims at documenting, analysing, and modeling the dynamical zippering process and its role in axon shaft fasciculation, both in vitro and in vivo. All these aspects are new and they provide a paradigm shift in the field of axon fasciculation, as stated by reviewer 1.

*2) It is perhaps not clear enough what new insight has been gained from this work about how neural wiring develops in vivo. At the very least I suggest expanding on the three points in the eleventh paragraph of the Discussion.*

As suggested by reviewer 2 and 1, we have reorganized extensively our Discussion, and have included a subsection devoted to the regulation and functional significance of zippering in vivo. In this subsection, we indeed expanded the paragraph mentioned by reviewer 2.

*3) There is a lot of analysis of the topological properties of the axon network (e.g. number of vertices). This may be of mathematical interest, but the biological relevance is currently not clear. Does this network behaviour confirm or rule out particular microscopic mechanisms, or is some importance of the structure itself being claimed?*

The quantification of network characteristics was instrumental for evaluating the overall outcome of the individual zippering and unzippering processes. Using these characteristics, we were, for example, able to clearly demonstrate that cytochalasin (which lowers tension) promotes fasciculation (new Figure 6G-I). Individual zippers are highly dynamic and often alternate between zippering and unzippering periods, depending on the network rearrangements occurring in their vicinity. The network structure provides a robust reading of the overall tendency to fasciculate or defasciculate (as opposed to attempting to track all individual zippers).

At this point we do not claim a particular biological significance of the network structure, beyond the formation of bigger or smaller fascicles. Such potential significance would need to be evaluated in the context of a particular in vivo system.

*4) There are a few places where results seem to be based on very low n values. For instance, in Figure 1H the results seem to be just from one small window from one explant. It would be reassuring to know that these results generalise. More generally, the number of explants contributing to each result should always be stated.*

In the revised manuscript, we now clearly state the number of explants contributing to each result. For example, see subsection “Progressive fasciculation in cultures of olfactory epithelium explants is due to axon shaft zippering”, second paragraph; subsection “Manipulation of axon tension alters the relative abundance of zippering and unzippering”, first paragraph; subsection “Measurement of axon tension allows to estimate the axon-axon adhesion energy”, third and seventh paragraphs; subsection “Induced or spontaneous dynamics of individual zippers”, last paragraph; subsection “Progressive fasciculation is reflected in the network distribution of zipper angles”, second paragraph; subsection “Analysis of the videomicroscopy recordings”, first and last paragraphs. Moreover, the total number of analyzed experiments has been very significantly expanded for the revised submission.

*5) Readability could be improved. Although very clearly written, two problems are as follows. (i) The manuscript is currently extremely long – about 15,000 words by my estimate, making it a bit of a marathon for the reader. (ii) Neither the Results or Discussion sections are currently split up into any clearly delineated subsections.*

We are grateful to reviewer 2 for this excellent suggestion. We have now introduced subsections in both the Results and Discussion sections, with titles summarizing the main message of each subsection, which indeed improves significantly the readability of the manuscript.

We have also reduced the length of some subsections, notably on the BFP experiments (particularly in Methods). We have also removed one figure (former Figure 12) and most of the associated text from the manuscript; it is replaced by a simplified analysis presented as part of the current Figure 14. As a result, the corresponding subsection is substantially shortened.

*6) New results are introduced in the Discussion section (Figure 14): I feel these should be in the Results.*

We agree with the reviewer. The new organization of the paper accommodates this change in a natural way. These results now form the last subsection of Results, allowing us to finish the Results with an analysis of zippering in vivo, and to clearly distinguish it from the discussion of the functional role of zippering in vivo in the corresponding subsection of Discussion. (Please note that the former Figure 14 is now Figure 15).

*7) The reader is assumed to be familiar with the biophysics of foams. Some introductory background information in this regard would be helpful, and more discussion of what the reader should take away from this comparison.*

We acknowledge that an introduction of foams for non-physicist readers was missing, and that the relevance of the discussion of foams in the context of this work was not sufficiently highlighted. As reviewer 2 will see, we have now remedied this by preparing a subsection of the Discussion (“Topological analogies between progressive axon fasciculation and the coarsening of liquid foams”) which introduces the foam structure and dynamics, discusses the topological and dynamical analogies to our system (including the remarks that had been made in the Results section of the original manuscript), and makes a comparison to other biological literature in which analogies to foams were invoked.

*Reviewer #3:*

*In this manuscript, Smit et al. present measurements of axon tension in a 2D culture system, observe axonal network morphologies, zippering and unzippering events of axons, and develop a mathematical model to describe the process. The biophysics of axon fasciculation has so far rarely been considered, and a thorough understanding of the process would be important to better understand the development of the nervous system.*

*While the presented idea is very nice, and the model looks good to me, the experiments (on which the model is based) seem to be rather preliminary and not conclusive. I have the following specific concerns with the manuscript in its current form:*

*1) The relation of branching angles (which are at least mathematically very similar to un/zippering angles) and axon tension has been investigated in the past. Particularly the 1979 paper from Dennis Bray and Shefi at al., Neurocomputing, 2004, which already present mathematical models to connect tension and axon morphology / branching angles, need to be cited.*

Indeed the condition of force balance at a branching point, formulated by Bray and followers, is related to our Equation 1. We now acknowledge this when Equation 1 is introduced, and explicitly discuss the differences as compared to the case of an axon zipper. We also refer to additional literature in which this force balance at a branching point was experimentally tested.

*2) The measurement of axon tension (only in static neurons) using the BFP technique is not ideal. Biotinylation of axons might already change their mechanical properties, calibration of the force probe (an RBC) seems to be a bit crude and not very accurate (how large is the estimated error of the measurements?), numbers of experiments are very small, and a dynamic system could not be investigated with it. To validate the model, and to directly test the relation between tension and zippering angle, one would have to measure the development of tension during zippering and unzippering events, which really should be done. This would also show if un/zippering of axons in the network is primarily because of a change in tension or adhesion. Other methods (such as magnetic tweezers or the analysis of membrane fluctuations, see Garate et al., Phys Biol, 2015) might be better suited for such measurements.*

While the BFP technique did not allow us to perform dynamic manipulations of zippers, we consider the measurements of axon tension we performed with BFP to be reliable. The error of calibration of the BFP probe is about 15%, as estimated by us and as stated in previous BFP literature. This error is lower than the variability of tensions recorded in our set of measurements from 7 axons. All the errors contributing to BFP measurements are quantified in the Methods (subsection “Analysis of BFP data”, third paragraph). We wish to point out that the BFP technique has been successfully used in numerous previous biophysical investigations. Please also see the methodological paper of (Simson et al., Biophysical Journal 74 (1998) p. 2080–2088), in which the BFP technique was analyzed in detail and verified using data obtained with magnetic tweezers.

We did attempt to use optical tweezers at an early stage of our experiments, but encountered the same difficulty as with BFP – the force was insufficient to significantly move the axons. Moreover, this method was likewise dependent on biotinylation, and was significantly less flexible than the micropipette-based BFP.

Due to material and time constraints, we have been unable to implement an alternative method of probing the tension, although we agree with the reviewer that it would be of benefit.

*3) The force balance in Equation 1 does not seem to take the force required to deform / bend the axons into account. I'm not sure if this is an oversimplification.*

Indeed, the bending force was neglected in the model. As we argued in the original manuscript, this simplification is justified for zippers consisting of single axons or thin fascicles. In the revised manuscript, we expand on this point in the second paragraph of the subsection “Dynamical model of axon zippering”. We make a theoretical estimate of the bending force, showing that it is roughly one order of magnitude lower than the forces we include in the model, for a zipper made of single axons. We then use a scaling argument to show that the bending force will play an important role only for fascicles containing more than about 10 axons.

*4) An important experiment, which is currently missing but which would be required to test Equation 1, is the direct measurement of the adhesion force. This should be done. Again, potentially suitable techniques include magnetic tweezers, calibrated microneedles, or AFM.*

Please refer to the response to point 1 of reviewer 1.

*5) Ideally, the system should also be experimentally perturbed to test the model. For example, tension could be manipulated by interfering with contractility (e.g., using blebbistatin), and adhesion using knockdown of cell adhesion molecules.*

We thank reviewer 3 for his suggestion to experimentally perturb the system to test the model. Regarding adhesion, OSNs express many adhesion molecules in vivo, but the expression of these molecules is not yet characterized in vitro. Therefore, targeting specific adhesion molecules would have required first characterizing their expression in our system (an approach requiring a significant amount of time) before developing knock-down strategies, which furthermore would not be obvious. Using RNA interference through transfection would be the fastest way to knock-down a candidate adhesion molecule, but the knock-down would concern only a subset of neurons in such explant cultures. Viral vectors would be likely to transduce higher proportions of OSNs in the explants, but producing viruses requires time consuming constructs preparation and viral production. Taking into account the fact that we could not target specific adhesion molecules, we attempted to decrease global adhesion through proteolysis of extracellular proteins using trypsin treatment. Unfortunately, such trypsin treatments induced a rapid collapse of the network, as described in the answer to point 2 of reviewer 1. We thus decided to concentrate our efforts onto manipulations allowing to perturb axon tension.

We are grateful to reviewer 3 for suggesting the use of blebbistatin to interfere with axon contractility. It turns out that blebbistatin's main effect, in our system, was to stabilize the network dynamics, inhibiting the coarsening of the network while the individual zippers remained mobile. However, blebbistatin did not have any significant effect on axon outgrowth in our system, in contrast to its previously reported effects on dorsal root ganglion axons, for example, which were either positive or negative, depending of the studies and experimental conditions (see for details the second paragraph of the subsection “Structure and dynamics of the axon network in light of the zippering framework” in Discussion). We have also tested several other treatments aiming at interfering with axon tension, including a biologically-derived solution, the foetal bovine serum (FBS). The rationale for testing FBS was as follows. We first sought specific biological molecules to boost axon growth, motility or contractility in our system, but it was challenging because guidance cues or growth factors having such effects in vitro on axons growing from olfactory epithelia explants are not yet known. FBS, known to contain a number growth factors, appeared to be an interesting alternative candidate, considering that some of its active molecules may have a boosting effect on growth cone motility, which may increase their pulling force onto shafts. Even though we were aware of the fact that we would have no clue about the identity of the active molecule in the event that we did find an interesting effect on zippering, we considered that this would not be an issue in this particular context. The question here was not to *identify* a biological molecule having a specific effect, but to find a means to experimentally perturb axon tension. It turned out that FBS did not have any significant effect on OSN growth cones but, very interestingly, it induced a rapid contraction of the whole explant itself, leading to the pulling of axons by the explant core, leading to the de-coarsening of the network (Figure 5A-D). As we discuss in the paper (subsection “Structure and dynamics of the axon network in light of the zippering framework”, third paragraph), such a serum-induced “rounding” effect, already observed by Jalink and Moolenaar (1992) on cultured differentiating neural cells, may be due to LPA (Jalink et al., 1993), but we did not attempt to characterize further this phenomenon in the limited time we had for these revisions.

Often, however, FBS-induced strong pulling of axons resulted in a rapid collapse of the network, probably due to the pull-generated disturbance of the axon attachment onto the substrate. Having observed that blebbistatin was able to stabilize the network (see above), we used it as a stabilizer *before* treating it with FBS, allowing us to observe repeatedly a reliable de-coarsening induced by the explant contraction- dependent pulling of axons (as illustrated in Figure 5E-L).

As a complement to the observed decoarsening induced by a pulling force, we sought to decrease the tension of axons in view of testing whether it would enhance coarsening, as expected from our framework. Because Dennerll et al. (1988) previously showed that cytochalasin, an inhibitor actin polymerization, significantly decreases the tension of PC-12 cell neurites, we selected this drug for this purpose. We observed that cytochalasin B changed the network dynamics in a way which is consistent with a drop of average axon tension, leading to coarsening (Figure 6A-C). As the networks generally have a tendency to coarsen in our experimental paradigm, we sought to better isolate the effect of cytochalasin by applying it to networks that were pre-stabilized by blebbistatin. As shown in Figure 6D-I, cytochalasin B induces strong network coarsening within 30 minutes of application.

Overall, we have shown in the course of these new experiments that FBS-dependent pulling of axons induced unzippering and de-coarsening, while treating the network with a drug known to decrease neurite tension increased coarsening (i.e., zippering became more dominant over unzippering). These results are in full agreement with our core framework, in which axon shaft fasciculation is controlled by the competition between tension and adhesion.

*6) It appears that the zippering process is considered a purely passive event. Active cellular processes, such as transport etc., will, however, also be involved. This should at least be discussed.*

Indeed, this is an important point to acknowledge. We now discuss it in the third paragraph of the subsection “Dynamical model of axon zippering”.

*7) The interpretation of the development of zippering angles shown in Figure 12C is not supported by the data shown. For example, the statement "the mean angle decreased slightly from 62° to 56°" suggests a steady decrease over time, which, however, is not the case. The distributions at 60 minutes and 180 minutes look very similar to me. The statement "The predicted decrease in mean zipper angle is thus 7.5° (marked in Figure 13), in good agreement with the experimentally observed change of 6° (Figure 12)" doesn't work for that reason – it is only 1° difference after 2 hours (if comparing 60 mins and 180 mins), which might very well be noise. I doubt that a statistical analysis (which should be done) would reveal a low P value. Also, the boxplots actually suggest that the distribution is more asymmetric at 180 mins than at 60 mins – so I'm not convinced by the usefulness of the skewness parameter s. N is very low here as well for a proper statistical analysis. Together, I don't see the trend the authors mention, and I strongly recommend them to reconsider the interpretation of these experiments.*

In the revised manuscript, we have substantially improved this part of the paper. We have now analyzed a total of 5 experiments of the type originally presented in Figure 12. Rather than relying on a few time points from one experiment as in the original Figure 12, we now perform a correlation analysis of the relation between the median zipper angle and the network length, based on a much larger number of data points. Overall, we find a trend that is consistent with the originally stated behavior of the mean zipper angle. For the skewness, however, the originally stated behavior is not confirmed (the correlation is not significant). We now refrain from claiming any overall trend for changes in the *shape* of the distribution. We have updated and simplified the corresponding text, and removed the original Figure 12 (while retaining some data from it in the current Figure 14). We thank the reviewer for bringing us to improve the robustness of this analysis.

[Editors' note: further revisions were requested prior to acceptance, as described below.]

*[…] We request revisions that would incorporate the data and arguments in your rebuttal letter:*

*1) Comment on the blebbistatin experiments, as above.*

In line with the comment of reviewer 3, we now introduce blebbistatin in Results as a drug previously shown to reduce tension in non-neuronal cells (subsection “Manipulation of axon tension alters the relative abundance of zippering and unzippering”, third paragraph). We then comment on the blebbistatin experiments more extensively in Discussion, subsection “Structure and dynamics of the axon network in light of the zippering framework”, third paragraph. There we cite experiments from the literature on non-neuronal cells, in which a blebbistatin-induced decrease in tension was shown, but also discuss experiments from the literature on neuronal growth cones, in which some variable effects of blebbistatin were seen.

*2) Discuss the possibility (and caveats) of using thermal fluctuation spectroscopy).*

We have added a new paragraph in Discussion (subsection “Structure and dynamics of the axon network in light of the zippering framework”, last paragraph) in which we acknowledge that we did not monitor axon tension changes during the network development and zippering, and explain the potential use of thermal fluctuation spectroscopy for such a task, including the strengths and caveats of this method. Traction force microscopy is also briefly discussed.

*3) Highlight your (novel) use of embryonic olfactory epithelium explants, highlighting that this preparation presents "the right balance of axon tension and axon-axon adhesion" that provides insight into axon zippering and progressive fasciculation.*

We now highlight the novel use of embryonic olfactory epithelium explants. This is done both in the Abstract and in the first paragraph of Discussion, stressing the right balance of axon tension and axon-axon adhesion forces.

4) Add from your response letter the figure on FBS-induced increase of axon tension, as Figure 5—figure supplement 1.

This figure has been added as Figure 5—figure supplement 1, together with the corresponding source data file. The results shown in the figure are now described in the corresponding section of the Results (subsection “Manipulation of axon tension alters the relative abundance of zippering and unzippering”, fifth paragraph). The corresponding quantitative estimate of the increase in tension is given in Discussion (subsection “Structure and dynamics of the axon network in light of the zippering framework”, fifth paragraph).

*5) Add the data from the figure on cytochalasin-induced reduction of axon tension, pointing out, as per your prediction, that the cytochalasin-induced tension reduction was expected to be comparable for all directions within the network, in contrast to the FBS-induced pull in one dominant direction.*

We now describe these observations in the second paragraph of the subsection “Structure and dynamics of the axon network in light of the zippering framework”.

*6) Strengthen the "take-home" message that your results demonstrate that the extent of axon zippering can be controlled by axon tension forces, and that future work could aim for force measurements to directly verify the increase in tension in the FBS experiments and the decrease in tension in the cytochalasin experiments.*

This take-home message is now stressed more in several places, including the end of Introduction. The force measurements to verify the pharmacologically induced changes are discussed in connection with point 2 (see above).

*7) Throughout the manuscript, stress that axon zippering is controlled by tension changes of functionally relevant magnitude.*

This is now stressed in a new sentence in the last paragraph of the Introduction and also in two places in Discussion: in the paragraph following the discussion of the FBS experiment (subsection “Structure and dynamics of the axon network in light of the zippering framework”, sixth paragraph), and in the paragraph discussing changes in traction force on substrates of differing stiffnesses (subsection “Axon zippering in vivo: its regulation and functional significance”, eighth paragraph).